# On the Future Role of the Most Parsimonious Climate Module in Integrated Assessment

Mohammad M. Khabbazan and Hermann Held

Research Unit Sustainability and Global Change, Center for Earth System Research and Sustainability, Universität Hamburg, Grindelberg 5, 20144 Hamburg, Germany

*Correspondence to*: Mohammad M. Khabbazan (mohammad.khabbazan@uni-hamburg.de)

**Abstract.** In the following, we test the validity of a 1-box climate model as an emulator for Atmosphere-Ocean General Circulation Models (AOGCMs). The 1-box climate model is currently employed in the integrated assessment models FUND, MIND and PAGE, widely used in policy making. Our findings are two-fold. Firstly, when directly prescribing AOGCMs' respective equilibrium climate sensitivities (ECSs) and transient climate responses (TCRs) to the 1-box model, global mean temperature (GMT) projections are generically too high by 0.5 K at peak temperature for peak-and-decline forcing scenarios compatible with a maximum global warming of 2 K. Accordingly, corresponding integrated assessment studies might tend to overestimate mitigation needs and costs. We semi-analytically explain this discrepancy as resulting from the information loss resulting from the reduction of complexity. Secondly, the 1-box model offers a good emulator of these AOGCMs (accurate to within 0.1K for representative concentration pathways (RCPs), namely RCP2.6, RCP4.5, and RCP6.0), provided their ECS and TCR values are universally mapped onto effective 1-box counterparts and a certain time horizon (on the order of the time to peak radiative forcing) is not exceeded. Accordingly, we offer a method to re-interpret already published works based on the 1-box model. Results that are based on the model and have already been published are still just as informative as intended by their respective authors; however, they should be re-interpreted as being influenced by a larger climate response to forcing than intended.

**Keywords:** climate sensitivity, emulator, integrated assessment, mitigation scenarios, reduced climate models

## 1 Introduction

Climate-economy integrated assessment models (IAMs) are used to derive welfare-optimal climate policy scenarios (Kunreuther et al., 2014) or constrained welfare-optimal scenarios that comply with a prescribed policy target (Clarke et al., 2014). Most of them employ relatively simple climate modules emulating sophisticated climate models, Atmosphere-Ocean General Circulation Models (AOGCMs). These climate modules (hereafter: 'simple climate models' (SCMs)) offer computational efficiency and hence allow researchers to project a broader set of scenarios in orders of magnitude less time. For IAMs based on a decision-analytic framework involving intertemporal welfare optimization, SCMs are in fact

indispensable, as these IAMs' numerical solvers may need to access the climate module anywhere from ten to one hundred thousand times before numerical convergence is flagged.

The need to qualify the degree of accuracy with which SCMs mimic AOGCMs or properly represent ensembles of AOGCMs is increasingly being recognized (Calel & Stainforth, 2017; van Vuuren et al., 2011a), as this aspect might have immediate

monetary consequences in connection with derived policy scenarios (Calel & Stainforth, 2017). In previous work, Van Vuuren et al. (2011a) found that IAMs tend to underestimate the effects of greenhouse gas emissions.

Due to the centennial-scale quasi-linear properties of AOGCMs' global mean temperature (GMT) dynamics, SCMs have proven capable of emulating AOGCMs' behavior regarding GMT change, deviations being a function of spread of forcing, SCM complexity (Meinshausen et al., 2011a) and quality of SCM calibration. The climate component of MAGICC

(Meinshausen et al., 2011a) represents the most complex SCM currently in use. In some sense one could even call MAGICC an Earth System Model of Intermediate Complexity. It has demonstrated its capacity to emulate all AOGCMs' GMT even more precisely than the standard deviation of interannual GMT variability (Meinshausen et al., 2011a), with a fixed set of parameters, utilized for the whole range of representative concentration pathways (RCPs) (see van Vuuren et al., 2011b). This represents the current gold standard of AOGCM emulation using SCMs.

The most extreme opposite end of the scale of complexity within the model category of SCMs is provided by the 1-box model as introduced by Petschel-Held et al. (1999) (hereafter: 'PH99'), converting a radiative forcing time series into a GMT time series. The current role of this model as assessed in the literature is as follows: by fitting PH99 to GMT time series, it can be used as a diagnostic instrument, as Andrews & Allen (2008) have done. However, its main application is as an emulator of AOGCMs. In conjunction with the most parsimonious carbon cycle model (described in Petschel-Held et al. (1999) as well),

PH99 has been used to derive 'admissible' greenhouse gas emission scenarios in view of prescribed GMT targets (Bruckner et al., 2003; Kriegler & Bruckner, 2004). Furthermore, the following climate-economic IAMs are currently utilizing PH99: FUND (Anthoff & Tol, 2014), MIND (Edenhofer et al., 2005) and PAGE (Hope, 2006) – the last of which was used in the 'Stern Review' to the UK government (Stern, 2007). While MIND has since been succeeded by the IAM REMIND (Luderer et al., 2011) when it comes to spatial resolution or representing the energy sector by dozens of technologies, it currently serves

as a state-of-the-art IAM for decision-making under uncertainty (Held et al., 2009; Lorenz et al., 2012; Neubersch et al., 2014; Roth et al., 2015) or joint mitigation-solar radiation management analyses (Roshan et al., 2018; Stankoweit et al., 2015).

Kriegler and Bruckner (2004) validated PH99 in conjunction with a simple carbon cycle model. When diagnosing the effect of the IS92a emissions scenario (Kattenberg et al., 1996) on GMT, they demonstrated deviations of less than 0.2 K for the 21[st] century (see their Fig.5). Recently, Calel & Stainforth (2017) highlighted the potential future role of PH99 and hence further

validation of its behavior is warranted.

In this article, we ask: 'By what calibration procedure is PH99's temperature equation able to correctly map globally averaged radiative forcing anomalies onto GMT anomalies?' Hereby 'correctly' refers to an accuracy on the order of magnitude of the standard deviation of natural variability, i.e. ~0.1 K. Quite the contrary, in the context of this article we would judge a deviation of 0.5 K as inacceptable, because a proclaimed goal of the 2015 Paris Agreement (UNFCCC, 2016) is '…holding the increase

in the global average temperature to well below 2°C above pre-industrial levels and pursuing efforts to limit the temperature increase to 1.5°C above pre-industrial levels…' Hence in the policy domain, a difference in terms of 0.5 K does matter. In fact we believe that further validation is both necessary and possible at a higher level of consistency. Firstly, the respective GMT time series as checked in Kriegler and Bruckner (2004) is convexly increasing. However in the context of scenario generation in keeping with the well-below 2° target (UNFCCC, 2016), validation along GMT stabilization or even peaking scenarios is crucial, displaying a qualitatively different shape from IS92a. Secondly, in Kattenberg et al. (1996) the forcing was reconstructed by the additional assumption that non-CO2 greenhouse gas forcing approximately balances aerosol cooling.

Here we employ recently diagnosed forcings for 14 CMIP5-AOGCMs by Forster et al. (2013). As a main finding we diagnose that in the context of 2° stabilization scenarios, it would be necessary to implement a smaller ECS value in PH99 compared to the ECS value of the very AOGCM which PH99 is supposed to emulate. Hence previous work based on PH99 (see Hope (2006), Anthoff & Tol (2014) and all the MIND-based work on decision-making under ECS uncertainty (see citations above)) might require a re-interpretation. Needless to say, we are not claiming that the previously published IAM-based work mentioned above is 'worthless'. Rather, we argue that the parameters and probability density distributions need to be interpreted as transformed ones, essentially because a response has been sampled which is higher than that of the corresponding AOGCM. Hence we propose calibrating PH99 by mapping AOGCMs' ECS and TCR to respective effective values, which are suitable for a centennial time horizon, before using them in PH99.

In this way, PH99 could complement the use of increasingly complex climate modules, ranging from DICE's 2-box model (Nordhaus, 2013) to the complex upwelling-diffusion climate module used in MAGICC (Meinshausen et al. (2011a)). The potential benefits of doing so are two-fold: firstly, the most parsimonious SCM, PH99, ensures maximum comprehensibility. Secondly, in the context of numerically solving decision-making under climate response uncertainty (Kunreuther et al., 2014), having to simultaneously deal with dozens, hundreds or even thousands of alternate climate 'states of the world' (the economist's term for the uncertain system property) poses a significant challenge for numerical solvers and memory. In this regard, PH99 appears particularly attractive. Keeping the state space as slim as possible proves particularly relevant for decision-making under uncertainty with endogenous learning. For that reason, Traeger (2014) utilizes a 1-box rather than a 2-box model, however with an exogenously given time series somewhat mimicking the existence of a deep ocean layer.

Finally, our article represents a warning: if PH99 is to be used in the future, it should be done in a re-scaled manner, adjusted to the time horizon under investigation.

This article is organized as follows. Section 2 introduces the data-based part of our analysis. We call for a 3-step procedure, including: (i) a conventional, though not naïve calibration of PH99 with regard to climate sensitivity and transient climate response (i.e. the GMT change in response to a 1%/yr. increase in the CO2 concentration until doubling compared to the pre-industrial value); (ii) an AOGCM-specific calibration; and (iii) the validation of (ii). In Sect. 3 we first demonstrate that (i) would lead to emulation errors of up to 0.5 K for scenarios approximately compatible with the 2° target. We then show that this emulation error can generically be reduced to 0.1 K when choosing AOGCM-specific calibrations of PH99. This calibration is subsequently validated by independent scenarios. Note that, in Sect. 3, we focus on only RCP2.6 scenario for

calibration and use RCP4.5 and RCP8.5 for validation and leave further analyses, which show that PH99 can be generally calibrated to and validated by a variety of scenarios, for the sake of brevity, to Appendix 2. In Sect. 4 we present a scheme of how to calibrate PH99 for a given ECS, thereby avoiding AOGCM-specific calibrations. This results in a larger emulation error than achieved in Sect. 3, but one that would nevertheless suffice for most applications. In Section 5 we explain the

observed discrepancy between PH99 and AOGCMs as reported for step one of Sect. 2 by pursuing a semi-analytical, physically-based approach. In Sect. 6 we discuss the implications of our findings for the integrated assessment community, while Sect. 7 presents our conclusions and outlines further research needs.

Before we proceed, a brief note on the role of AOGCM data in our article is in order. We compare PH99 to AOGCM data because we utilize AOGCMs here as the entities closest to 'reality' available on the 'model market'. We do not, however,

claim that IAM modelers were using them or should be using them. AOGCM data is used to demonstrate how ECS and TCR data can skew the calibration of PH99, and how it should be corrected. The same correction should in principle be used for ECS data inferred from *any* source, e.g. abstract distributions such as those presented in Bindoff et al. (2013). Mirroring PH99 in AOGCM data, however, is currently the most direct way to infer the quality of a (not) re-calibrated PH99.

## 2 Method

This section introduces the analytic structure of PH99, relates it to ECS and TCR, to then describe a three-step scheme for a PH99 / AOGCM intercomparison.

PH99 projects the atmospheric GMT anomaly compared to its preindustrial level. Petschel-Held et al. (1999) specified the model for a CO2-only forcing scenario and accordingly PH99 reads

$$\frac{dT}{dt} = \mu \ln(c) - \alpha T . \tag{1}$$

Here $T$ denotes the GMT anomaly, $c$ is the CO2 concentration in units of its pre-industrial level, and $\alpha$ and $\mu$ are constant tuning parameters.

From Eq. (1) we can readily read the ECS, the equilibrium temperature anomaly in response to a doubling of the CO2 concentration compared to its pre-industrial value:

$$ECS = \frac{\mu}{\alpha} \ln(2) \tag{2}$$

also in line with Petschel-Held et al. (1999) and Kriegler and Bruckner (2004). In Appendix 1 we briefly derive the TCR (GMT) from a stylized experiment after the CO2 concentration has been exponentially increased with the rate $\gamma$ (of 1%/yr.) until the concentration has doubled for this model:

$$TCR = \frac{\mu\gamma}{\alpha^2}\left(-1 + 2^{-\frac{\alpha}{\gamma}} + \frac{\alpha}{\gamma}\ln(2)\right) = \frac{\gamma ECS}{\alpha \ln(2)}\left(-1 + 2^{-\frac{\alpha}{\gamma}} + \frac{\alpha}{\gamma}\ln(2)\right) \tag{3}$$

In the following we propose a 3-step validation approach to clarify PH99's range of applicability.

## 2.1 Step One

We first check whether simply calibrating PH99 from AOGCM-specific ECS and TCR data would deliver good emulations (i.e. accurate to within 0.1K) for 2°-target-compatible scenarios. After a technical derivation, we summarize this method of mapping AOGCMs' ECS and TCR onto PH99's two parameters.

Some difficulty arises due to the fact that AOGCMs have not been run for 2°-target-compatible scenarios for CO2-only forcing, but solely for a plethora of simultaneous forcings that would add up to a total forcing. Hence we generalize Eq. (1) to its total-forcing counterpart (see Eqs. (4)-(7)) to be driven by total forcing time series as reconstructed in Forster et al. (2013). Accordingly, we utilize scenarios generated by 14 AOGCMs (see Table 1) from CMIP5. From Forster et al. (2013), we also take the ECS and TCR for these 14 models to derive model-specific $\alpha$ and $\mu$, utilizing Eq. (2) and Eq. (3).

In order to generalize Eq. (1), we recall its derivation from an energy balance approach, as summarized in Kriegler and Bruckner (2004), allowing for a physical interpretation of the model. We start by introducing the general energy balance equation, expressing the change in oceanic heat content as the difference of ingoing ($F$) and outgoing ($\lambda T$) radiative flux while $h$ denotes the constant effective oceanic heat capacity (see also Geoffroy et al., 2013, Eqs. 1-4).

$$h\frac{\mathrm{d}T}{\mathrm{d}t} = F(t) - \lambda T(t) \tag{4}$$

$F$ also represents the total radiative forcing as applied in Forster et al. (2013). However the equation could still not be integrated as $h$ and $\lambda$ are yet to be determined. In order to solve the posed problem (CO2-only versus total forcing) we note that $h$ and $\lambda$ represent universal parameters of PH99 in the sense that their numerical values would not depend on the mix of substances (i.e. CO2, other greenhouse gases, aerosols, etc.) causing the total radiative forcing. Therefore, $h$ and $\lambda$ can be determined by

considering the CO2-only case and, hence, by tracing them back to the already determined $\alpha$ and $\mu$. For the CO2-only case, Eq. (4) reads

$$h\frac{\mathrm{d}T}{\mathrm{d}t} = -\lambda T(t) + Q_2 \frac{\ln c(t)}{\ln 2} \tag{5}$$

$Q_2$ denotes the additional forcing from the doubling of the CO2 concentration compared to its pre-industrial value and is listed for all of the AOGCMs (see Forster et al., 2013, Table 1).

If we then divide by $h$, we obtain:

$$\frac{\mathrm{d}T}{\mathrm{d}t} = -\frac{\lambda}{h}T(t) + \frac{Q_2}{h}\frac{\ln c(t)}{\ln 2} \tag{6}$$

A comparison with Eq. 1 readily reveals

$$\alpha = \frac{\lambda}{h} \quad \text{and} \quad \mu = \frac{Q_2}{h \ln 2} . \tag{7}$$

These equations would allow for determining $h = Q_2 / (\mu \ln 2)$ and $\lambda = \alpha h$. Utilizing these equations and Eq. (4), we generate

PH99's temperature response to the total radiative forcing as specified in Forster et al. (2013).

The derivation displayed so far can be summarized in terms of the following recipe to generate PH99's parameters on the basis of AOGCMs' ECS and TCR:

1. Set PH99's ECS and TCR equal to the selected AOGCM's ECS and TCR.
2. Numerically invert Eq. (3), right-hand side expression, to find $\alpha$ (no analytic expression possible).
3. Invert Eq. (2) to find $\mu$.
4. Derive $h$ and $\lambda$ from Eqs. (7), then utilize Eq. (4), divided by $h$.

Finally, to avoid differences occurring over the historical period (pre-2006 for the RCPs), we need to initialize PH99 with each AOGCM's 2006 temperature anomaly with respect to the pre-industrial value. To do this, for each AOGCM we calculate the mean temperature over the period 1881-1910 and set this as the pre-industrial value. We then calculate the mean temperature over the period 1991-2020 and use this as an indicator for the 2006 temperature level. The difference between these two values is fixed as the initial temperature anomaly for PH99.

Each temperature trajectory should be compared to the temperature data from the corresponding AOGCM. As for GMT-target-constrained economic optimizations (Clarke et al., 2014; Edenhofer et al., 2005), the maximum GMT (rather than the whole time series) is of special importance. Hence we use the difference between the respective 2071-2100 GMT time averages of PH99 and the AOGCM as an error metric. If the deviations are tolerable (accurate to within 0.1K), the climate module is validated; if they are intolerable, we proceed with steps two and three.

## 2.2 Step Two

For each AOGCM, $\alpha$ and $\mu$ are tuned such that the difference between PH99 and the AOGCM GMT anomaly for the RCP2.6 scenario in the period 2006-2100 is minimized using a least squares approach. For further diagnostics we then determine the new 'effective' ECS and TCR from Eq. (2) and Eq. (3). As in step one, the deviations in 2071-2100 means of GMT between PH99 and the respective AOGCM are determined as an accuracy check.

## 2.3 Step Three

Lastly, we validate the PH99 model versions generated in step two. For this purpose, independent temperature and forcing paths must be run as a nontrivial test to check whether the trained climate module can accurately project other temperature data trajectories. To do so, the values for $\alpha$ and $\mu$ determined in step two are implemented in PH99, the latter then being driven by the total climate forcing of the RCP4.5 and RCP8.5 scenarios. Similar to steps one and two, the deviations in 2071-2100 means of GMT between PH99 and the respective AOGCM are determined as an accuracy check.

One might be interested in seeing if the calibrated module is capable of mimicking other scenarios such as RCP6.0 or what if PH99 was calibrated to RCP4.5 or others. Stating that, in general, the procedure outlined above brings about similar results, for the sake of brevity of the main text, we present the respective results in Appendix 2.

# 3 Results

Table 1 shows the calculated $\alpha$ and $\mu$ together with the feedback response time $1/\alpha$ in step one. For all of the indicators we also compute the mean values and standard deviations of the samples. The mean value of the ECS for GCM data is 3.35 K, with a minimum and maximum of 2.11 K and 4.67 K, respectively. The mean value of the time scales is roughly 35 years.

Figure 1 represents the projected PH99 temperature evolution for the scenario RCP2.6 of each GCM in 2006-2100, using the data from Table 1 and RCP2.6s' forcings. PH99 clearly overestimates the temperature anomaly for all GCMs, especially over the last 30 years. The absolute values of the deviations of mean temperature over the last 30 years (hereafter: MTD) from the AOGCM data are shown in Figure 2. The MTD ranges from 0.22 K for MRI-CGCM3 to approximately 0.79 K for HadGEM2-ES. On average, the deviations are ca. 0.45 K. This is clearly a large error, both in units of annual GMT standard deviation as

well as the climate policy dimension. A proclaimed goal of the 2015 Paris Agreement (UNFCCC, 2016) is '…holding the increase in the global average temperature to well below 2°C above pre-industrial levels and pursuing efforts to limit the temperature increase to 1.5°C above pre-industrial levels…' Hence a difference in 0.5 K does matter. Accordingly, we must proceed with step two.

In step two, for each of the GCMs, we tune $\alpha$ and $\mu$ such that the GMT deviations for the whole period 2006-2100 are

minimized in a least squares manner as represented in Figure 3 and Figure 4. From the thereby adjusted $\alpha$ and $\mu$ we derive the ECS and TCR, which are presented in Table 2. MTDs for the various AOGCMs are shown in Figure 2.

The results tell us three main things. Firstly, the average of the absolute values of deviations is significantly reduced when $\alpha$ and $\mu$ are tuned. Indeed, the MTD average drops to below 0.02 K. Secondly, while the average ECS decreases by 0.9 K (from 3.35 K to 2.46 K), the average TCR increases by 0.14 K (from 1.90 K to 2.04 K). Thirdly, the mean value of feedback response

times decreases significantly, from roughly 35 years to less than 12 years.

For validation we move on to step three. We utilize the RCP4.5 temperature and forcing data as provided by Forster et al. (2013). In Figure 3 and Figure 4 the respective GMT trajectories for any AOGCM are contrasted with the PH99-generated ones, where $\alpha$ and $\mu$ are fixed to their values as determined in step two. The MTDs are shown in Figure 2. The results confirm that the climate module is sufficiently well trained in the second step that it can suitably mimic the actual temperatures (accurate

to within 0.1K) for RCP4.5 and RCP8.5. As shown, the average MTD is approximately 0.05 K for RCP4.5 and about 0.14K for RCP8.5. For RCP4.5, the deviations for three of the GCMs, namely CCSM4, CNRM-CM5 and NorESM1-M, are even better than those diagnosed for RCP2.6 in step two. See Appendix 2 for further analyses.

# 4 A mapping of ECS onto their PH99-specific counterparts $\alpha$ and $\mu$

Finally, we attempt to abstract from fitting PH99 to individual AOGCMs and provide an approximate way to calibrate PH99

within the cloud of AOGCMs simply by knowing the ECS. Then PH99 could be utilized for any ECS in analyses where the ECS is uncertain.

## 4.1 An existing mapping for PH99

Before diving into our suggestions, we examine one of the existing options (a reader solely interested in our improved method of utilizing PH99 can move straight onto Subsection 4.2). We inspect the curve suggested by Lorenz et al. (2012), which correlates $\alpha$ and $\mu$ to ECS. Using a sample from Frame et al. (2005) and assuming a strict relationship between $1/\mu$ and ECS,

Lorenz et al. (2012) suggest the following approximation:

$$\frac{1}{\mu} \approx \frac{1}{\overline{\mu}} - 10 \exp(-0.5 \, ECS) \tag{8}$$

where $\overline{\mu}$ is the mean value of $\mu$ in the sample (see Fig.7 in Lorenz et al., 2012, all quantities measured in the units utilized in Kriegler & Bruckner, 2004). Knowing $\mu$, Eq. (2) is used to determine $\alpha$. In turn, Eq. (2) and Eq. (8) have been repeatedly used in studies employing MIND and concerning uncertainties and ECS (Neubersch et al., 2014; Roshan et al., 2018; Roth et al.,

10    2015).

We employ Eq. (2) and Eq. (8) for all ECSs from Table 1 and show the MTDs for the RCP2.6 scenario in Figure 5. Notice that TCR can readily be calculated using Eq. 3. Clearly, on average, employing Lorenz's curve does not result in a better situation than step one. However, this might not necessarily be a case of comparing like with like. At the time of Frame et al. (2005), the two-dimensional uncertainty information was obtained by reconstructing the 20[th] century's warming signal from

fingerprinting by means of a single AOGCM and then using this observational data as a constraint. It is well known that observational constraints may lead to different distributions than ensembles of AOGCMs do (Andrews & Allen, 2008). Nevertheless we include this piece of information here for the sake of completeness.

## 4.2 A multiple AOGCM-based mapping for PH99

Given the inferred estimates in Table 2, one can directly relate $\alpha$ and $\mu$ to the ECS. To do so, we generate polynomial fits (of orders 2 and 3) of $\alpha$ and $\mu$ against all AOGCMs' ECSs. Predicting a two-dimensional manifold from ECS alone implicitly exploits the fact that AOGCMs' TCRs can be predicted well using ECSs (see e.g. Meinshausen et al., 2009) in a statistical sense. Another option would be to derive $\alpha$ and $\mu$ analytically (like in the first step) when the inferred ECS and TCR are correlated to the ECS and TCR of AOGCMs.

Figure 6 relates $\alpha$ and $\mu$ (from Table 2) to the ECS (from Table 1), using linear, quadratic and cubic polynomial approximations. For the case of a linear approximation, we put the model GISS_E2_R out as an outlier. Figure 5 indicates that on average all approximations mimic the actual temperature paths better than a non-fitted one. The cubic estimation projects significantly smaller deviations compared to the quadratic approximation and slightly smaller deviations compared to the linear approximation. The maximum MTD in the cubic approximation is 0.3 K for IPSL-CM5A-LR, which is roughly a third of the

maximum in the quadratic approximation that is revealed for CSIRO-Mk3-6-0.

We also consider alternative ways to map ECS and TCR from the 14 utilized AOGCMs onto PH99-intrinsic properties, going beyond the scheme displayed in Figure 6. As one option, shown in Figure 7, we linearly regress the ECS and TCR values inferred from step two against their original AOGCM counterparts respectively and obtain

$$ECS_{PH99} \approx a\, ECS_{AOGCM} + b \tag{9}$$

with $a$= 0.5846, $b$= 0.5095 K, and $R^2$ =0.8158, as long as $ECS_{PH99} < ECS_{AOGCM}$

and

$$TCR_{PH99} \approx c\, TCR_{AOGCM} + d \tag{10}$$

with $c$= 0.9763, $d$= 0.1829 K, and $R^2$ =0.667.

The other option consists in using Eq. (9) along with a linearly regressed $TCR_{PH99}$ over $ECS_{AOGCM}$, that is

$$TCR_{PH99} \approx m\, ECS_{AOGCM} + n \tag{11}$$

with $m$= 0.4582, $n$= 0.5044 K, and $R^2$ =0.7876.

The respective MTDs are shown in Figure 5. Although both approximations mimic the actual temperature paths better than a non-fitted one, regressing both the inferred effective ECS and TCR solely against AOGCMs' ECS (hereafter: ETE) clearly offers the best overall approximation.

Using the ETE has four major advantages over all other options dealt with here, especially for the IAM community. Firstly, its approximation is better than all options but the cubic fit. Secondly the ETE still has an advantage over the cubic fit because one can easily use a broader range of climate sensitivities, for example, from 1 K to 9 K, which may not be accurately determined by the cubic fit. Even though the cubic fit may yield a better approximation, in our analysis it is only better by 0.03 K at the expense of a non-intuitive shape that might result in even worse deviations for out of sample data. Thirdly, prior knowledge regarding the TCR is no longer a decisive factor. Note that prior knowledge regarding the TCR can make approximations better. However, as we tested, for example, in the case of linearly regressing both the inferred effective ECS and TCR against both AOGCMs' ECS and TCR, the R-squares for Eq. (9) and Eq. (11) only improve by 6% and 7% respectively, and the MTD is no better than the ETE. Finally, in the case of ETE, we do not need to re-evaluate our sample and possibly drop any model as an outlier. Given the explorations already done and their performance, we leave explorations beyond the linear approximation for future research.

## 5 An analytic interpretation of the AOGCM-PH99 intercomparison

In the following, we explain why PH99 systematically overestimates maximum GMT for peaking scenarios when fitted for exponentially growing scenarios. As an AOGCM is analytically not accessible, we investigate an intermediate step of model replacement by moving from a 1-box to a 2-box SCM (as utilized in DICE (Nordhaus, 2013)). In fact we qualitatively trace back the effects reported so far to the information loss incurred by replacing a 2-box SCM with a 1-box SCM like PH99. We then also investigate the quality of alternative fitting schemes based on our semi-analytic analysis, which complements our previously mentioned AOGCM-based validation.

Following Geoffroy et al. (2013) we introduce a 2-box SCM as a more universal emulator of AOGCMs' mapping from radiative forcing onto temperature.

$$C\,\frac{dT_{2B}}{dt} = F - \lambda_{2B} T_{2B} - \delta(T_{2B} - T_0) \tag{12}$$

$$C_0 \frac{dT_0}{dt} = \delta(T_{2B} - T_0)$$

$T_{2B}$ denotes the 2-box analogue of the 1-box temperature $T$ in Eq. (1). The upper and the lower equation represent the upper and the lower ocean, respectively.

In order to contrast PH99 with this 2-box model, we search for analytic approximations of generic shapes of the forcing $F(t)$ and examine the long-term projections under various RCPs as depicted in Meinshausen et al. (2011b) – an excerpt is included in Figure 8 for the reader's convenience. Particularly in view of the peaking, mitigation-oriented lowest forcing scenario, we approximate forcing paths in three phases: zero forcing, linear increase, and linear decrease, under a continuity assumption.

$$F(t) = \begin{cases} 0 & \text{for } t < 0 \\ k_1 t & \text{for } 0 \le t \le t_1 \\ k_2(t - t_1) + k_1 t_1 & \text{for } t > t_1 \end{cases} \tag{13}$$

We approximately identify $t_1$ with the year 2035 and $t=0$ with 100 years earlier, i.e. we assume a ramp-up time $t_1$ for the forcing of roughly 100 years. Furthermore, $k_2 < 0$ and $|k_2 / k_1| =: \varepsilon \ll 1$. From Figure 8 we approximate a generic value of $\varepsilon = 0.2$. For $0 \le t \le t_1$ we draw on Geoffroy et al. (2013 – see their Eq. (14))

$$T_{2B}(0 \le t \le t_1) = \frac{k_1}{\lambda_{2B}}\left(t - \tau_f a_f\left(1 - e^{-\frac{t}{\tau_f}}\right) - \tau_s a_s\left(1 - e^{-\frac{t}{\tau_s}}\right)\right) \tag{14}$$

This represents two linear modes of amplitudes $a_f$ and $a_s$ (with sum equal to 1), delayed by the characteristic time scales of a fast and a slow mode, $\tau_f$ and $\tau_s$, respectively, and continuously matched to the initial condition '0' by an exponential. In Geoffroy et al. (2013) the 2-box model is fitted to 16 AOGCMs. After having reviewed their results for our order-of-magnitude estimates of PH99's accuracy, we can make the following two simplifying assumptions: (i) both amplitudes $a_f$ and $a_s$ approximately equal 1/2 (see their Fig. 3a – amplitudes range from 0.35 to 0.65), (ii) $\tau_f \approx 0$ (values range from 1 yr. to 5.5 yrs., see their Table 4; for centennial effects, this mode would nearly match the equilibrium response). Furthermore we can see that $\tau_s$ ranges from 100 yrs. to 300 yrs. for 15 out of 16 AOGCMs. Hence the 2-box model is characterized by a marked time-scale separation between the two linear modes. With the aid of these two approximations, the last equation can be simplified to

$$T_{2B}(0 \le t \le t_1) \approx \frac{k_1}{\lambda_{2B}}\left(t - \frac{\tau}{2}\left(1 - e^{-\frac{t}{\tau}}\right)\right) \text{ with } \tau := \tau_s. \tag{15}$$

We then extend the analytic range of that formula, given the two approximations above, for $t > t_1$ (for a derivation, see Appendix 3):

$$T_{2B}(t > t_1) \approx \frac{k_1}{\lambda_{2B}}\left(-\varepsilon t + (1 + \varepsilon)t_1 + \frac{\tau}{2}\left(\varepsilon + e^{-\frac{t}{\tau}} - (1 + \varepsilon)e^{-\frac{(t-t_1)}{\tau}}\right)\right) \tag{16}$$

The analogous expressions for the 1-box model read

$$T(0 \le t \le t_1) = \frac{k_1}{\lambda}\left(t - \theta\left(1 - e^{-\frac{t}{\theta}}\right)\right), \theta := \frac{1}{\alpha}, \quad \lambda \text{ from Eqs. (7)}, \tag{17}$$

and

$$T(t > t_1) = \frac{k_1}{\lambda}\left(-\varepsilon(t - \theta)+(1 + \varepsilon)t_1 + \theta\left(e^{-\frac{t}{\theta}} - (1 + \varepsilon)e^{-\frac{(t-t_1)}{\theta}}\right)\right). \tag{18}$$

## 5.1 Explaining the PH99-AOGCM discrepancy for equal ECS and TCR values

We are now prepared to mimic Step One in Section 2: we calibrate the 1-box model such that it is characterized by the same ECS and TCR as the 2-box model. As $\lambda = Q_2/ECS_{2B}$ , equal ECS values for both models deliver $\lambda = \lambda_{2B}$.

Determining the second degree of freedom of PH99 (e.g. as expressed by $\theta$ ) from some transient property proves more intricate. We request

$$T(t_{TCR}) = T_{2B}(t_{TCR}) \tag{19}$$

whereby we introduce $t_{TCR}$ as the moment in time when $T$ needs to be evaluated in order to determine TCR. In Appendix 1 we note, by definition, that $t_{TCR} = (\ln 2)/\gamma \approx 70\text{yrs}$ for a growth rate $\gamma = 1\%/\text{yr}$ of the carbon dioxide concentration, hence $0 < t_{TCR} < t_1$.

Therefore, when exploiting Eq. 19, Eqs. 15 and 17 (rather than 16 and 18) apply and result in the expression

$$h\left(\frac{\theta}{t_{TCR}}\right) = \frac{1}{2}h\left(\frac{\tau}{t_{TCR}}\right) \tag{20}$$

with $h$ denoting the auxiliary function (see Figure 9)

$$h(x) := \left(1 - e^{-\frac{1}{x}}\right)x, \tag{21}$$

where

$$\lim_{x \to 0} h(x) = 0, \lim_{x \to \infty} h(x) = 1, \quad h(x) \approx x \text{ for } x \ll 1. \tag{22}$$

From this, we can already get a first impression of the scale of $\theta$, prior to numerical inversion: as $\tau$ is generically markedly larger than $t_{TCR}$, the right-hand side of the defining equation above approximates ½. Further, if we boldly assume a slight time-scale separation between $\theta$ and $t_{TCR}$, the former being smaller than the latter, then the linear approximation of $h$ would apply and $\theta \approx t_{TCR}/2 \approx 35$ yrs. For a centered value of $\tau \approx 250$ yrs, this approximation is confirmed in a direct numerical treatment of Eq. (20).

Hence from the twin time-scale separation of 'the 1-box model mode,' 'defining time scale for TCR,' and the 'slow mode of the 2-box model' we have explained why TCR-oriented fitting exercises of the 1-box model would generically result in time scales of roughly 30 to 40 years (see e.g. Anthoff & Tol, 2014; Kriegler & Bruckner, 2004). The factor ½ between the 1-box model's time scale and the TCR-defining time scale goes back to Geoffroy et al.'s (2013) observation that the fast and the slow mode both enter the superposition result with approximately equal weights of ½. The slow mode is then too slow to be of much relevance for TCR – a phenomenon not revealed by the 1-box model.

We are now equipped to compare the two models' temperature projections and apply the 3-phase forcing as defined above for $\varepsilon = 0.2$. $a_1/\lambda$ is chosen such that peak temperatures enter the 2° regime for illustrative purposes. We exploit the coincidence that $t_{TCR}$ just happens to approximately correspond to our starting year 2006 for PH99 (because 2035-100+70=2005). Hence the

formulas for the 1-box model do not need to be adapted for an explicit initial condition for this purpose. Figure 10 shows that by construction, both temperature responses match at $t_{\text{TCR}} \approx 70$ yrs., although the 1-box model's maximum exceeds the maximum by 0.5 K. This phenomenon can be explained as follows. As the 1-box model responds with a finite time scale, its derivative must be continuous in response to a continuous forcing. Hence the leading term is quadratic when the forcing starts.

In contrast, the 2-box model contains a virtually degenerate time scale (the fast one); hence its leading term is linear. If the two curves are to nevertheless match at $t_{\text{TCR}}$, the 1-box model's derivative at $t_{\text{TCR}}$ must transcend the 2-box model's derivative. This, together with the right-bending kink in the 2-box model's response at $t_1$, leads to a larger maximum in the 1-box model. In summary, on time-scales much smaller than the slow mode, the slow mode, compared to the fast mode, cannot develop yet; hence the fast mode will dominate the slow mode. As such, fitting a 1-modal model in a convex regime is likely to yield poor

predictions of a temperature maximum for mitigation-based forcings.

This explains the discrepancies found in our PH99-AOGCM comparison when directly transferring AOGCMs' ECS and TCR onto PH99. Figure 10 further suggests that if PH99 were used to predict correct maxima and emulate AOGCMs in this time regime, it would need to be used with a markedly smaller time scale. However, a simple reduction in time scale would lead to a new inter-model discrepancy before the kink; hence the overall amplitude of PH99's response would need to be reduced as

well. The latter scales with the ECS; hence the ECS must be reduced by a certain factor towards a new 'effective ECS,' which could also be called a 'transient climate sensitivity.'

## 5.2 Testing the validity of a recalibrated PH99 for a 2-box model

In Sect. 5.1 we derived an analytic explanation for why a naïve transfer of an AOGCM's ECS and TCR to PH99 results in a maximum GMT which is too large when driven by a mitigation forcing scenario. However we show in Sections 3 and 4 that

PH99 in fact is a good emulator of an AOGCM within 0.1K if it either were directly fitted to that AOGCM or if the AOGCM's ECS and TCR were transformed into effective quantities for PH99. Hereby 'good emulator' expresses the fact that the same parameter set can be utilized for any RCP (2.6, 4.5, 6.0, 8.5). From a practical point of view, we could stop our analysis here and suggest that this type of validation might be sufficient to generate trust in PH99 as an emulator for any forcing scenario. However for further validation, in this Subsection we would like to exploit the fact that for a 2-box / 1-box intercomparison

we can validate PH99 for an order-of magnitude larger set of forcing scenarios (again presupposing that a 2-box model would emulate an AOGCM qualitatively better than a 1-box model). We systematically test the previously suggested adjustment formulas Eqs. (9) to (11) for a range of $t_1$ and $\varepsilon$ values, hence varying mitigation scenarios, given alternative ECS and slow mode's time scale $\tau$ for the 2-box model. We find numerically that $\theta$ is on the order of 10 years, and the ECS needs to be reduced by 1/4 to 1/3. We test for the centred ECS values of 3 K and 4 K and a slow mode's time scale, which generically

ranges from 100 yrs. to 300 yrs (see Geoffroy et al., 2013).

In principle, for any forcing scenario characterized by varying $t_1$ and $\varepsilon$, we would need to compare GMT as calculated by Eqs. (17)-(18) vs. Eqs. (15)-(16). However any of these Eqs. derive GMT for the boundary condition of zero temperature at $t$=0.

On the contrary, our validation scheme as utilized in Sections 3-4 would fix PH99 to the AOGCM at the year 2006. The latter point in time we denote by $t_0$ ($\approx t_{TCR}$). Having transformed ECS and TCR according to Eqs. (9)-(11) we cannot expect any longer $T(t_0)=T_{2B}(t_0)$. Therefore we have to force the solution of PH99 to the solution of the 2-box model at $t_0$ and call the thereby initialized solution of PH99 '$T_{init}$':

$T_{init}(t_0)=T_{2B}(t_0)$.               (23)

We generate $T_{init}(t)$ from $T(t)$ (see Eqs. 17 and 18) by adding a suitably scaled solution of the homogenous counterpart of Eq. 4:

$$T_{init}(t \geq t_0) = T(t) + \left(T_{2B}(t_0) - T(t_0)\right)e^{-\frac{(t-t_0)}{\theta}}. \qquad (24)$$

Figure 11 shows the relative deviations of the GMT maxima of the 1-box and the 2-box model for the extrapolation scheme ETE (Eqs. (9) and (11)). In a certain regime, the extrapolation delivers sufficiently accurate results, however, not everywhere. When utilizing the mapping scheme represented by Eqs. (9) and (10), the results look similar. The overall impression is that the mapping removes the bias. However, it does not deliver a universal correction as found for the direct intercomparison between PH99 and AOGCMs. Hence we cannot exclude the possibility that AOGCMs are easier to emulate as they contain

many more time scales than the 2-box model and their effects might in part cancel.

While we observe a qualitative gain, Figure 11 reveals there is still room for improvement. Accordingly, we further transform the ECS to request perfect matching for $t_1=100$ yrs, $\varepsilon=0.2$; the results can be seen in Figure 12. The fit is much further improved such that a major fraction of ($t_1$, $\varepsilon$) values would lead to a relative error of $<5\%$, and another large fraction to a relative error of $<10\%$. As the standard deviation of annual GMT is between 0.1°C and 0.2°C and a typical application might be a cost-

effectiveness analysis of the 2°C target, such errors might still seem tolerable. However we observe structural problems for very small values of $\varepsilon$, the latter implying very late assumption of a maximum. In this case, the slow mode becomes more relevant, and hence the quality of the calibration deteriorates. The calibration is valid for a time horizon on the order of $t_1$ to 2 $t_1$, i.e. on the order of the time to peak forcing.

**6 Discussion**

The previous section offers a key mechanism to explain why, for given ECS and TCR, GMT responses generated by PH99 in response to peak-and-decline forcing scenarios are biased towards higher temperatures. How does this relate to the observation that PH99 tend to underestimate the effect of greenhouse gas emissions (van Vuuren et al. (2011a)) as mentioned in our introduction? In fact, van Vuuren et al. (2011a) describe a different forcing experiment: a step-function in the course of time (see their Fig. 3). Here FUND, based on PH99, displays a GMT lower than that of MAGICC-4 by more than 0.8 K at certain

times during the most transient phase, although both models share the same ECS. This can be explained by the lack of time scales faster than 35 yrs (the latter characterizing PH99 in standard calibrations) within PH99. Whether PH99 over- or

underestimates GMT is hence a strong function of the functional shape of forcing. Our article highlights the effects of a naively calibrated PH99 on mitigation scenarios.

However, one should not forget about potential additional mechanisms. Firstly, the statistical errors in determining AOGCMs' ECS, TCR and $Q_2$ may lead, mediated through the nonlinear mapping on PH99's parameters, to an overall bias in PH99's

GMT. Furthermore, diagnosing the total radiative forcing active in an AOGCM is a complex undertaking (see e.g. Meinshausen et al., 2011a, for a discussion). A bias to the high end here would also result in inaccurately large GMT responses by PH99. However, in the context of this article, we contend that the information loss when moving from a 2-box to a 1-box model is the key source of the observed discrepancy – last but not least, we find Figure 10 compelling in this regard. Complying with the latter interpretation raises a key question: Can PH99 be seen as a 'physical model' and if so, what are the implications for

users? It is readily apparent that a 1-box model cannot mimic a 2-box model, characterized by a marked time-scale separation for all forcings at all times. However it is equally clear that the simplest temperature equation is in fact the one that treats the ocean as a single box. It would still explain warming with forcing in a quasi-linear manner, though with some delay. If we are willing to accept that the calibration of PH99 is time-horizon-specific, then PH99 still holds some semi-physical meaning. If, however, this is seen as unacceptable, then we would have to recognize that PH99 is more an efficient emulator than a physical

model. In this context we would like to recall that virtually every model has a limited range of validity – and as such, PH99 is no different from most other models.

When investigating the 1-box / 2-box-models' differences, our research also suggests that within the class of peak-and-decline scenarios PH99 provides a good emulation (accurate to within 0.2 K for a generic AOGCM setting such as ECS=4 K, a peaking of forcing between 2020 and 2100, and a ratio of slopes of pre- and post-peaking forcing of 0.1 to 0.4). For the AOGCM/PH99

intercomparison, PH99 performs even better: for RCP2.6, 4.5, 6.0 (~0.1K) and, to a lesser extent, 8.5.

What are the ramifications of our findings for previous publications based on PH99? Those authors who claimed to have worked with PH99 in conjunction with ECS=3°C have effectively worked with a more complex model in conjunction with ECS≈4°C for the centennial time horizon. Much of the work done based on MIND in conjunction with PH99 and the log-normal distribution for ECS by Wigley & Raper (2001), has essentially been based on a log-normal distribution shifted to

larger ECS values. The 5%, 50% and 95% quantiles of the log-normal distribution by Wigley & Raper (2001) are 1.2 K, 2.6 K and 5.8 K, respectively. When interpreting these values as PH99 values, as they have in fact been utilized in PH99 for the MIND model since Lorenz et al. (2012), in the sense of a rough estimate one could ask what were the corresponding effective ECS values of a more complex model according to our Figure 7. The respective values are 1.2 K, 3.6 K and 9.0 K. From Figure 13, which reflects IPCC AR5's synopsis of current knowledge regarding ECS (Bindoff et al., 2013), we can see that these are

still in line with the range spanned by instrumental studies. Hence the results obtained by PH99 in conjunction with the distribution by Wigley & Raper (2001) are not erroneous, but simply need to be re-interpreted as rather high-end representatives within the collection of ranges as seen in IPCC AR5.

For future applications we can conclude that PH99 must be applied and interpreted with greater care – utilizing transformed values for ECS and TCR – than in the past, if it is not to be replaced by at least a 2-box model as suggested by Geoffroy et al.

(2013) and implemented in DICE (Nordhaus, 2013). 1-box models like PH99 can be crucial for modelling decision-making under uncertainty and anticipated future learning. As an illustration, execution of the MIND model currently demands between hours and days for 20 different values of climate sensitivity in conjunction with one learning step (E. Roshan, pers. comm.). The execution time needed will grow exponentially with the number of learning steps and at least linearly with the number of state variables influenced by uncertainty. For endogenous learning in a recursive design, computation time scales factorially with the numerical resolution per state variable. The change from a 1-box to a 2-box model might hence imply an order of magnitude larger execution time (C. Traeger, pers. comm. in conjunction with Traeger (2014)). So a 1-box model will remain an attractive alternative in numerical applications addressing decision-making under anticipated future learning. Users who would like to go that road might, however, also consider the augmented 1-box model by Traeger (2014) as an alternative to PH99, employing an additional exogenous forcing of that single box to somewhat emulate two boxes.

## 7 Summary and Conclusion

We utilize recent data on total radiative forcing (Forster et al., 2013) from 14 state-of-the-art CMIP5 Atmosphere Ocean General Circulation Models (AOGCMs) in order to test the validity of the 1-box climate module by Petschel-Held (1999, 'PH99') for scenarios approximately compatible with the 2° target. PH99 is currently utilized within the integrated assessment models FUND, MIND and PAGE.

We find that when prescribing the equilibrium climate sensitivity (ECS) and transient climate response (TCR) of these AOGCMs to the emulator PH99, global mean temperature (GMT) is generically projected 0.5 K higher. In contrast, by directly fitting PH99 to the RCP2.6 time series and validating with the RCP4.5 and RCP6.0 series, we find that PH99 can emulate AOGCMs to a degree of accuracy better than 0.1 K. Even for RCP8.5 the error is on the same order of magnitude, although somewhat larger (up to 0.2 K).

We numerically demonstrate that PH99 can be used to excellently emulate AOGCMs (accurate to within 0.1 K on average) within centennial-scale integrated assessment of the 2° target, provided its ECS and TCR are re-interpreted as effective values and mapped from original ECS and TCR values. We suggest such a mapping.

Furthermore we explain the observed discrepances and the need to reduce PH99's ECS compared to the AOGCM's ECS as being due to the information loss produced by approximating a 2-box-based energy balance model with a 1-box-based model (assuming that a 2-box model mimics an AOGCM better than a 1-box model). The key point is that PH99 has a fundamentally different response shape to an AOGCM and hence ECS alone does not allow one to easily move between the two. The transformation we propose adjusts PH99's ECS, sacrificing agreement in the long-term response in order to gain agreement in the centennial response (which is useful given it is more often than not the timescale of interest).

In fact the slow mode of the 2-box model is so slow that in a climate-policy-relevant context it can unfold only up to a relatively small extent; hence for practical purposes the 2-box model's ECS cannot fully develop. Accordingly, adjusting the ECS to

lower values also proves to be compatible with reducing PH99's response time. When comparing PH99 and AOGCMs, the match is even better – a phenomenon for which the explanation is beyond the scope of this article.

Hence older work based on PH99, executed within FUND, MIND and PAGE, may need to be re-interpreted in the sense that a response had been sampled which is higher than that of the corresponding AOGCM. This effect, in turn, proves equivalent
to utilzing higher ECS values in the more complex model. Even when having dealt with distributions of ECS as for the MIND model, ECS values re-interpreted in that sense are still within the range outlined by IPCC AR5 (see Figure 13). Hereby we see this 're-interpretation' as a mere numerical fix. In terms of the underlying physics, we stress that using ECS alone to characterise climate response on a few hundred year timescale is fundamentally flawed, given that ECS takes on the order of a thousand years to emerge.

For future work, we propose the following steps: (i) By comparison with more sophisticated, multi-box climate modules it should be tested again whether the effect of a transient climate sensitivity (and TCR) alone could explain our observed PH99-AOGCM discrepancy; (ii) Future discussions with the AOGCM community should illuminate to what extent the further explanations we suggested might also apply, thereby potentially reducing the need to correct for PH99; (iii) An AOGCM- and scenario class-independent, yet centennial time-scale-specific two-dimensional mapping from ECS/TCR onto ECS/TCR and
designed for PH99 should be derived in conjunction with two-dimensional distributions inferred from observations as done in Frame et al. (2005). The IAM community could then be offered both options for emulation: the one presented here, trained by AOGCMs, and one based on observational data and mediated by more complex SCMs.

In summary, PH99 could continue to be used as the most parsimonious emulator of AOGCMs, and is especially efficient for decision-making under climate response uncertainty. However its calibration proves to be much more involved than previously
assumed. Future users should carefully consider whether they actually want to use PH99, or whether they prefer a less parsimonious solution.

**Appendix 1: An Analytic Expression of TCR in PH99**

We rearrange Eq. (1) as

$$\dot{T} = \mu \ln(c) - \alpha T \tag{A1}$$

TCR is defined as the temperature change in response to a 1%/yr. increase in CO2 concentration, starting from preindustrial conditions. Hence the concentration, expressed in units of the pre-industrial concentration, reads

$$c = \exp(\gamma t) \tag{A2}$$

with $\gamma$ denoting the above rate of change. As Eq. (A1) represents a linear ordinary differential equation with constant coefficients, and the initial temperature anomaly is to vanish, its solution reads

$$T = \mu\gamma \exp(-\alpha t) \int t \exp(\alpha t) \mathrm{d}t = \frac{\exp(-\alpha t)\,\mu\gamma(1 + \exp(\alpha t).(-1 + \alpha t))}{\alpha^2} \tag{A3}$$

Temperature should be evaluated at $t_2$ when the concentration is doubled. $t_2$ is determined by $c(t_2)=2 \Rightarrow t_2=\ln 2/\gamma$. From this and Eq. (A3) we conclude Eq. (3). (In fact we find the same result using an expression provided in Andrews & Allen, 2008, when we plug in our expression for $t_2$ into theirs, which is phrased in terms of ECS.)

**Appendix 2: Further Analysis on Calibration and Validation**

As further validation of the trained PH99 calibrated to RCP2.6, Figure 14 shows the respective GMT trajectories of AOGCMs for RCP6.0 scenario contrasted with its respective PH99-generated ones where $\alpha$ and $\mu$ are fixed to their value as determined in step two. MTDs are shown in the 3rd columns of Table 3. The missing models are due to either lack of temperature trajectories for AOGCM or lack of total forcing. Notice that 1st, 2nd, and 4th columns are exactly the numbers related to the Figure 2. The results confirm that the climate module is so well trained in the second step that it can appropriately mimic the actual temperatures (accurate to within 0.1K) for RCP6.0. As shown, the average value of MTD is about 0.06 K for RCP6.0.

Column 5 thereafter in Table 3 show MTDs in the situations when PH99 is calibrated to the other RCP scenarios and is validated as against the others.

**Appendix 3: Derivation of Eqs. (16)-(18)**

We start by rewriting Eq. (14) in a way that it is most consequently decomposed into the contributions from the two modes $i \in \{f, s\}$ (for 'slow' and 'fast' mode, respectively).

$$T_{2\mathrm{B}}(0 \le t \le t_1) = \frac{k_1}{\lambda_{2\mathrm{B}}} \sum_i a_i \left( t - \tau_i + \tau_i \mathrm{e}^{-\frac{t}{\tau_i}} \right) \tag{A4}$$

One could derive Eq. (16) from an intuitive perspective by noticing that for any of the modes $i$, its contribution to the temperature response would consist of an equilibrium response, delayed by $\tau_i$, and a summand of exponential decay which would ensure continuity with respect to the initial condition. This very principle can be followed again for the time horizon beyond $t_1$.

However, for those readers who would like to see a more formal derivation, we provide the following ansatz: For $t>t_1$, we decompose $T_{2\mathrm{B}}$ into three contributions, according to the superposition principle for linear differential equations:

1. $T_1$, induced by a forcing $k_2(t-t_1)$ with $T_1(t_1)=0$. This contribution can be treated analogously to $T_{2\mathrm{B}}(0<t<t_1)$ when noticing the replacements $k_1 \to k_2$, $t \to t-t_1$. From Eq. (A4) we infer

$$T_1(t \ge t_1) = \frac{k_2}{\lambda_{2\mathrm{B}}} \sum_i a_i \left( t - \tau_i + \tau_i \mathrm{e}^{-\frac{(t-t_1)}{\tau_i}} \right). \tag{A5}$$

2. $T_2$, induced by a constant forcing $k_1 t_1$ with $T_2(t_1)=0$. Also this problem has been solved by Geoffroy et al. (2013) in terms of their Eq. (9) which we rewrite in our notation: $T_2(t \ge t_1) = \frac{k_1 t_1}{\lambda_{2\mathrm{B}}} \sum_i a_i \left( 1 - \mathrm{e}^{-\frac{(t-t_1)}{\tau_i}} \right).$ (A6)

3. $T_3$ as the decaying initial condition at $t = t_1$. For reasons of continuity, this initial condition is identical to the terminal condition according to Eq. (A4). Hence, $T_3(t \geq t_1) = \frac{k_1}{\lambda_{2B}} \sum_i a_i \left( t_1 - \tau_i + \tau_i e^{-\frac{t_1}{\tau_i}} \right) e^{-\frac{(t-t_1)}{\tau_i}}$ . (A7)

When we add these three components, we receive

$$T_{2B}(t \geq t_1) = \frac{1}{\lambda_{2B}} \left( \sum_i a_i \left( k_1 t_1 + k_2(t - t_1 - \tau_i) + e^{-\frac{t}{\tau_i}} \left( k_1 \tau_i - e^{\frac{t_1}{\tau_i}} (k_1 - k_2) \tau_i \right) \right) \right). \quad (A8)$$

Allowing for the limit $\tau_f \to 0$ and noticing that $k_2 = -\varepsilon k_1$ we verify Eq. (16) by a summand-by-summand comparison. Allowing for $\tau_f = \tau_s = \theta$ (i.e. simulating a 1-box setting by a 2-box approach), we obtain Eq. (17) from Eq. (A4) and Eq. (18) from Eq. (A8).

**Authors' Contributions**

M.M.K. performed the statistical analysis. H.H. provided the analytic analysis. M.M.K. suggested and developed the alternative scheme. Both participated in the writing of the article.

**Competing Interests**

The authors declare that they have no conflicts of interest.

**Acknowledgments**

The authors would like to thank J. Marotzke for drawing their attention to the Forster et al. (2013) article, discussing these results on total forcing and providing the relevant data. In addition, the authors would like to thank C. Li for supporting the data handling process and making the authors aware of Geoffroy et al. (2013), which discusses negligible AOGCM drift. The authors are also grateful to E. Roshan for her help with the visualizations and providing quantiles of Wigley's & Raper's
(2001) distribution on ECS. We thank M. Fentem for proofreading the second version of our manuscript from a native speaker's perspective, as well as B. Blanz and M. Wifling for further proofreading. All remaining errors are ours. M.M.K. was supported by the Cluster of Excellence 'Integrated Climate System Analysis and Prediction' (CliSAP, DFG-EXC177). Finally, the authors would like to thank three anonymous referees for their valuable criticism and constructive suggestions.

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

**Table 1: PH99 parameters ($\alpha$ and $\mu$) and feedback response times ($1/\alpha$) utilizing data ($ECS$ and $TCR$) from AOGCMs.**

| | PH99 Parameters | | Climate Sensitivities | | Feedback Response Times |
|---|---|---|---|---|---|
| | $\alpha$ [1/yrs] | $\mu$ [K/yrs] | $ECS$ [K] | $TCR$ [K] | $1/\alpha$ [yrs] |
| bcc_csm1_1_m | 0.052 | 0.217 | 2.87 | 2.10 | 19.1 |
| bcc_csm1_1 | 0.033 | 0.132 | 2.82 | 1.70 | 30.8 |
| CanESM2 | 0.038 | 0.204 | 3.69 | 2.40 | 26.1 |
| CCSM4 | 0.035 | 0.145 | 2.89 | 1.80 | 28.7 |
| CNRM_CM5 | 0.038 | 0.177 | 3.25 | 2.10 | 26.5 |
| CSIRO_Mk3_6_0 | 0.019 | 0.111 | 4.08 | 1.80 | 53.2 |
| GISS_E2_R | 0.048 | 0.147 | 2.11 | 1.50 | 20.8 |
| HadGEM2_ES | 0.027 | 0.177 | 4.59 | 2.50 | 37.4 |
| IPSL_CM5A_LR | 0.022 | 0.130 | 4.13 | 2.00 | 45.9 |
| MIROC5 | 0.027 | 0.107 | 2.72 | 1.50 | 36.6 |
| MIROC_ESM | 0.021 | 0.140 | 4.67 | 2.20 | 48.0 |
| MPI_ESM_LR | 0.027 | 0.143 | 3.63 | 2.00 | 36.7 |
| MRI_CGCM3 | 0.034 | 0.127 | 2.60 | 1.60 | 29.5 |
| NorESM1_M | 0.023 | 0.093 | 2.80 | 1.40 | 43.5 |
| **Multimodel Mean** | **0.032** | **0.146** | **3.35** | **1.90** | **34.5** |
| **Standard Deviation** | 0.010 | 0.036 | 0.792 | 0.342 | 10.350 |

**Table 2: PH99 parameters ($\alpha$ and $\mu$), climate sensitivities ($ECS$ and $TCR$), and feedback response times ($1/\alpha$) after fitting PH99 GMT time series to AOGCM RCP2.6 GMT time series.**

| | PH99 Parameters | | Climate Sensitivities | | Feedback Response Times |
|---|---|---|---|---|---|
| | $\alpha$ [1/yrs] | $\mu$ [K/yrs] | $ECS$ [K] | $TCR$ [K] | $1/\alpha$ [yrs] |
| bcc_csm1_1_m | 0.058 | 0.199 | 2.37 | 1.79 | 17.20 |
| bcc_csm1_1 | 0.080 | 0.267 | 2.32 | 1.90 | 12.51 |
| CanESM2 | 0.093 | 0.377 | 2.81 | 2.37 | 10.74 |
| CCSM4 | 0.082 | 0.264 | 2.24 | 1.85 | 12.21 |
| CNRM_CM5 | 0.084 | 0.329 | 2.73 | 2.26 | 11.97 |
| CSIRO_Mk3_6_0 | 0.079 | 0.280 | 2.45 | 2.00 | 12.61 |
| GISS_E2_R | 0.345 | 0.746 | 1.50 | 1.44 | 2.90 |
| HadGEM2_ES | 0.114 | 0.485 | 2.94 | 2.57 | 8.75 |
| IPSL_CM5A_LR | 0.046 | 0.201 | 3.01 | 2.11 | 21.58 |
| MIROC5 | 0.158 | 0.455 | 1.99 | 1.81 | 6.32 |
| MIROC_ESM | 0.096 | 0.478 | 3.45 | 2.93 | 10.41 |
| MPI_ESM_LR | 0.088 | 0.344 | 2.70 | 2.26 | 11.33 |
| MRI_CGCM3 | 0.059 | 0.178 | 2.09 | 1.58 | 16.93 |
| NorESM1_M | 0.105 | 0.292 | 1.92 | 1.66 | 9.49 |
| **Multimodel Mean** | **0.106** | **0.350** | **2.46** | **2.04** | **11.78** |
| **Standard Deviation** | 0.074 | 0.152 | 0.512 | 0.409 | 4.639 |

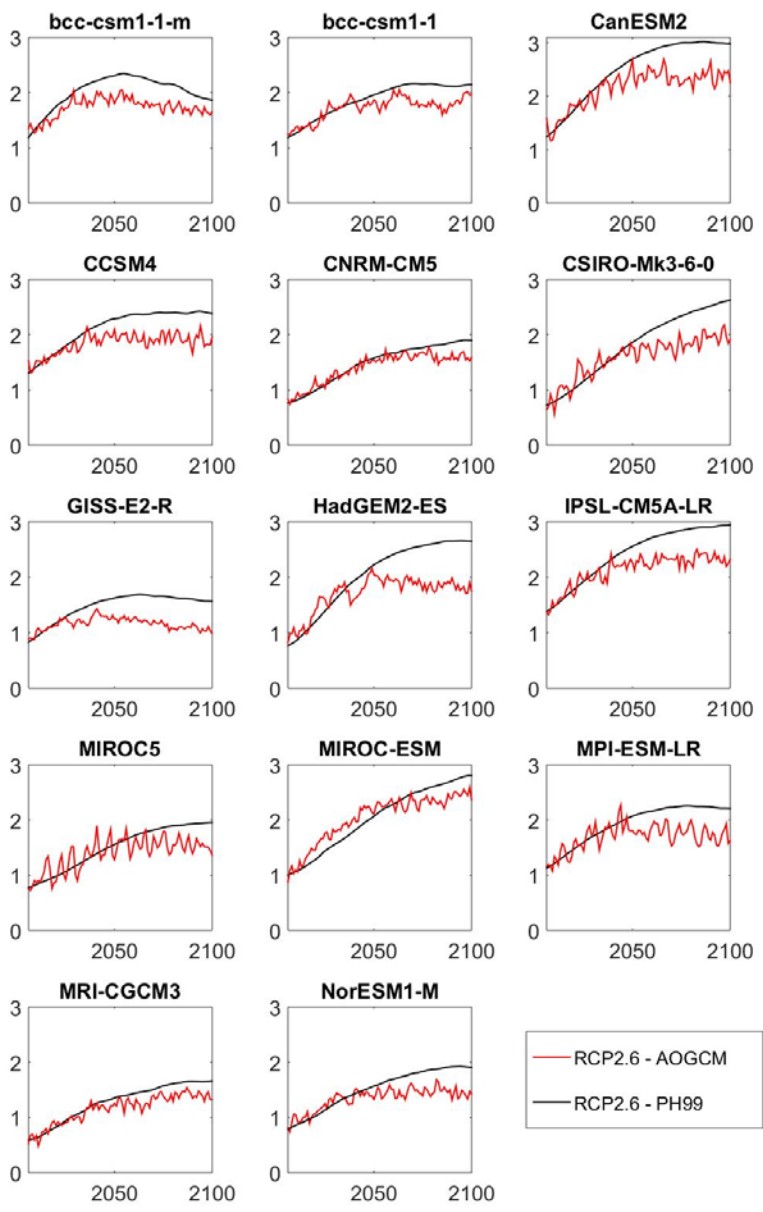

**Figure 1: Comparison of temperature paths [K] projected by PH99 (black curve), calibrated by an AOGCM's ECS and TCR, to the corresponding AOGCM's temperature paths (red curve). Deviations on the order of 0.5 K for 2100 are observed.**

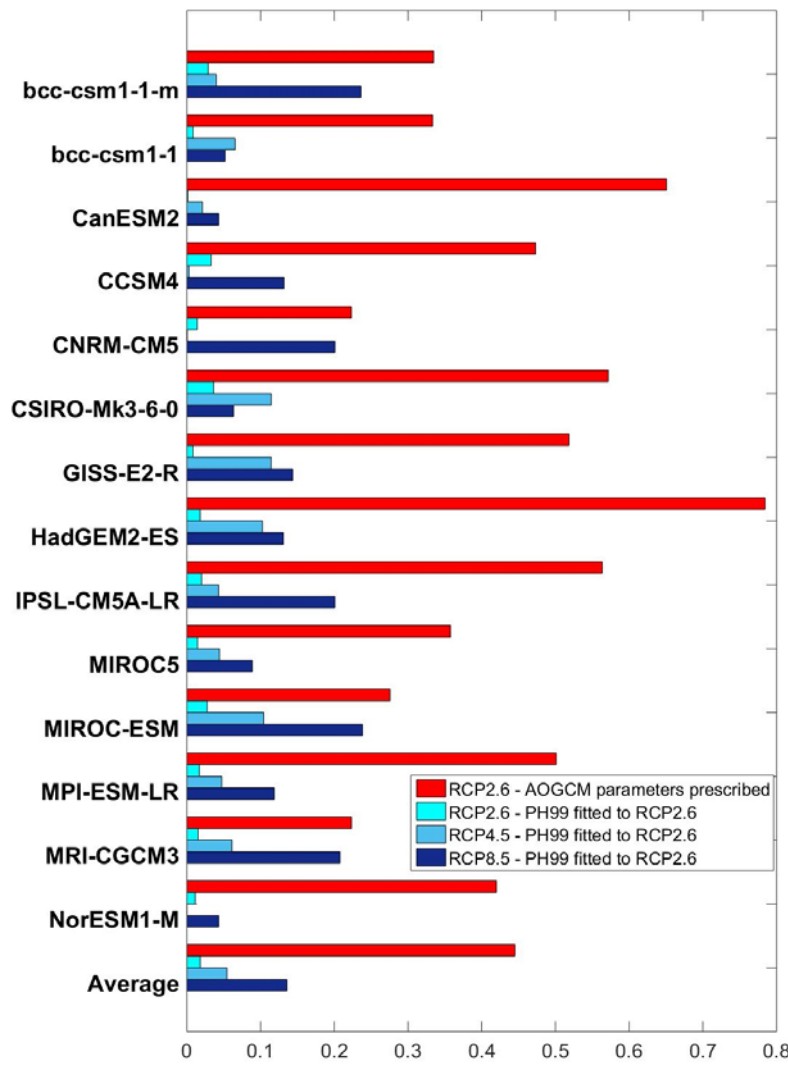

**Figure 2: Modulus of deviations of GMT [K] mean values of PH99 over the period 2071-2100 from corresponding AOGCM means. The red bars show the deviations for RCP2.6 when $\alpha$ and $\mu$ are from Table 1 and not fitted. The cyan bars show the deviations in RCP2.6 when $\alpha$ and $\mu$ are fitted to the AOGCM's RCP2.6 data. The light blue bars show the deviations for RCP4.5 when $\alpha$ and $\mu$ are kept at their RCP2.6-fitted values (validation). The dark blue bars show the deviations for RCP8.5 when $\alpha$ and $\mu$ are kept at their RCP2.6 fitted values (validation).**

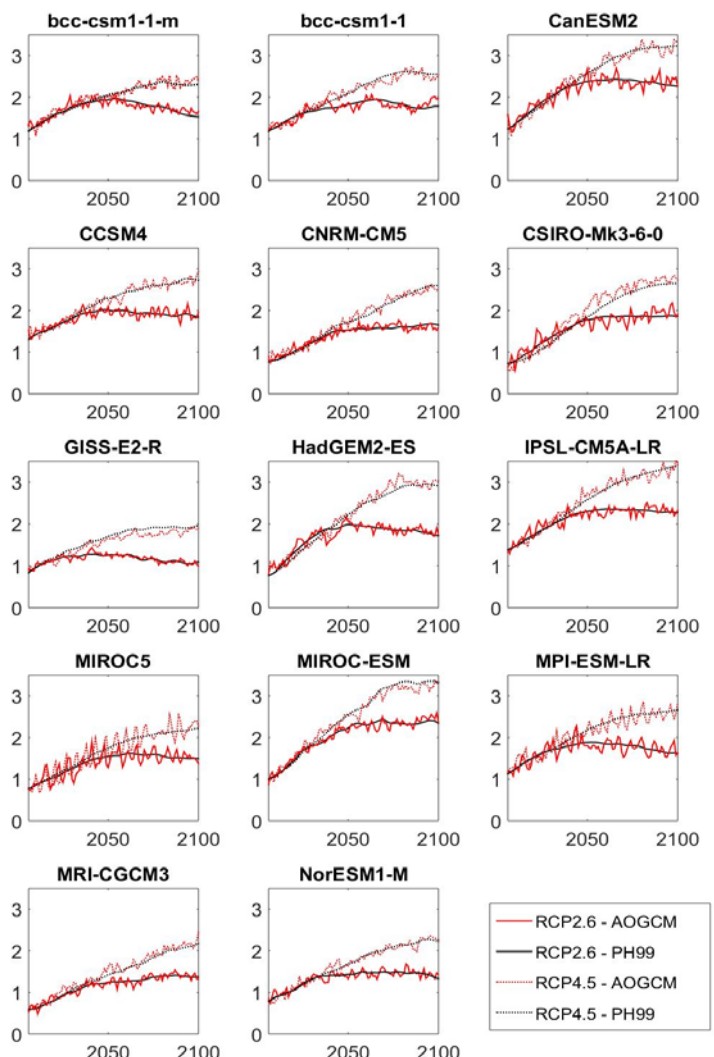

**Figure 3: Comparison of temperature evolutions [K] projected by the climate module PH99 (solid and dotted black curves) to the actual AOGCM's temperature (solid and dotted red curves). $\alpha$ and $\mu$ have been tuned to fit the PH99 temperature path (solid black curve) to the respective AOGCM's RCP2.6 temperature path (solid red curve). Using the fitted $\alpha$ and $\mu$, and taking the forcing reconstructed for RCP4.5 into account, PH99 also reproduces the projected RCP4.5 (dotted black curve). The dotted red curve shows the actual RCP4.5 temperatures.**

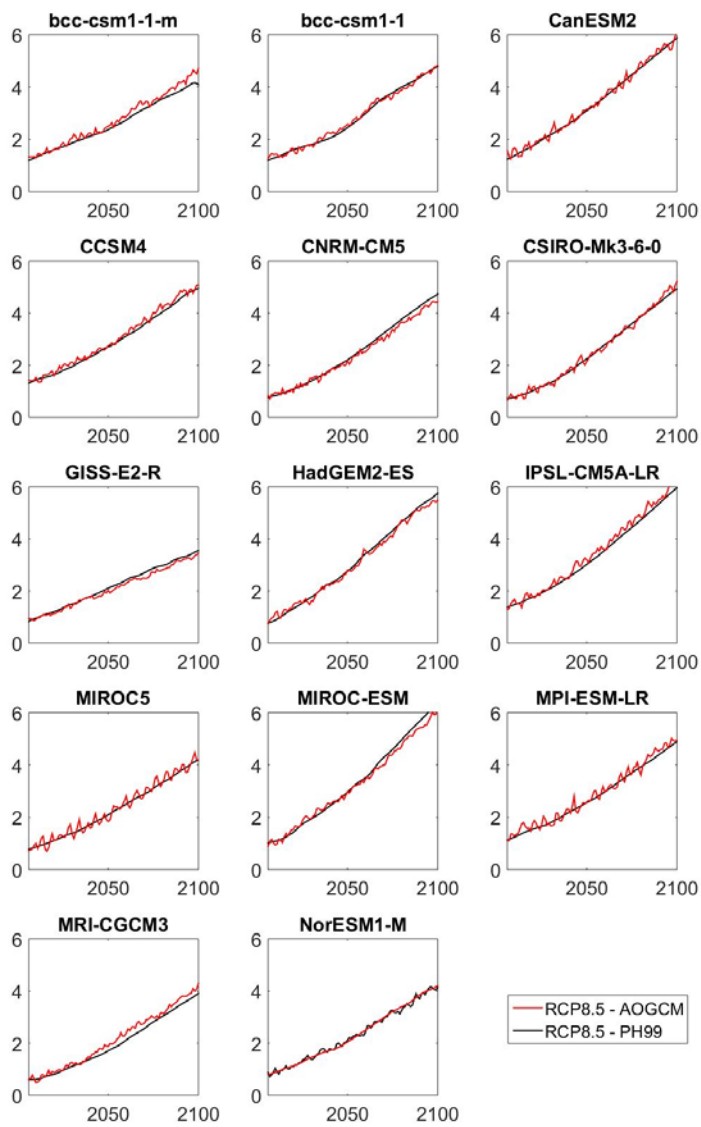

**Figure 4: Comparison of temperature evolutions [K] projected by the climate module PH99 (black solid curves) in RCP8.5 scenario to the actual AOGCM's temperature (red solid curves) in RCP8.5 scenario. $\alpha$ and $\mu$ are taken from the second step, where PH99 is calibrated to RCP2.6 scenario.**

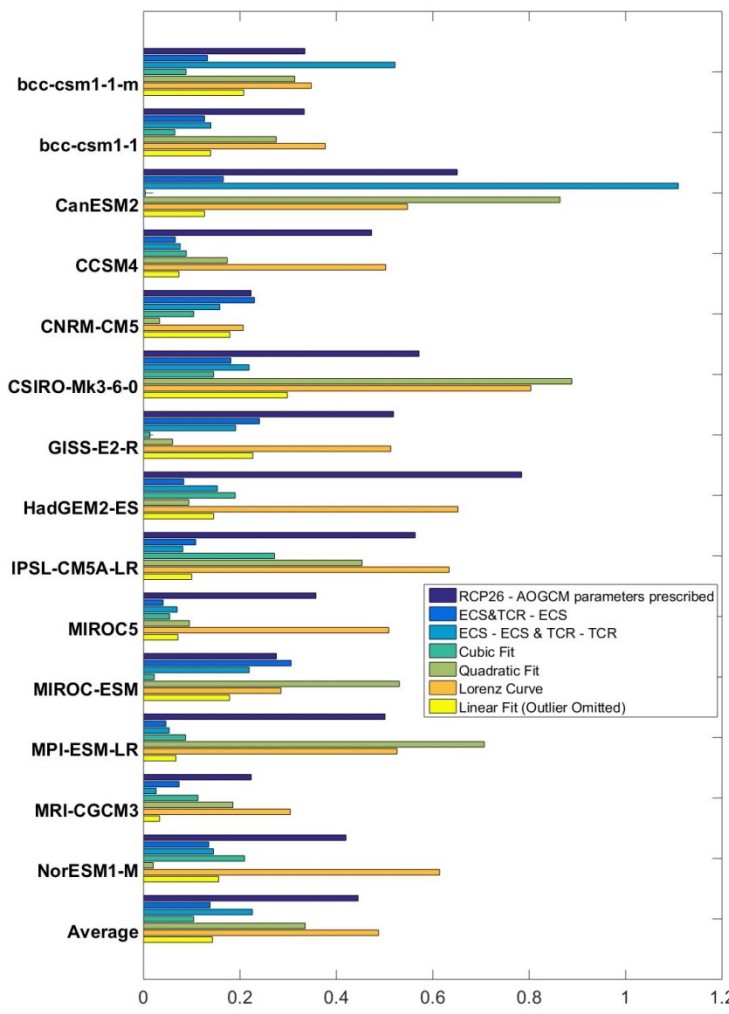

**Figure 5: Modulus of mean temperature deviations [K] over the period 2071-2100 (MTD) for PH99 from AOGCMs when $\alpha$, $\mu$, ECS, and TCR from Table 2 are related to ECS and TCR in Table 1. Using linear (yellow bars), quadratic (light green bars), and cubic functions (dark green bars), $\alpha$ and $\mu$ are related to ECS when outlier is put out for the linear case. Using linear fits, ECS and TCR are related to ECS (blue bars). Using linear fits, ECS and TCR are related to ECS and TCR respectively (light blue bars). The dark blue bars show the deviations for RCP2.6 when $\alpha$ and $\mu$ are from Table 1 and not fitted (the same as Fig.2). The orange bars indicate MTD using Lorenz's curve.**

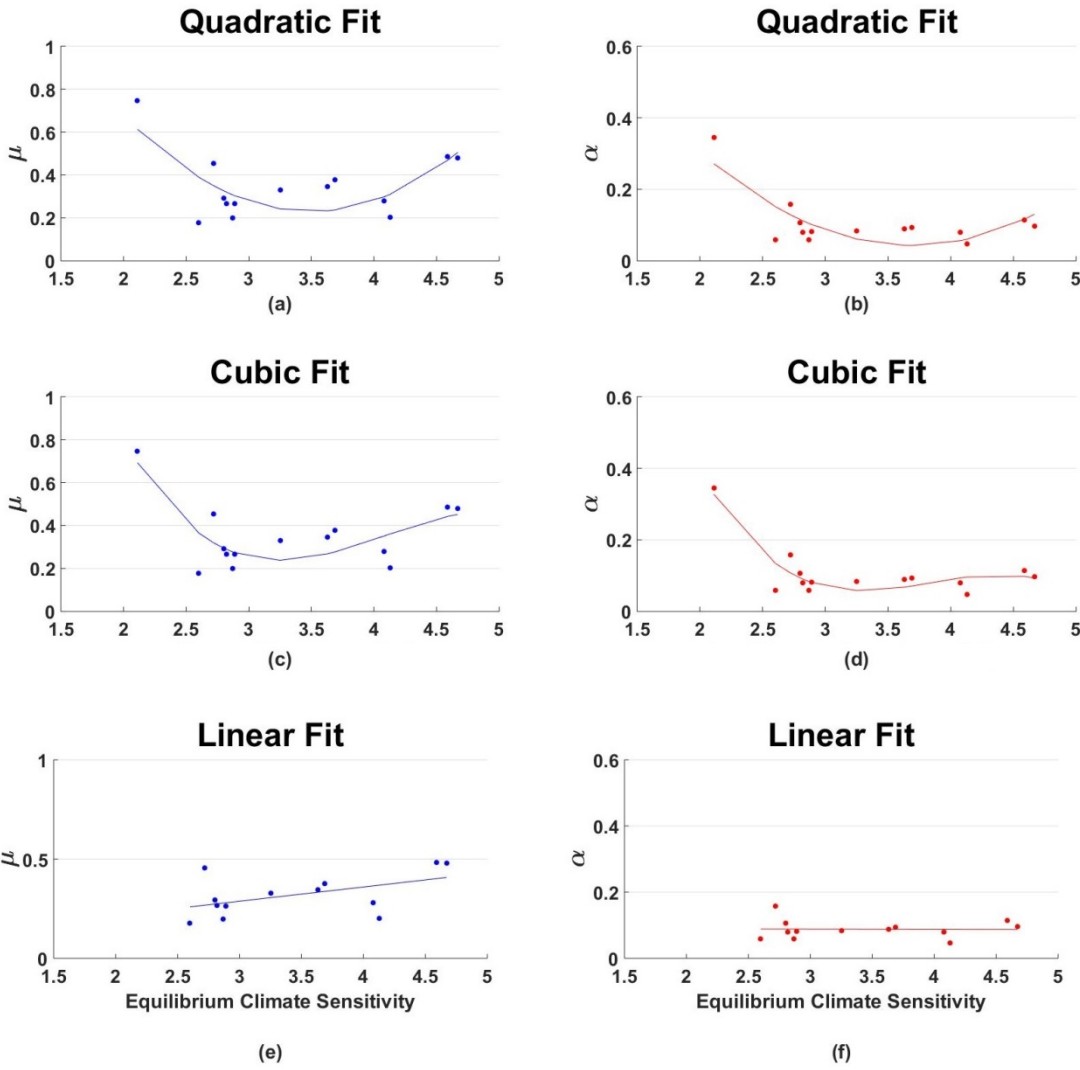

**Figure 6: Quadratic (up), cubic (middle), and linear (down) relationships of $\mu$ (left) and $\alpha$ (right) in Table 2 to ECS in Table 1. Notice that in the linear case the model GISS_E2_R (the upper left sample), as an outlier, is out.**

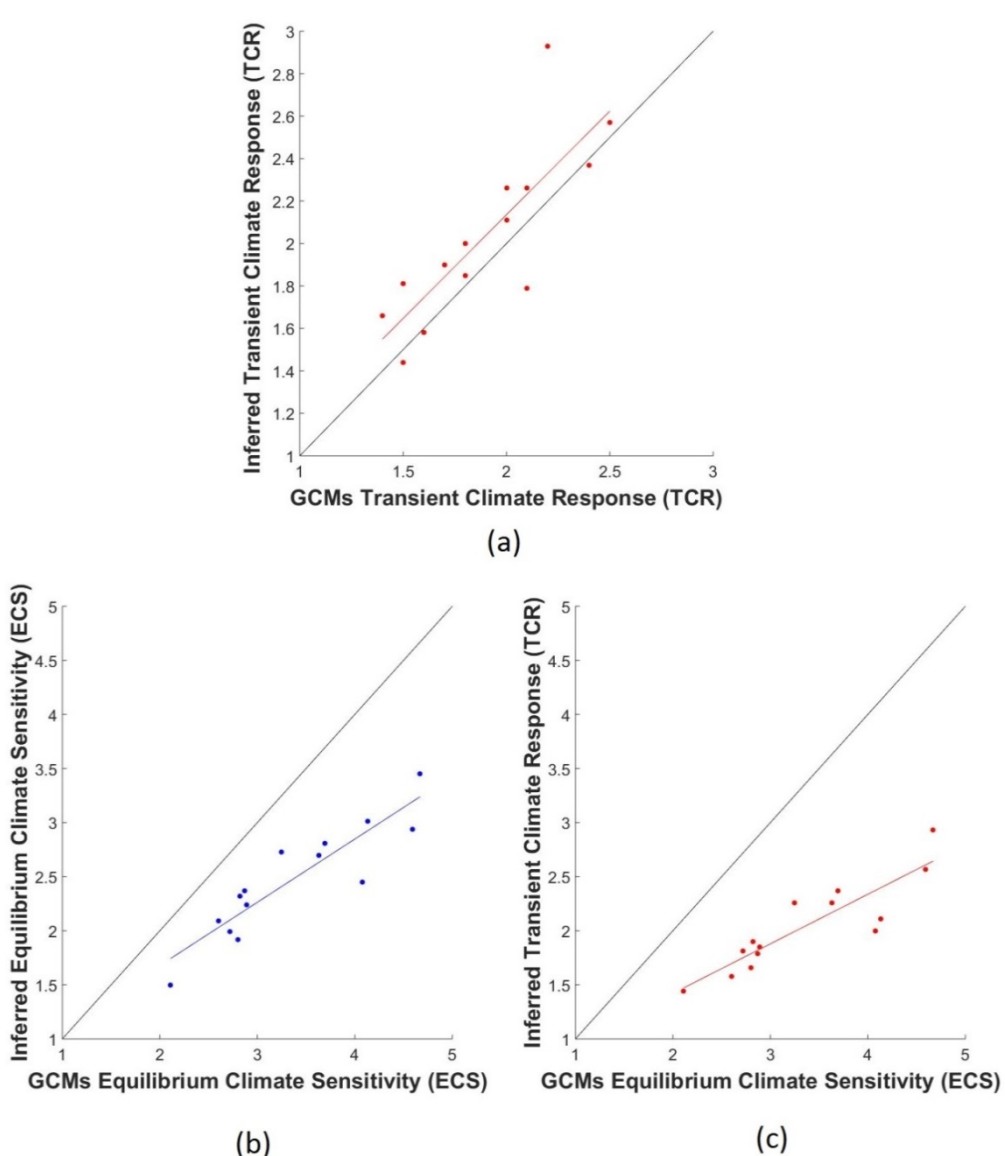

**Figure 7: Inferred effective TCR [K] vs. AOGCMs' TCR [K] (a), inferred effective ECS [K] vs. AOGCMs' ECS [K] (b), and inferred effective TCR [K] vs. AOGCMs' ECS [K] (c). While the TCRs differ by less than 0.2 K, the ECSs differ by up to 2 K. This opens the door for a discussion as to whether PH99 should be calibrated using scenario-class-adjusted effectively lower ECS values.**

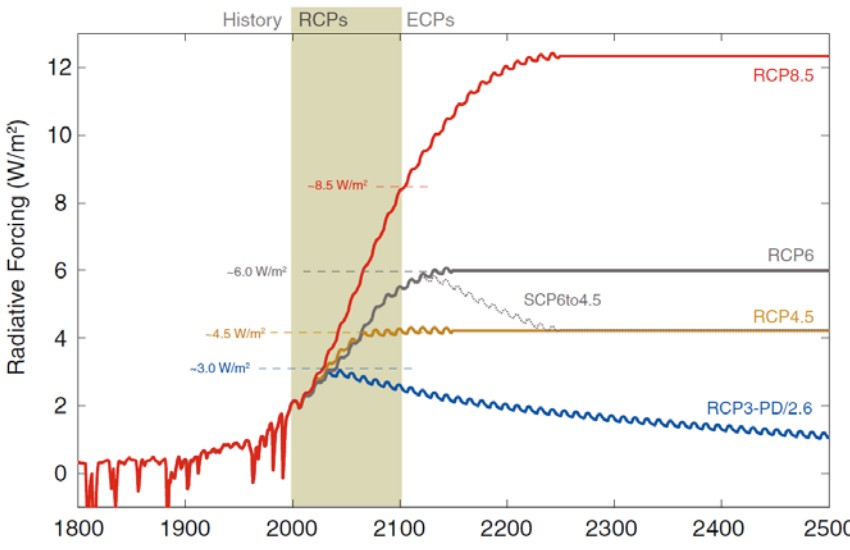

**Figure 8: Total radiative forcing (anthropogenic plus natural) for RCPs – supporting the original names of the four pathways, as there is a close match between peaking, stabilization and 2100 levels for RCP2.6 (also called RCP3-PD), RCP4.5 & RCP6, and RCP8.5, respectively (taken from Meinshausen et al. (2011b)).**

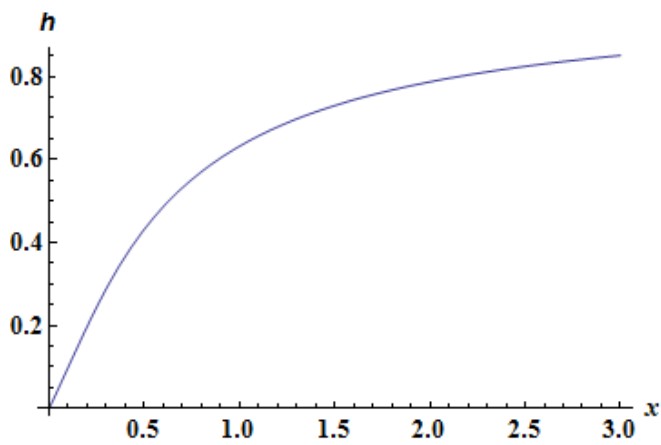

**Figure 9: The auxiliary function $h(x)$, which links the slow time scale of the 2-box model and the time scale of the 1-box model.**

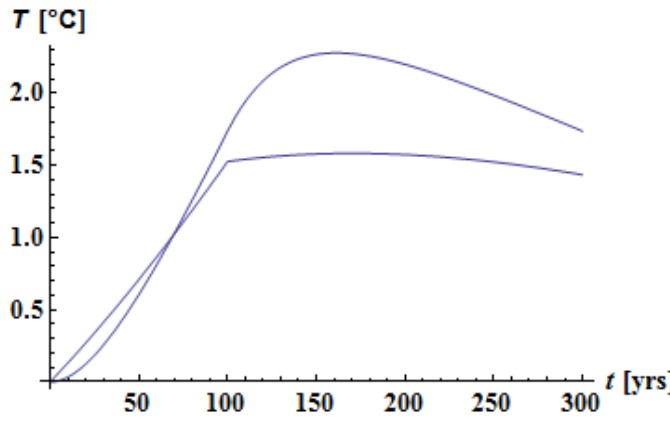

**Figure 10: 1-box vs. 2-box model in response to kink-linear forcing as a stylized interpretation of mitigation-oriented forcing paths and for equal levels of ECS and TCR in both models. Kink-linear curve: 2-box model, smooth curve: 1-box model. The temperature development of the 1-box model overshoots the maximum of the 2-box model by roughly 50%.**

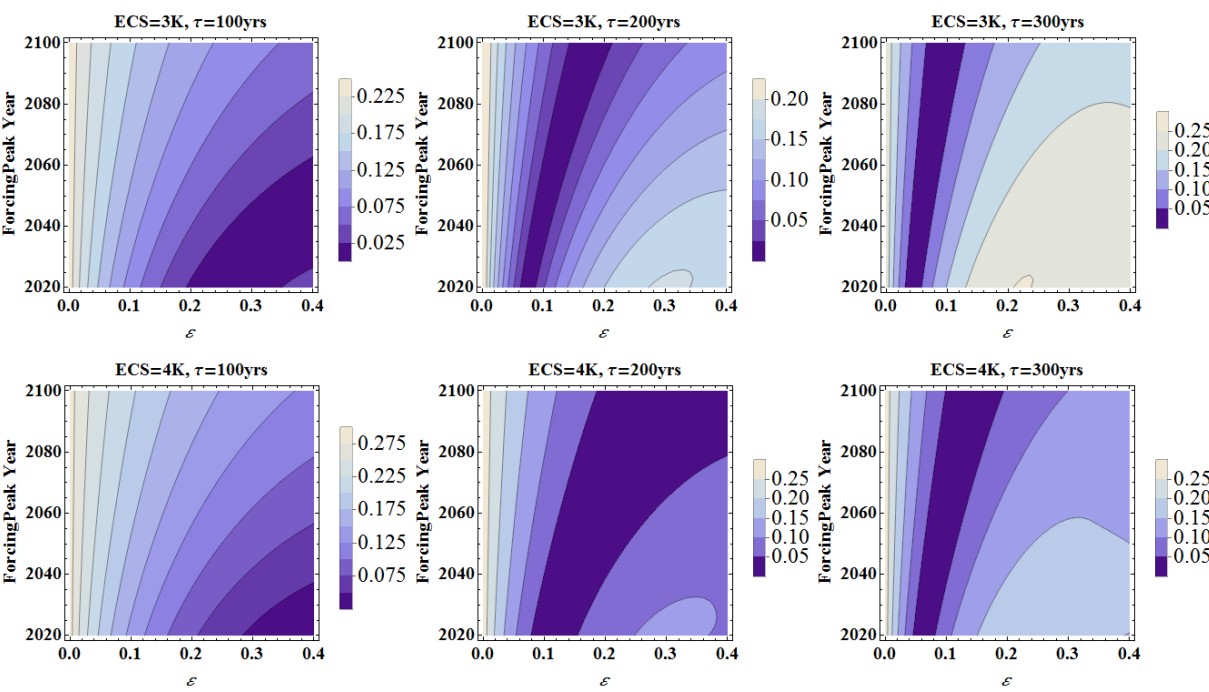

5    **Figure 11: : Comparing GMT [K] maxima of the 2-box model and the 1-box model, the latter being adjusted to the former by prescribing the linearly transformed ECS and TCR according to the scheme ETE. Abscissa: $\varepsilon$, ordinate: changed peaking year $t_1$, however transformed to years, for the 2-box ECS of 3 K and 4 K, and $\tau$=100, 200, 300 yrs, respectively. The relative error (max. GMT difference normalized by the max. GMT of the 2-box model) is markedly smaller than for the case of prior adjustment.**

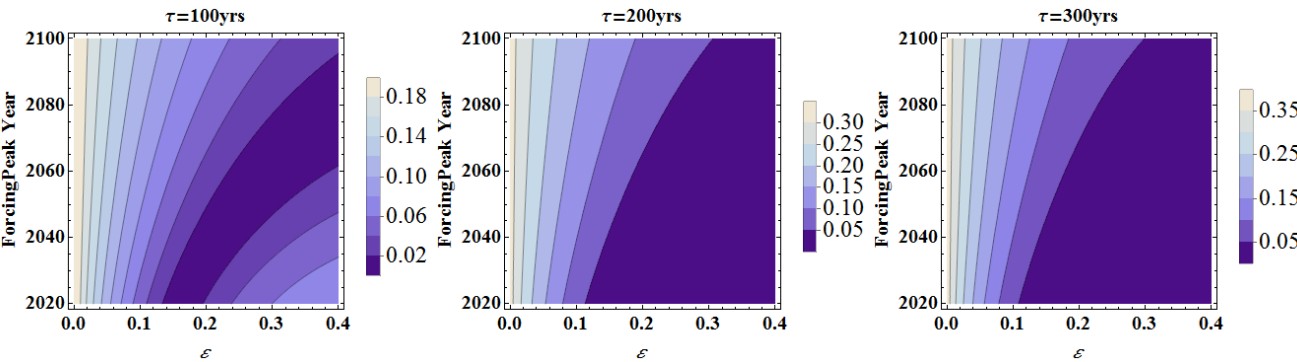

**Figure 12: Similar to the previous figure (relative max. GMT error with abscissa: $\varepsilon$, ordinate: $t_1$ [yrs],), however for a further adjusted ECS of the 1-box model, such that perfect matching is achieved for $t_1$=100 yrs, $\varepsilon$=0.2, and a 1-box time scale of 12 yrs. For most of the parameter settings, the relative error is below 10%.**

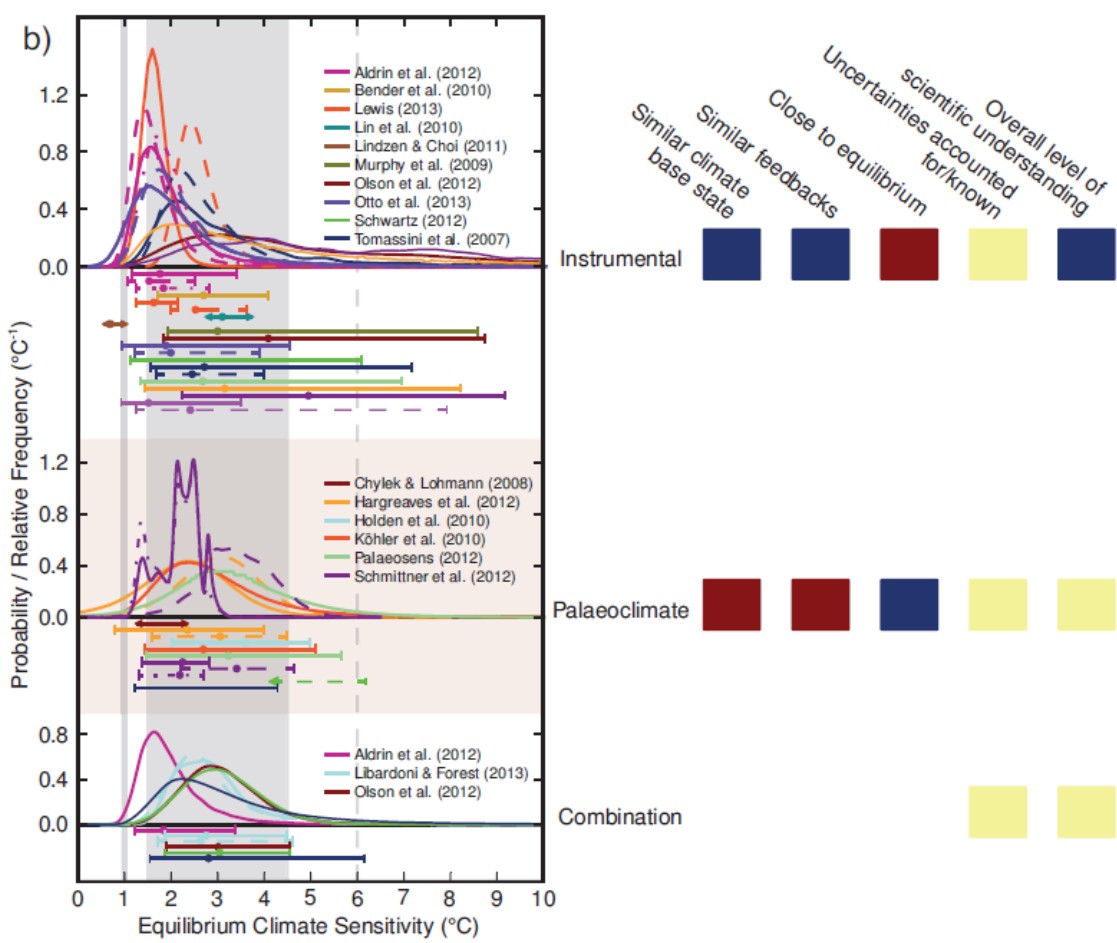

**Figure 13: Probability density distributions of ECS according to IPCC AR5 WG-I (Bindoff et al., 2013, Fig. 10.20).**

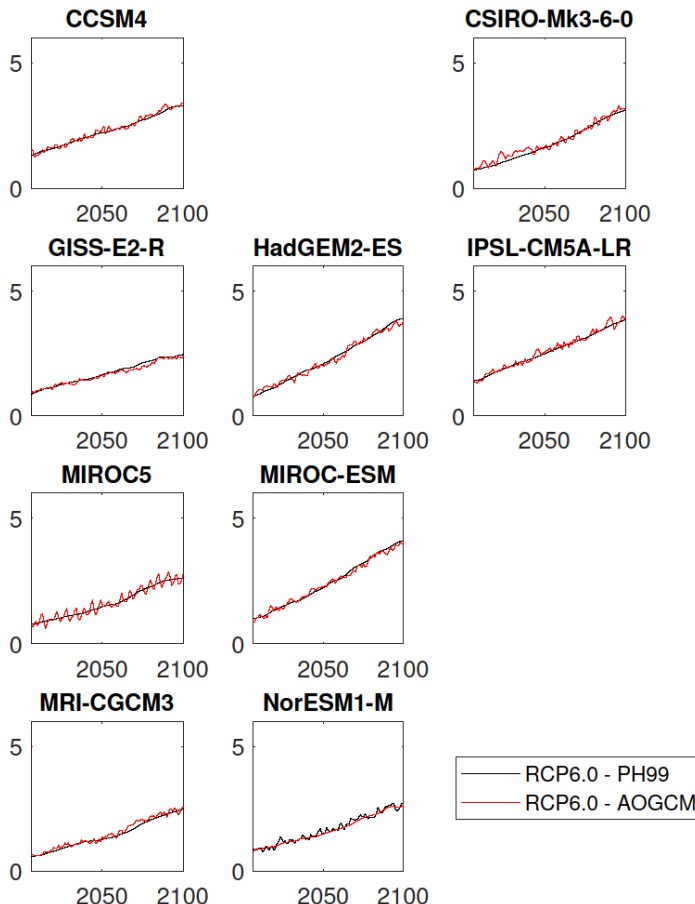

**Figure 14: The comparison of temperature evolutions projected by the climate module PH99 (black solid curves) in RCP6.0 scenario to the actual AOGCM's temperature (red solid curves) in RCP6.0 scenario. $\alpha$ and $\mu$ are taken from the second step, where PH99 is calibrated to RCP2.6 scenario**

Table 3: Modulus of mean temperature deviations over the period 2071-2100 (MTD) for PH99 from corresponding AOGCM. In the first 4 columns, PH99 is calibrated to RCP 2.6. In the second 4 columns, PH99 is calibrated to RCP 4.5.

| | Calibrated to RCP 2.6 | | | | Calibrated to RCP 4.5 | | | |
|---|---|---|---|---|---|---|---|---|
| | **MTD RCP2.6** | MTD RCP4.5 | MTD RCP6.0 | MTD RCP8.5 | MTD RCP2.6 | **MTD RCP4.5** | MTD RCP6.0 | MTD RCP8.5 |
| bcc_csm1_1_m | **0.029** | 0.040 | | 0.236 | 0.018 | **0.007** | | 0.154 |
| bcc_csm1_1 | **0.009** | 0.066 | | 0.052 | 0.064 | **0.021** | | 0.059 |
| CanESM2 | **0.001** | 0.021 | | 0.043 | 0.039 | **0.003** | | 0.018 |
| CCSM4 | **0.033** | 0.003 | 0.069 | 0.132 | 0.024 | **0.005** | 0.064 | 0.128 |
| CNRM_CM5 | **0.014** | 0.001 | | 0.201 | 0.005 | **0.012** | | 0.273 |
| CSIRO_Mk3_6_0 | **0.036** | 0.115 | 0.040 | 0.063 | 0.017 | **0.015** | 0.168 | 0.278 |
| GISS_E2_R | **0.008** | 0.114 | 0.094 | 0.144 | 0.064 | **0.003** | 0.027 | 0.015 |
| HadGEM2_ES | **0.018** | 0.103 | 0.036 | 0.131 | 0.057 | **0.020** | 0.097 | 0.211 |
| IPSL_CM5A_LR | **0.020** | 0.043 | 0.050 | 0.201 | 0.121 | **0.013** | 0.017 | 0.033 |
| MIROC5 | **0.015** | 0.044 | 0.029 | 0.089 | 0.032 | **0.009** | 0.034 | 0.106 |
| MIROC_ESM | **0.028** | 0.104 | 0.079 | 0.238 | 0.140 | **0.012** | 0.044 | 0.241 |
| MPI_ESM_LR | **0.017** | 0.047 | | 0.119 | 0.108 | **0.015** | | 0.060 |
| MRI_CGCM3 | **0.015** | 0.061 | 0.083 | 0.208 | 0.001 | **0.007** | 0.004 | 0.061 |
| NorESM1_M | **0.011** | 0.000 | 0.026 | 0.043 | 0.024 | **0.000** | 0.006 | 0.082 |
| **Multimodel Mean** | **0.018** | 0.054 | 0.056 | 0.136 | 0.035 | **0.005** | 0.039 | 0.078 |
| **Standard Deviation** | **0.010** | 0.041 | 0.025 | 0.071 | 0.044 | **0.006** | 0.053 | 0.093 |

**Table 3 (continued): Modulus of mean temperature deviations over the period 2071-2100 (MTD) for PH99 from corresponding AOGCM. In the third 4 columns, PH99 is calibrated to RCP 6.0. In the fourth 4 columns, PH99 is calibrated to RCP 8.5.**

| | Calibrated to RCP 6.0 | | | | Calibrated to RCP 8.5 | | | |
|---|---|---|---|---|---|---|---|---|
| | MTD RCP2.6 | MTD RCP4.5 | **MTD RCP6.0** | MTD RCP8.5 | MTD RCP2.6 | MTD RCP4.5 | MTD RCP6.0 | **MTD RCP8.5** |
| bcc_csm1_1_m | | | | | 0.287 | 0.257 | | **0.027** |
| bcc_csm1_1 | | | | | 0.091 | 0.008 | | **0.039** |
| CanESM2 | | | | | 0.008 | 0.025 | | **0.010** |
| CCSM4 | 0.038 | 0.067 | **0.018** | 0.086 | 0.059 | 0.004 | 0.010 | **0.004** |
| CNRM_CM5 | | | | | 0.117 | 0.151 | | **0.005** |
| CSIRO_Mk3_6_0 | 0.161 | 0.199 | **0.019** | 0.062 | 0.119 | 0.019 | 0.034 | **0.015** |
| GISS_E2_R | 0.041 | 0.037 | **0.019** | 0.046 | 0.045 | 0.023 | 0.011 | **0.001** |
| HadGEM2_ES | 0.146 | 0.233 | **0.021** | 0.063 | 0.146 | 0.252 | 0.073 | **0.017** |
| IPSL_CM5A_LR | 0.016 | 0.077 | **0.001** | 0.095 | 0.052 | 0.078 | 0.030 | **0.002** |
| MIROC5 | 0.067 | 0.079 | **0.011** | 0.032 | 0.025 | 0.006 | 0.019 | **0.019** |
| MIROC_ESM | 0.187 | 0.070 | **0.005** | 0.198 | 0.309 | 0.235 | 0.140 | **0.007** |
| MPI_ESM_LR | | | | | 0.011 | 0.082 | | **0.012** |
| MRI_CGCM3 | 0.092 | 0.068 | **0.003** | 0.042 | 0.008 | 0.014 | 0.055 | **0.027** |
| NorESM1_M | 0.068 | 0.021 | **0.016** | 0.136 | 0.070 | 0.055 | 0.054 | **0.013** |
| **Multimodel Mean** | 0.091 | 0.095 | **0.007** | 0.084 | 0.096 | 0.086 | 0.029 | **0.014** |
| **Standard Deviation** | 0.060 | 0.072 | **0.008** | 0.053 | 0.096 | 0.096 | 0.041 | **0.011** |

