# Peer review of "On the Future Role of the Most Parsimonious Climate Module in Integrated Assessment"

_Earth System Dynamics, 2017_

## Referee Comment (RC1) · Anonymous Referee #1 · 6 Jun 2017

The manuscript by Khabbazan and Held assesses the performance of a very simplified climate module currently in use in some IAMs. In particular, the study is motivated by the need to adjust the existing tools to the capability of this module in the light of assessments of below 2°C scenarios. To that end, it is fitted to different CMIP5 RCP2.6 AOGCMs.

The manuscript contains no fundamental flaws although a re-read is in order and the literature list should be checked. One key paper (Foster et al. 2013) is for example missing from the literature list. I presume it's Forster, P. M., T. Andrews, P. Good, J. M. Gregory, L. S. Jackson, and M. Zelinka, 2013: Evaluating adjusted forcing and model

spread for historical and future scenarios in the CMIP5 generation of climate models. J. Geophys. Res. Atmos., 118, 1139– 1150

More fundamentally, however, the scientific advancement presented of this study in my assessment is rather poor and it neglects important recent literature in this context (in fact, the literature list is rather short and at least 3 years old).

On the methodological approach: What's the justification of using the PH99 model (apart from it 'being there')? The authors argue that it's computational efficiency, but how exactly are they convinced that their treatment of non-CO2 GHGs is appropriate. For strong mitigation pathways, these 'minor' differences may become very important, last but not least to determine net-zero global GHG forcing etc. I'd think they would need validate their fit using other strong mitigation scenarios with different non-GHG trajectories (if no others are available then from the GeoMIP experiment) rather than RCP4.5. In particular, it appears that non-CO2 gases obscure our assessments of ECS (see e.g. Myre et al. 2016)

As it stands, I'm not convinced that the simplified model is capable of including non-GHG forcing in a sufficient fashion for the question at hand (i.e. staying below 2°C or 1.5°C).

On the application: I didn't fully the motivation for step 1. What was the reasoning for the authors to assume that their PH99 model would work with AOGCM diagnostics from Forsters et al. directly? Obviously, the derived feedback response time parameter 1/alpha of 34.5 years in the multi-model mean is quite unphysical. It seems that the PH99 model is not equipped to be used in that context.

In a next step, the authors find that with two free parameters they are capable of achieving better fits. That's not particularly surprising, but a physical interpretation of these differences is virtually absent? ECS is substantial decreased by almost 1°C. Can this be understood? The authors continue with fitting derived and fitted ECS and TCR, but I would rather like to see a physical interpretation or an extension of the PH99 model that

would correct for this. The authors should also consider their approach in the light of alternative simplified approaches out there i.e. based on a response function approach as in Ragone et al. (2016).

The authors then want to apply an effective correction for their dubious model in the first place. Their results here appear to be prone to outliers. Compare e.g. the low ECS outlier in Fig. 5. When removing it, I guess even a linear fit would deliver decent results and I'm not sure I can deduce any robust trends from these graphs. . .

On the application of this. It seems that the model that is used in these IAMs has many flaws. The question then becomes why it is used at all? And not abandoned for a carbon budget approach that would be even more computational effective and can be determined with more complex models also for these low emissions scenarios (i.e. Rogelj 2016). That becomes in particular relevant since the mitigation challenge ahead is to define pathways that hold warming 'well below 2°C'. "Below 2°C" was interpreted as a 66% chance of non-exceedance (IPCC 2014). What's the added value of using a PH99 model in this context? Would they select an ECS at the 66% quantile and then use this as a basis for the IAM derivations? And if so, why not use carbon budgets directly?

---

## Referee Comment (RC2) · Anonymous Referee #2 · 16 Jun 2017

**1   General Comments**

Khabbazan and Held's paper checks the performance of a one box energy balance model (PH99), currently in use in the integrated assessment models FUND and MIND, against output from AOGCMs before suggesting a simple, improved way to use it in future. Their major conclusion is that, for strong mitigation scenarios, prescribing ECS and TCR to PH99 from Forster et al. (2013) with no further calibration implicitly causes researchers to sample much larger temperature responses than they intend to. They show that a simple fitting exercise rectifies this and validate the fit by checking PH99's performance under one other scenario. This scenario is very similar to the one they

used for fitting. They then explore different methods to map AOGCM ECS and TCR onto 'effective' PH99 values which could provide researchers with a simple method of revealing the temperature response they are actually considering.

My major concerns focus on whether the analysis shows that PH99 is a valid energy balance model rather than a fitting tool. I also think that the writing style could be greatly improved. I think the authors point out some key errors which arise if PH99 is used without care and explore a few ways for modellers to quickly relate their parameters to AOGCM ECS and TCR values. However, given that the authors argue for mapping AOGCM properties onto 'scenario-class-specific values before using them in PH99', which appears to undermine any physical basis for PH99, I'm left wondering if this paper highlights the limitations of PH99 rather than providing strong arguments for its use.

**2 Major concerns**

1. The re-callibration of PH99 is only validated for RCP4.5. There is no other testing of the performance outside of RCP2.6 and RCP4.5, two very similar scenarios, nor testing of the effect of different non-$CO_2$ forcing pathways. Thus the authors have shown that a good fit to AOGCM GMT output can be done with two free parameters and that this fit is good for a similar scenario. I wonder if testing over a greater range of other scenarios would strengthen the justification for the use of PH99.

2. The initial testing of the performance of PH99 against AOGCMs reveals a key, hidden, bias of this model if used without validation in strong mitigation scenarios. This is a good bit of analysis. As a result of this analysis, the authors advocate mapping AOGCM climate system properties onto 'scenario-class-specific values before using them in PH99'. Whilst this seems to be necessary for accept-

able performance of PH99, it also appears to undermine any physical basis for PH99. If you have to re-callibrate PH99 every time you want to use it in a different scenario class then its parameters lose all physical meaning and instead simply become fitting parameters. Thus the authors appear to advocate shifting PH99 from an energy balance model to a function that can be fitted to AOGCM data and then used for a limited range of scenarios?

3. I don't think I am wrong in saying that this model is ultimately meant to be used by those who are looking for simple emulators of global mean temperature response and hence may not be climate modellers themselves. If this target audience can't pick up this paper and get some sense of what is going on then they will struggle to use any of the fits provided. A paper on 'the most parsimonious climate module' should have a style which reflects its title. Given that parsimonious is synonymous with 'simple' in this context, it makes sense for the communication to be as plain, clear and simple as possible too. With this goal in mind, I suggest numerous technical corrections and ask for multiple clarifications.

4. The exploration of different possible parameterisations of the relationship between AOGCM ECS/TCR and effective ECS/TCR is, in my opinion, worthwhile. My impression is that they recognise that a parameterisation would be nice but don't have strong enough evidence to recommend any of the ones they have tried and so the results here are underwhelming.

**3   Specific Comments**

1. As an exercise, the fitting that is done is scientifically sound re methods, assumptions, results, and reproducibility as far as I can tell. I can also see that it would be useful for modellers who wish to use a simple emulator but don't wish to do the calibration themselves.

2. I think this paper shows that PH99 is closer to a fitting tool rather than a physical model. Hence I wonder, if modellers are after computational simplicity and a fitting tool, why wouldn't they use a simple carbon budget target or emissions pathway to constrain their model. There is already research on how emissions pathways and targets map to temperature targets so this could be used to back out emissions constraints from a given temperature target for a given scenario class. This approach seems far simpler than introducing an energy balance module which requires atmospheric concentration and radiative forcing input, has little physical basis and hasn't been validated over a wide range of $CO_2$ and non-$CO_2$ scenarios so might not produce realistic temperature projections anyway.

3. The introduction calls IAMs an 'indispensable tool'. I acknowledge that this comment is made in the context of 'driving welfare-optimal climate policy scenarios' so it is accurate. However given that there are many who disagree with using economic analyses for determining 'welfare-optimal scenarios' because of the need to monetise many things which arguably can't be monetised (e.g. the environment), using this term seems to open the paper up to unwanted distractions. I think this could be avoided with a simple re-wording; calling IAMs a 'tool which are used to derive welfare-optimal scenarios' rather than an 'indispensable tool used to derive welfare-optimal scenarios'. This change would avoid opening up an economic debate (in the reader's mind) which is completely outside the scope of this paper.

4. page 8, line 22: 'personal conviction'. I don't think personal convictions have any place in scientific papers. Either the evidence is there to support using log-normal distributions or it's not. I also don't understand what the sentence beginning with 'This conviction rests' means. Does it mean 'Schneider von Deimlinig et al. show that constraining ECS by paleo data results in thin-tailed distributions'? If yes, then there is no need for a 'personal conviction', circling back to my first point.

5. I really appreciated the discussion of low pass filtering and think this was well done.

6. In section 4 (page 7, lines 32-34), the authors state that 'regressing both inferred effective ECS and TCR solely against AOGCMs' ECS obviously is the overall better approximation'. Whilst this is borne out by taking a pure average of all the results, there are clearly two strong outliers which are having a major effect on the performance of the ECS-ECS & TCR-TCR mapping. I wonder what is causing such large outliers (they seem hugely anomalous) and if removing them would be justified. If they are removed, how much does this change the conclusions.

**4  Technical Corrections**

4.1  Graphs

Labelling of axes and units could be greatly improved on all figures

Figure 1: Adding something to indicate that 2071-2100 was the most important region for validating the model might be useful for the reader

Figure 2: Is the legend covering part of the figure?

Figure 4: Is the legend covering part of the figure?

Figure 5: Would it be worth adding fitting parameter values to the figure or caption?

Figure 6: Would it be worth adding fitting parameter values to the figure or caption?

Figure 7: Please add more explanation of the caption to the figure e.g. what do the different colours on the right hand side of the figure represent?

**4.2 Text**

Abstract: It might be worth swapping sentences 2 and 3 with sentences 4 and 5?

page 1, line 6: This is effectively an 'energy balance' model, I don't think the term 'climate' model is appropriate for one box models

page 1, line 15: replace 'effective' with 'equivalent'?

page 1, line 15: delete 'intrinsic'

page 1, line 16-17: 'in particular when computationally demanding decision-making under climate response uncertainty continues to be modelled' should either be 'in particular as computationally demanding decision-making under climate response uncertainty continues to be modelled' or 'in particular when computationally demanding decision-making under climate response uncertainty is modelled'

page 1, line 18: delete 'thereby'

page 1, line 18: 'determined effective' should match with whatever language is used to describe these ECS values in line 15

page 1, line 19: delete 'now'

page 1, line 24: 'emulating' → 'to emulate'

page 1, line 27: 'reduced time' → 'reduced amount of 'time'

page 1, line 28-29: "SCMs are indispensable as those IAMs' numerical solvers would call the climate module from ten thousands to hundred thousand times before numerical convergence is flagged." → "whose numerical solvers would call the climate module from ten to a hundred thousand times before numerical convergence is flagged, SCMs are indispensable."

page 2, line 6: 'capable of emulating the behavior of AOGCMs regarding GMT change,

deviations being a function of spread of forcing,' → 'capable of emulating the GMT change behavior of AOGCMs with observed deviations being a function of the spread of forcing,'

page 2, line 7: 'callibration' please provide reference

page 2, line 10: delete commas and 'utilized'

page 2, line 13: 'Here we address the most extreme opposite end of scale of complexity within the model category of SCMs' → 'Here we address the opposite end of the SCM scale'

page 2, line 14: delete 'Its role is as described in the following.'

page 2, line 15: add comma after 'diagnostic instrument'

page 2, line 17: comma after '(1999))'

page 2, line 27: 'further validation is both necessary and possible on a higher level of consistency' → 'further validation is necessary and results in a higher level of consistency'

page 2, line 29: 'scenario generation' → 'generating scenarios'

page 2, line 29: comma after '(UNFCCC, 2016)'

page 2, line 30: 'scenarios is crucial, displaying' → 'scenarios is crucial because they display'

page 2, line 32: 'we utilize' → 'we instead utilize'

page 2, line 33: 'Finally we find current practice, directly' → 'Finally we find that the current practice of directly'

page 2, line 33-34: 'and a second, time- scale relevant property for calibrating PH99,' → 'and a second, time- scale relevant, property for calibrating PH99'

page 3, line 1: 'in' → 'by' and 'on' → 'onto'

page 3, line 16: I don't understand what 'generically' means in this context, is it meant to be 'generally'?

page 3, line 26: 'Hereby' → 'where'

page 3, line 26: 'GMT' → 'GMT anomaly'

page 3, line 26: 'in units' → 'as a fraction' (?)

page 3, line 31: delete 'also in line with Petschel-Held et al. (1999) and Kriegler and Bruckner (2004).'

page 4, line 3: Would it be worth pointing out that for an increase rate, $r$, in %/yr, $\gamma = \ln(1 + \frac{r}{100})$?

page 4, line 4: add comma after 'In the following'

page 4, line 8: 'We address this difficulty by a chain of arguments along the equations as given below' → 'We address this difficulty now'

page 4, line 9: 'As starting point' → 'As a starting point,'

page 4, line 13: delete 'so derived'

page 4, line 13-28: I found this section very hard to understand, is it possible to simplify the language and process, perhaps?

1. convert equation 1 into an equation for heat flux by multiplying by $h$ so it reads

$$hTt = \mu h \ln(c) - \alpha h T(t)$$

2. recognise that $\mu h \ln(c)$ is radiative forcing due to $CO_2$

3. recognise that radiative forcing due to $CO_2$ is also given by $\frac{Q_2}{\ln 2}\ln(c)$ so

$$\mu h = \frac{Q_2}{\ln(2)} \Rightarrow \mu = \frac{Q_2}{h\ln(2)}$$

4. use equations 2, 3 and 6 with input ECS, TCR and $Q_2$ to determine $\alpha$, $\mu$ and $h$ and hence be able to time-integrate equation 5

page 4, line 27: 'Thereby' → 'Thus'

page 4, line 28: 'are projected' → 'can be projected'

page 4, line 29-: Sentence is too long: 'To derive the initial levels (2006) of the temperature anomaly with respect to the preindustrial value, for each AOGCM we calculate the mean temperatures over the period 1881-1910 and 1991-2020, respectively, as the preindustrial level of temperature and indicator for 2006 temperature level. The difference between these two is fixed as the initial temperature anomaly.' → 'Finally, to avoid differences occurring over the historical period (pre-2006 for the RCPs), we need to initialise PH99 with each AOGCM's 2006 temperature anomaly with respect to the pre-industrial value. To do this, for each AOGCM we calculate the mean temperature over the period 1881-1910 and set this as the pre-industrial value. We then calculate the mean temperature over the period 1991-2020 and use this as an indicator for the 2006 temperature level. The difference between these two values is fixed as the initial temperature anomaly for PH99.'?

page 5, line 3: 'As for' → 'For'

page 5, line 4: I can't find the Edenhofer et al. reference in the reference list and I don't think I'm being completely stupid...

page 5, line 5-6: 'is of special importance, as an error metric, the respective 2071-2100 GMT time averages of PH99 and AOGCM are subtracted.' → 'is of special importance.

Hence we use the difference between the respective 2071-2100 GMT time averages of PH99 and the AOGCM as an error metric.'

page 5, line 9: 'the deviations from the annual temperature data of' → 'the difference between PH99 and the AOGCM GMT anomaly for the'

page 5, line 10: 'are' → 'is'

page 5, line 11: '30-year means' → '2071-2100 means'

page 5, line 12: 'APGCM' → 'AOGCM' (this typo happens a few times, find and replace all 'APGCM' with 'AOGCM' should eradicate it)

page 5, line 22: delete 'needed to be run'

page 5, line 22: 'to check whether the trained climate module can' → 'of the trained climate module's ability to'

page 5, line 23: 'in PH99 the latter then being driven' → 'in PH99. PH99 is then'

page 5, line 24: 'final 30-year' → '2071-2100'

page 5, line 25: 'APGCM' → 'AOGCM'

page 5, line 27: 'time $1/\alpha$ calculated' → 'time, $1/\alpha$, calculated' or maybe 'time $(1/\alpha)$ calculated' to be consistent?

page 5, line 28: (being pedantic) units of 2.11 and 4.67

page 6, line 6: 'the' → 'from a'

page 6, line 6-8: "Paris 2015 agreement (UNFCCC, 2016) stated goal is '...holding the increase in the global average temperature to well below 2°C above pre-industrial levels and pursuing efforts to limit the temperature increase to 1.5°C above pre-industrial levels...'" → "2015 Paris agreement stated goal is '...holding the increase in the global average temperature to well below 2°C above pre-industrial levels and pursuing efforts to limit the temperature increase to 1.5°C above pre-industrial levels...' (UNFCCC,

2016)"

page 6, line 8: 'difference in 0.5K does matter' → 'difference of 0.5K matters'

page 6, line 10: 'deviations' → 'GMT deviations'

page 6, line 10: delete 'actual temperature'

page 6, line 11: delete 'thereby'

page 6, line 18: 'any' → 'each' and 'whereby' → 'where'

page 6, line 20: ', hence' → 'i.e.'

page 6, line 23: '4 A mapping of $\alpha$, $\mu$, and ECS onto their PH99-specific counterparts' → '4 A mapping of ECS onto its PH99-specific counterparts $\alpha$ and $\mu$'

page 7, line 3: 'on ECS' please provide reference

page 7, line 5: 'hereby' → 'here'

page 7, line 6: 'as of' → 'in'

page 7, line 7: 'by means of' → 'with'

page 7, line 8: 'as' → 'as a'

page 7, line 10: 'needed' → 'need about'

page 7, line 13: 'exploits' → 'exploits the fact'

page 7, line 21: 'extrapolate from the 14 utilized AOGCMs on any ECS' → 'extrapolate ECS and TCR from the 14 utilized AOGCMs'

page 7, line 21-22: Is line 22 meant to read 'Fig. 5'? as saying 'going beyond the scheme displayed' contradicts with figure 4 which actually does show all the schemes you discuss and the text on page 4, line 32 'are shown in Fig. 4'.

page 8, line 4: 'use of broader' → 'use of a broader'

page 8, line 5: 'use of' → 'using'

page 8, line 5: delete 'not only it projects a better approximation, but also'

page 8, line 6: 'on' → 'of'

page 8, line 13: 'This means, that, firstly' → 'For example'

page 8, line 14: 'on' → 'onto'

page 8, line 15: 'active' → 'forcing' or 'forcing active'

page 8, line 19: 'announced' → 'thought' or 'intended'

page 8, line 24: 'at' → 'to'

page 8, line 27-28: ', hence expresses' → '. It indicates that one is'

page 8, line 29: 'accepted above' → 'accepted the above'

page 8, line 29: 'boefore' → 'before'

page 8, line 30: 'on ECS' → 'of ECS'

page 8-9, line 30-1: 'When interpreting these values as PH99 values, as they have been in fact utilized in PH99 since Lorenz et al. (2012) for the MIND model,' → 'When interpreting these values as PH99 values (as they have in fact been utilized in PH99 since Lorenz et al. (2012) for the MIND model)'

page 9, line 2: 'From Fig. 7, from' → 'Fig.7 from' and '(Bindoff et al., 2013) we see' → '(Bindoff et al., 2013), shows'

page 9, line 21: 'Quite' → 'On'

page 9, line 27: 'Hereby' → 'Here'

page 9, line 29: 'those' → 'the'

page 9, line 30-32: Unclear. The only thing that made sense to me was to re-write as

'deviations from the RCP2.6 scenario are most relevant so we expect sufficiency for scenarios approximately in-line with the 2° target.'

page 9, line 32: 'approximantely' → 'approximately'

page 10, line 6: delete 'when'

page 10, line 7: 'AOGCMs to the emulator PH99, generally leads to an overestimation of global mean temperature (GMT) by 0.5 K. Quite' → 'AOGCMs to the emulator PH99 generally leads to an overestimation of global mean temperature anomaly (GMT) by 0.5 K. On'

page 10, line 8: 'by' → 'with'

page 10, line 10: 'AOGCM' → 'AOGCMs'

page 10, line 10: Is it worth pointing out that the 'several explanations' are all in the discussion as only one is presented in the conclusion?

page 10, line 12: 'as already' → 'as is already'

page 10, line 13: Add comma after 'However'

page 10, line 14: 'are re-interpreted as effective, 2°C-scenario-class specific values and mapped from original ECS and TCR values' → 'are mapped from original ECS and TCR values onto effective, 2°C-scenario-class specific values'

page 10, line 17: 'operated' → 'used'

page 10, line 17: 'For the' → 'Nonetheless, for the' may read better

page 10, line 19: 'It has to be checked to what extent the transformations on ECS and TCR' → 'check the extent to which the transformations of ECS and TCR'

I would suggest

page 10, line 20: '. (ii) By' → ' (ii) by'

page 10, line 22: '. (iii) F' → ' (iii) f'

page 10, line 23: '. (iv) A' → ' (iv) a'
* * *
page 10, line 23: 'mapping from ECS/TCR' → 'mapping from desired ECS/TCR values' or some other words to clarify?

page 10, line 28: 'parsimoneous' → 'parsimonious'

page 17, line 3: 'before' → 'at their RCP2.6 fitted values'

page 18, line 6: Delete 'Hence the validation is successful'. That's for the reader to decide.

page 19, line 1: 'over the last 30 years' → 'over the period 2071-2100'

page 19, line 1: 'vs.' → 'from'

page 21, line 2: 'discussion' → 'discussion of'

page 21, line 3: delete 'effectively lower'

---

## Author Comment (AC1) · 22 Jul 2017

Original comments by the referee are highlighted in italic font.

*The manuscript by Khabbazan and Held assesses the performance of a very simplified climate module currently in use in some IAMs. In particular, the study is motivated by the need to adjust the existing tools to the capability of this module in the light of assessments of below 2°C scenarios. To that end, it is fitted to different CMIP5 RCP2.6 AOGCMs.*

This is the very point of the first version of our ms indeed.

*The manuscript contains no fundamental flaws although a re-read is in order and the literature list should be checked.*

This will be done for the next version (that we call 'V2' for 'version 2' thereafter).

*One key paper (Foster et al. 2013) is for example missing from the literature list. I presume it's Forster, P. M., T. Andrews, P. Good, J. M. Gregory, L. S. Jackson, and M. Zelinka, 2013: Evaluating adjusted forcing and model spread for historical and future scenarios in the CMIP5 generation of climate models.*
*J. Geophys. Res. Atmos., 118, 1139– 1150*

The referee is right. This is the very reference that represents the data basis for our work. We sincerely apologize for this flaw. We will make sure that flaws like this one will not occur in V2.

[Figure]

**Reply-Figure 1: Intercomparison of global mean temperature from various AOGCMs and PH99 for RCP8.5. PH99 was fitted to RCP2.6.**

*More fundamentally, however, the scientific advancement presented of this study in my assessment is rather poor and it neglects important recent literature in this context (in fact, the literature list is rather short and at least 3 years old).*

V2 will state clearer what the point of our ms is. We recommend recalibrating PH99 before using it and thereby we also allow for a re-interpretation of existing literature. Furthermore, we will add a Section that demonstrates a mechanism that might be the key cause of the PH99-AOGCM discrepancy. We will show that already when moving from a 2-box model as used by the most influential work of W. Nordhaus in the integrated assessment literature (see e.g. Nordhaus and Sztorc) to PH99 the so far reported discrepancies can occur (for a mild detuning of Nordhaus' model). From this we will strive at explaining why a recalibration of PH99 is able to solve the reported problem and would do so not only for RCP2.6, but would then do so with identical calibration for *all* four RCPs, including RCP8.5 (see Reply-Fig. 1 for illustration). Thereby we will extend the scope of our article. Regarding omitted literature, to our impression the referee mainly refers to the effects outlined in the following paragraph. As outlined below the next paragraph, our ms is *not* about the issue raised by the referee in that paragraph.

*On the methodological approach: What's the justification of using the PH99 model (apart from it 'being there')? The authors argue that it's computational efficiency, …*

PH99 can be interpreted as an energy balance model (for details of the justification see Petschel-Held et al., 1999, and references therein; the authors' list contains Klaus Hasselmann! Furthermore, the model was validated in Kriegler and Bruckner, 2004 – however the validation was done in a different way and also did not have the forcing reconstructions by Forster et al., 2013, at hand). Even today, computational efficiency is key, e.g. when it comes to decision-making under endogenous learning (see e.g. Webster et al., 2012).

[Figure]

**Reply-Figure 2: Start of an intercomparison of global mean temperature from various AOGCMs and PH99 for RCP6.0. PH99 was fitted to RCP2.6.**

*…but how exactly are they convinced that their treatment of non-CO2 GHGs is appropriate. For strong mitigation pathways, these 'minor' differences may become very important, last but not least to determine net-zero global GHG forcing etc. I'd think they would need validate their fit using other strong mitigation scenarios with different non-GHG trajectories (if no others are available then from the GeoMIP experiment) rather than RCP4.5. In particular, it appears that non-CO2 gases obscure our assessments of ECS (see e.g. Myre et al. 2016)*
*As it stands, I'm not convinced that the simplified model is capable of including non- GHG forcing in a sufficient fashion for the question at hand (i.e. staying below $2°C$ or $1.5°C$).*

Apparently here we provoked a key misunderstanding what our article is about. It is *not* about how to generate a global total forcing out of regional forcings. Our ms is about how to get from a global total forcing to global mean temperature and whether PH99 does a good enough job in that, i.e. could substitute for AOGCMs or more complex integrated assessment climate modules in that regard. We will make this point very clear in V2. We nevertheless started assembling validation data on RCP6.0 which contains a strong ozone component. This will be completed in V2. Reply-Fig. 2 shows the success of PH99 also in this case. We are not aware of other scenarios rather than the RCP2.6 for which the total forcings had been reconstructed.
Furthermore, we are grateful for the hint that non-CO2 gases might obscure an assessment of ECS. This provides an alternative explanation for the discrepancies

that we report prior re-calibration of PH99. While above we report a failure of PH99 to reproduce certain features of even as simple model as a 2-box model, in fact ECS of various AOGCMs might come with some reconstruction errors. So part of the problem might lie outside the 'responsibility of PH99'.

Finally, any author within integrated assessment of the coupled climate-economy-problem has to deal with how to construct a meaningful global forcing out of regionally disaggregated AOGCM forcings, as AOGCMs cannot directly be utilized in economic optimizations. This is a difficult discussion indeed, but *any* climate module within integrated assessment would face this issue, not only PH99. This discussion, however, is not the subject of our ms.

*On the application: I didn't fully the motivation for step 1. What was the reasoning for the authors to assume that their PH99 model would work with AOGCM diagnostics from Forsters et al. directly? Obviously, the derived feedback response time parameter 1/alpha of 34.5 years in the multi-model mean is quite unphysical. It seems that the PH99 model is not equipped to be used in that context.*

This is the very point. We want to highlight that *current practice* in integrated assessment of directly prescribing ECS and other parameters from AOGCMs leads to biases. As an AOGCM also contains time scales faster than 35 years, it is not immediately clear that an average time scale of smaller and larger time scales would

be meaningless.

*In a next step, the authors find that with two free parameters they are capable of achieving better fits. That's not particularly surprising, but a physical interpretation of these differences is virtually absent? ECS is substantial decreased by almost 1°C. Can this be understood? The authors continue with fitting derived and fitted ECS and TCR, but I would rather like to see a physical interpretation or an extension of the PH99 model that would correct for this. The authors should also consider their approach in the light of alternative simplified approaches out there i.e. based on a response function approach as in Ragone et al. (2016).*

V2 will deliver on this. By having found that the discrepancy we report can be explained by the move from a 2-box to a 1-box model, we have generated an anchor for explaining the reported ECS effect. A key problem of the 1-box model is that it replaces the slow-component response by the averaged faster one and hence would lead to an overreaction for peak-and-decline forcing scenarios such as RCP2.6. In that sense we expect RCP2.6 as particularly difficult to emulate. Kriegler and Bruckner, 2004, could not do this validation, as mitigation scenario forcing reconstructions were not available at that time.

[Figure]

**Reply-Figure 3: PH99 parameters vs. ECS.**

*The authors then want to apply an effective correction for their dubious model in the first place. Their results here appear to be prone to outliers. Compare e.g. the low ECS outlier in Fig. 5. When removing it, I guess even a linear fit would deliver decent results and I'm not sure I can deduce any robust trends from these graphs…*

V2 will avoid the impression that we advertised utilizing PH99. We simply want to state how to interpret older work based on PH99, and how it could be used if someone wants to use it in the future. In addition to computational efficiency, for some applications also analytic tractability or conceptual simplicity might be

arguments for using PH99.

Furthermore we have done the sensitivity study the referee suggested by repeating some analyses w/o outliers. W/o outliers linear fits would make PH99 parameters predictable by ECS indeed (see Reply-Fig. 3). The scenario fit results deliver similar quality as before (see Reply-Fig. 4). However we would like to stress that our ms recommends direct correction of ECS along ms-Fig.6 where outliers play a less prominent role. V2 will have a discussion of the outlier issue.

[Figure]

**Reply-Figure 4: The quality of fit does not change by avoiding outliers and using a linear fit instead.**

*On the application of this. It seems that the model that is used in these IAMs has many flaws. The question then becomes why it is used at all? And not abandoned for a carbon budget approach that would be even more computational effective and can be determined with more complex models also for these low emissions scenarios (i.e. Rogelj 2016). That becomes in particular relevant since the mitigation challenge ahead is to define pathways that hold warming 'well below 2°C'. "Below 2°C" was interpreted as a 66% chance of non-exceedance (IPCC 2014). What's the added value of using a PH99 model in this context? Would they select an ECS at the 66% quantile and then use this as a basis for the IAM derivations? And if so, why not use carbon budgets directly?*

We are grateful for this exciting suggestion. It is a fascinating question indeed when a dynamic climate module could simply be replaced by the carbon budget approach to deliver similar – or even better quality – in emulation of an AOGCM. This discussion is beyond the scope of this ms, however we would highlight this option in V2. Here we simply would like to stress that there *are* applications in climate economics where timing matters, such as cost benefit analyses (see e.g. Nordhaus and Sztorc, 2013) or cost risk analyses (see e.g. Neubersch et al., 2014).

In summary we are optimistic that a new version V2 could be acceptable for the reviewer if (i) it clarified its scope: being about the link from total forcing to temperature and (ii) if it delivered a physical interpretation of the observed effects. We are grateful for the referee's comments as they will have triggered an – in our view – considerably upgraded version of our ms.

**References Review 1**

Forster, P. M., Andrews, T., Good, P., Gregory, J. M., Jackson, L. S., & Zelinka, M. Evaluating adjusted forcing and model spread for historical and future scenarios in the CMIP5 generation of climate models. Journal of Geophysical Research: Atmospheres, 118(3), 1139-1150, 2013.

Kriegler, E. and Bruckner, T.: Sensitivity analysis of emissions corridors for the 21st century, Climatic Change, 66, 345–387, 2004.

Neubersch, D., Held, H., & Otto, A. Operationalizing climate targets under learning: An application of cost-risk analysis. Climatic change, 126(3-4), 305-318, 2014.

Nordhaus, W., Sztorc, P., DICE 2013R: Introduction and User's Manual, dicemodel.net , 2013.

Petschel-Held, G., Schellnhuber, H.-J., Bruckner, T., Toth, F. L., and Hasselmann, K.: The tolerable windows approach: Theoretical and methodological foundations, Climatic Change, 41, 303–331, 1999.

Webster, Mort, Nidhi Santen, and Panos Parpas. An approximate dynamic programming framework for modeling global climate policy under decision-dependent uncertainty. Computational Management Science 9.3: 339-362, 2012.

---

## Author Comment (AC2) · 22 Jul 2017

First of all, we would like to thank the referee for an exceptionally thorough and thoughtful review!
Original comments by the referee will be highlighted in italic font below.

**1  General Comments**

*Khabbazan and Held's paper checks the performance of a one box energy balance model (PH99), currently in use in the integrated assessment models FUND and MIND, against output from AOGCMs before suggesting a simple, improved*

*way to use it in future. Their major conclusion is that, for strong mitigation scenarios, prescribing ECS and TCR to PH99 from Forster et al. (2013) with no further calibration implicitly causes researchers to sample much larger temperature responses than they intend to. They show that a simple fitting exercise rectifies this and validate the fit by checking PH99's performance under one other scenario. This scenario is very similar to the one they used for fitting. They then explore different methods to map AOGCM ECS and TCR onto 'effective' PH99 values which could provide researchers with a simple method of revealing the temperature response they are actually considering.*

We feel our ms perfectly perceived by the referee.

*My major concerns focus on whether the analysis shows that PH99 is a valid energy balance model rather than a fitting tool.*

That is a difficult question, an almost philosophical one. If we were invited on a new version (termed 'V2' thereafter) we would present a physical mechanism that represents one option to explain the discrepancies produced by PH99. We can show that the discrepancies already occur when replacing a 2-box model like the one implemented in DICE (Nordhaus and Sztorc, 2013) by PH99. So if one had the request PH99 should be able to work with AOGCM parameters, then PH99 would

no longer qualify as an energy balance model. On the other hand, by interpreting the observed effects one could still give it some physical meaning.

Also, V2 will avoid the impression that we advertised utilizing PH99. We simply want to state how to interpret older work based on PH99, and how it could be used if someone wants to use it in the future. In addition to computational efficiency, for some applications also analytic tractability or conceptual simplicity might be arguments for using PH99.

*I also think that the writing style could be greatly improved.*

V2 will be improved in that regard.

[Figure]

**Reply-Figure 1: Intercomparison of global mean temperature from various AOGCMs and PH99 for RCP8.5. PH99 was fitted to RCP2.6.**

*I think the authors point out some key errors which arise if PH99 is used without care and explore a few ways for modellers to quickly relate their parameters to AOGCM ECS and TCR values. However, given that the authors argue for mapping*

*AOGCM properties onto 'scenario-class-specific values before using them in PH99', which appears to undermine any physical basis for PH99, I'm left wondering if this paper highlights the limitations of PH99 rather than providing strong arguments for its use.*

In fact we expand on the scope of utilizing PH99. We can show that after proper recalibration on AOGCM-RCP2.6, that calibration can be utilized for all RCPs – see Reply-Figure 1 on RCP 8.5 for illustration.

**2   Major concerns**

1. *The re-callibration of PH99 is only validated for RCP4.5. There is no other testing of the performance outside of RCP2.6 and RCP4.5, two very similar scenarios, nor testing of the effect of different non-CO2 forcing pathways. Thus the authors have shown that a good fit to AOGCM GMT output can be done with two free parameters and that this fit is good for a similar scenario. I wonder if testing over a greater range of other scenarios would strengthen the justification for the use of PH99.*

We will show in V2 that PH99 emulates *all* RCPs by the identical recalibration. This also contains RCP6.0 with some stronger ozone component. Furthermore, this ms is

*not* about how to generate a meaningful global forcing from non-CO2 agents, but how to prognose global mean temperature from total global forcing. V2 will be much clearer about this scope. The effects by moving from a 2-box to a 1-box model will play a prominent role in V2. How to get to a total forcing, however, is a discussion that hits *any* climate module utilized in integrated assessment, and is beyond the scope of this ms.

2. *The initial testing of the performance of PH99 against AOGCMs reveals a key, hidden, bias of this model if used without validation in strong mitigation scenarios. This is a good bit of analysis. As a result of this analysis, the authors advocate mapping AOGCM climate system properties onto 'scenario-class-specific values before using them in PH99'. Whilst this seems to be necessary for acceptable performance of PH99, it also appears to undermine any physical basis for PH99. If you have to re-callibrate PH99 every time you want to use it in a different scenario class then its parameters lose all physical meaning and instead simply become fitting parameters. Thus the authors appear to advocate shifting PH99 from an energy balance model to a function that can be fitted to AOGCM data and then used for a limited range of scenarios?*

We now know that we undersold PH99 in that regard. This underselling will be stopped in V2. Instead the underlying physical mechanisms will be explained from which also the observed discrepancy will be derived.

3. *I don't think I am wrong in saying that this model is ultimately meant to be used by those who are looking for simple emulators of global mean temperature response and hence may not be climate modellers themselves. If this target audience can't pick up this paper and get some sense of what is going on then they will struggle to use any of the fits provided. A paper on 'the most parsimonious climate module' should have a style which reflects its title. Given that parsimonious is synonymous with 'simple' in this context, it makes sense for the communication to be as plain, clear and simple as possible too. With this goal in mind, I suggest numerous technical corrections and ask for multiple clarifications.*

We fully agree and are extremely open to the detailed changes the reviewer detailed below.

4. *The exploration of different possible parameterisations of the relationship between AOGCM ECS/TCR and effective ECS/TCR is, in my opinion, worthwhile. My impression is that they recognise that a parameterisation would be nice but don't have strong enough evidence to recommend any of the ones they have tried and so the results here are underwhelming.*

V2 would strive at discussing those fitting procedures with the new physical

interpretation at hand.

**3   Specific Comments**

1. *As an exercise, the fitting that is done is scientifically sound re methods, assumptions, results, and reproducibility as far as I can tell. I can also see that it would be useful for modellers who wish to use a simple emulator but don't wish to do the calibration themselves.*

Thank you!

2. *I think this paper shows that PH99 is closer to a fitting tool rather than a physical model. Hence I wonder, if modellers are after computational simplicity and a fitting tool, why wouldn't they use a simple carbon budget target or emissions pathway to constrain their model. There is already research on how emissions pathways and targets map to temperature targets so this could be used to back out emissions constraints from a given temperature target for a given scenario class. This approach seems far simpler than introducing an energy balance module which requires atmospheric concentration and radiative forcing input, has little physical basis and hasn't been validated over a wide range of CO2 and non-CO2 scenarios so might not produce realistic temperature*

*projections anyway.*

We are grateful for this exciting suggestion. It is a fascinating question indeed when a dynamic climate module could simply be replaced by the carbon budget approach to deliver similar – or even better quality – in emulation of an AOGCM. This discussion is beyond the scope of this ms, however we would highlight this option in V2. Here we simply would like to stress that there are applications in climate economics where timing matters, such as cost benefit analyses (see e.g. Nordhaus and Sztorc, 2013) or cost risk analyses (see e.g. Neubersch et al., 2014).

3. *The introduction calls IAMs an 'indispensable tool'. I acknowledge that this comment is made in the context of 'driving welfare-optimal climate policy scenarios' so it is accurate. However given that there are many who disagree with using economic analyses for determining 'welfare-optimal scenarios' because of the need to monetise many things which arguably can't be monetised (e.g. the environment), using this term seems to open the paper up to unwanted distractions. I think this could be avoided with a simple re-wording; calling IAMs a 'tool which are used to derive welfare-optimal scenarios' rather than an 'indispensable tool used to derive welfare-optimal scenarios'. This change would avoid opening up an economic debate (in the reader's mind) which is completely outside the scope of this paper.*

For V2 we will comply with the reviewer's suggestion.

4. *page 8, line 22: 'personal conviction'. I don't think personal convictions have any place in scientific papers. Either the evidence is there to support using log-normal distributions or it's not. I also don't understand what the sentence beginning with 'This conviction rests' means. Does it mean 'Schneider von Deimlinig et al. show that constraining ECS by paleo data results in thin-tailed distributions'? If yes, then there is no need for a 'personal conviction', circling back to my first point.*

We will eliminate this paleo discussion from V2. For V2 it is sufficient to say that lognormal distributions for ECS are used in the literature and that under an affine transformation (as suggested in ms-Fig.6) they would be mapped onto lognormal distributions the quantiles of which are still compatible with what is reported in IPCC AR5 WGI on ECS.

Yes, the point was that Schneider von Deimling et al., 2006, would allow for a thin tail on ECS. The term 'conviction' might not have been the accurate term for what we tried to express. Our point was that a single paper does not yet trigger a paradigm shift. Major fractions, if not the majority of climate researchers – in contrast to us – doubt that ECS can be constrained by paleo data. This fundamental question is still open and hence also subject to personal believe and intuition that might guide the choice of the always in-part

subjective distributions of ECS (as any of them rests on Bayesian learning that needs a subjective prior as an input).

5. *I really appreciated the discussion of low pass filtering and think this was well done.*

Thank you.

[Figure]

**Reply-Figure 2: The quality of fit does not change by avoiding outliers and using a linear fit instead.**

*6. In section 4 (page 7, lines 32-34), the authors state that 'regressing both inferred effective ECS and TCR solely against AOGCMs' ECS obviously is the overall better approximation'. Whilst this is borne out by taking a pure average of all the results, there are clearly two strong outliers which are having a major effect on the performance of the ECS-ECS & TCR-TCR mapping. I wonder what is causing such large outliers (they seem hugely anomalous) and if removing them would be justified. If they are removed, how much does this change the conclusions.*

We will have an extended discussion on that issue in V2. First results on the PH99 parameter-based fit method (see ms-Fig. 5) show that eliminating outliers and then moving to a linear fit would not significantly change the quality of emulation (see Reply-Fig. 2).

**4   Technical Corrections**

The innumerous suggestions made by the referee appear very meaningful and appropriate to us and will be implemented in V2.

In summary we are optimistic that a new version V2 could be acceptable for the

reviewer if (i) it clarified its scope: being about the link from total forcing to temperature, (ii) if it delivered a physical interpretation of the observed effects, (iii) reflected all of the technical corrections. We are grateful for the referee's comments as they will have triggered an – in our view – considerably upgraded version of our ms.

**References Review 2**

Neubersch, D., Held, H., & Otto, A. (2014). Operationalizing climate targets under learning: An application of cost-risk analysis. Climatic change, 126(3-4), 305-318.

Nordhaus, W., Sztorc, P., DICE 2013R: Introduction and User's Manual, dicemodel.net , 2013.

Petschel-Held, G., Schellnhuber, H.-J., Bruckner, T., Toth, F. L., and Hasselmann, K.: The tolerable windows approach: Theoretical and methodological foundations, Climatic Change, 41, 303–331, 1999.

Schneider von Deimling, T., Held, H., Ganopolski, A., and Rahmstorf, S.: Climate sensitivity estimated from ensemble simulations of glacial climate, Clim Dyn, 27, 149–163, doi:10.1007/s00382-006-0126-8, 2006.

---

## Author Comment (AC3) · 4 Aug 2017

Dear Editor,

We have carefully studied the two referee reports on our ms "On the Future Role of the most Parsimonious Climate Module in Integrated Assessment". We found these reports extremely helpful as they triggered further analyses of ours that will lead to a considerable upgrade of the ms in case we were invited on a new version. This new version would differ from the previous one in four major aspects:

1) We apparently undersold the climate module PH99 that is the subject of our analysis.

[Figure]

Once re-calibrated in the way we suggest, PH99 can emulate an AOGCM for any of the four RCPs, sticking to that very re-calibration. Hence PH99 is more physical than expected.

2) We explain the discrepancy observed by a physical mechanism, as we can reproduce that very discrepancy by the move from a 2-box model to a 1-box model.

3) The first, very critical reviewers misperceived the scope of our ms. It is not about how to aggregate spatially inhomogeneous forcings into a global forcing, but it is rather how to get from a global total forcing to global mean temperature.

4) The new version would be prepared much more carefully on the technical level, absorbing the innumerous technical comments by referee #2.

In total, we are the more convinced that our ms is a timely addition to the literature on how to represent a complex system like the climate system by a simplest-possible approach. We would be delighted if the Editor gave us the chance to present our arguments in a new version.

Sincerely,

Khabbazan and Held

---

## Short Comment (SC1) · 23 Sep 2017

We have received an appeal from the authors regarding the editorial decision to reject the manuscript.

This appeal was based on two aspects: First, the authors argue that reviewer #1 essentially wanted a different study even though the manuscript contains no fundamental flaws, and second, the editor did not seem to consider the reply of the authors in the description of the editorial decision.

The first issue could have been clarified by being more active during the discussion

phase, and, possibly, by a clearer description of the goals of the study, while the second issue could have been avoided by a more extensive explanation of the editorial decision.

After consulting the authors and the editor involved, we decided to re-open the discussion and obtain a third review for the assessment of this manuscript.

Axel Kleidon on behalf of the ESD Chief Editors

---

## Referee Comment (RC3) · Anonymous Referee #3 · 27 Nov 2017

**General comments**

This manuscript investigates the performance of a one-box energy balance model (PH99) as an AOGCM emulator for strong mitigation scenarios. The authors find that this simple climate model (SCM) consistently over-predicts future temperatures when the ECS and TCR are transferred directly from AOGCMs. Fitting the PH99 directly to the AOGCM temperature time series eliminates this bias, and reveals that the AOGCMs time series imply a substantially lower ECS and higher TCR than what they had transferred directly. The manuscript briefly discusses the physical interpretation of this discrepancy, and also explore alternative ways of fitting the one-box model that

might be more reasonable for extrapolation in parameter space (of the kind performed when these SCMs are used to investigate the optimal dynamic behaviour of a decision maker under uncertainty).

Before continuing further, I want to briefly highlight two important factors that might reasonably affect how you read this review. First, I have not only read the manuscript, but also the previous reviews and the responses from the authors. My comments primarily address the manuscript itself, but I will also sometimes explicitly agree or disagree with comments that have been made earlier in the process. Second, I have not approached this manuscript as a climate physicist, but rather from the perspective of a researcher who uses the integrated assessment models with SCMs like PH99. My comments will therefore differ in spirit from those of earlier reviewers, and I focus more on issues I believe to be more relevant to those who would use this research.

**Specific comments**

My overall assessment is that this manuscript offers an interesting contribution and should be published. A previous reviewer expressed concern that the scientific contribution may be inadequate, but I feel that this comment does not adequately consider the policy influence that the one-box model wields (or rather, simple integrated assessment models that use PH99 in one form or another). For instance, two of the three climate-economy models used by the US federal government to calculate the social cost of carbon incorporate one-box energy balance models (notwithstanding recent political developments). The policy analysis in the *Stern Review*, which was commissioned and used by the UK government to formulate climate policy, was also based on a coupled climate-economy model that incorporated a one-box energy balance model. By their simplicity, these SCMs are also have come to serve as tools for translating new climate science for communities that use climate information but generally lack

extensive physics training. Even a relatively small improvement in our understanding and handling of these models would provide a significant contribution.

I do have some concerns about the manuscript, though. First, I think there are parts of the manuscript that will be difficult to decypher for many of the researchers that actually work with SCMs in the context of simple climate-economy models. Second, I think that authors have tended to focus excess attention on concerns related to interpolation and extrapolation of parameter values, at the expense of a fuller and clearer discussion of the physical interpretation of their primary findings. I discuss each point in turn, and offer a few minor comments at the end. Let me state clearly, though, that I expect these concerns can be fully redressed, so I wouldn't consider them reasons for rejecting the manuscript.

1. The heart of this manuscript, as I see it, is the direct transfer of AOGCM characteristics (section 2.1). The central issue is whether or not it is appropriate to use this physical method for deducing the parameter values in one-box model. The subsequent question about whether alternative methods for fitting the parameter values perform better, is also tied to this baseline method. So section 2.1 is really the foundation for all of the analysis in this paper. Yet two cruicial pieces seem to be missing from it.

   First, at this point in the manuscript the authors should be offering a childishly clear explanation of how (and which) AOGCM outputs can be used to deduce the values of $\alpha$ and $\mu$, which can then be plugged into equations (2) and (3) to retreive the implicit ECS and TCR, respectively. But I must admit to having some difficulty following their derivations (e.g. not understanding how $h$ is determined in equation (7) where both $h$ and $\mu$ appear to be unknowns, and not seeing any expression for $\alpha$ in terms of AOGCM output). A climate physicist will perhaps be so familiar with this material as to be able to perform these calculations with little prompting from the authors, but as a presumptive member of the intended

audience, I would appreciate it if the authors exercised greater pedagogical care in this section.

Second, and just as important, is that the authors have not offered any information to suggest that this is how modellers are currently choosing values for the ECS and TCR. In my experience, many users will not themselves try to deduce these parameters from AOGCM outputs, but rather plug in values of ECS and TCR reported in IPCC chapters or specific academic papers without fully understanding how these values are inferred from AOGCM simulations (and sometimes adding a bit of 'calibration' to make sure the results don't look too dissimilar from MAGICC, say). If those reported ECS and TCR values are derived in this way generally, the authors should state this clearly and cite examples. If not, they should consider whether it would be more appropriate to use a different baseline.

As a suggestion, I think it would be worthwhile to run the model using the actual parameter values assumed in FUND and MIND as a baseline (and maybe PAGE, which the authors seem to have ignored, even though it incorporates a one-box model). If I were to speculate, I would guess that using the default parameter values from these models will give even more discrepant predictions, so in addition to being more relevant to current practice, it might illustrate your point even more clearly.

2. SCMs are used for two distinct purposes: (1) as devices for summarizing and communicating climate science to other modelling communities, and (2) as computationally efficient AOGCM emulators. The analysis performed in this manuscript has important implications for both uses, but the authors are failing to distinguish clearly between them. This creates unnecessary confusion (seen especially clearly in the exchanges with previous reviewers), and has in my opinion led to an unbalanced treatment.

The authors appear to recognise the role of PH99 as a communication device when they, in their Discussion (section 5), briefly mention the idea that the 'tran-

sient climate sensitivity' might be lower than the ECS, as a potential physical explanation for the lowering of the ECS when the parameters are calculated by fitting the one-box model to AOGCM temperatures instead of derived from AOGCM forcings. But what is the chief physical mechanism behind this? And does this mean that PH99 users should interpret $\frac{\mu}{\alpha}\ln(2)$ as the 'transient climate sensitivity' rather than the ECS? Can we do this without undermining the physical basis for the one-box model? And what about the higher TCR value that you get when fitting the one-box parameters instead of transfering them directly? What is the physical interpretation of this second important change? You also acknowledge that measurement error in AOGCM outputs could lead to biased values of the ECS and TCR in the one-box model, but can you do anything to show that the biases would actually go in the direction of inflating the ECS and deflating the TCR? Perhaps you could just add a random sample of Gaussian deviations to your input data and feed them through the non-linear PH99 mapping to see what the resulting distribution of ECS and TCR would look like?

I realise I have given you a lot of questions to answer, but I really do feel that this part of the discussion has been unduly neglected, and the paper would benefit greatly from extending it. It seems a very interesting fact that, for a given TCR, the ECS value transferred directly from an AOGCM is systematically higher than the value that would yield the best fit to that same AOGCM (and vice versa for TCR). Anything the authors are able to do to help the reader understand the causes of the differences between fitted and transferred parameter values, and how this might affect the physical interpretation of the one-box model paramters, would be very welcome.

I would, compensatingly, recommend shortening the discussion of the second use of PH99, as an AOGCM emulator, which currently takes up the majority of sections 4 and 5. I think it is interesting to consider the advantages and disadvantages of alternative methods for choosing ECS and TCR for emulation purposes,

but I often felt lost in this discussion and think it can be done more concisely. The fitting method, perhaps unsurprisingly, does a pretty good job of fitting the AOGCM temperature time series. But the key drawback of the fitting-method is that it's inappropriate for obtaining probability distributions for the ECS and TCR that can be used to simulate PH99 under uncertainty. These kinds of simulations are now standard practice for economic assessments of climate policy based on coupled climate-economy models, so this is indeed an important issue to wrestle with.

The quadratic and cubic fitting in Figure 5 seems useful mostly as a cautionary example of 'what not to do.' The authors already explain that it's likely to lead to unphysical parameter values, and as a previous reviewer pointed out, the curvature seems largely a consequence of a single AOGCM run with a low ECS. Overall, the authors can probably devote less space on this particular exercise and be even clearer that it is ill-advised. Instead, they should focus on the more physical interpolation/extrapolation methods considered in section 4, and try to offer users more concrete advice about when they might prefer the Lorenz curve method, or the ECS-to-ECS and TCR-to-TCR fit, or the ECS/TCR-to-ECS fit, or when all three are likely to perform poorly.

I think a slight reorganization of sections 4 and 5 would probably be the most effective way of accomplishing all of this. The new section 4 would take the first two paragraphs of the current section 5 as its starting point, but elaborate along the lines I have discussed above in order to offer a discussion of the physical interpretation of fitted ECS and TCR relative to the directly transferred ones. The new section 5 would merge the current section 4 with the remainder of the current section 5, in order to offer a discussion of the appropriate and inappropriate ways to interpolate and extrapolate ECS and TCR values in PH99, in light of their physical reinterpretation in the new section 4. This separation would also make it much clearer how the choice of parameter values for PH99 depends on whether

one is using it as a communications device or as an emulator.

Technical corrections

1. p. 1, line 9 (and throughout): The manuscript refers to FUND and MIND as two coupled climate-economy model that employ a one-box model. PAGE does as well (see discussion in Calel & Stainforth, 2017, BAMS, already cited).

2. p. 2, line 30: Typically, these models are used to study optimal climate policy, so it would be good if you could cite a few studies where these models are used specifically to study 2 degree stabilisation scenarios.

3. p. 5, line 12: Typo. "APGCM" should be "AOGCM."

4. p. 5, lines 17-22: While RCP4.5 is certaintly out-of-sample, it's less obvious to me that it serves the purpose of validating the method for 2 degree stabilisation scenarios. As a validation exercise, wouldn't it be preferable to fit $\alpha$ and $\mu$ using RCP4.5 and then drive the one-box model using the RCP2.6 forcings?

5. p. 8, line 11: Typo. "againsta" should be "against a."

6. p. 8, line 15: Typo. "radiative active" should be "radiative activity" or "radiative forcing."

7. p. 8, line 19: I think it's inaccurate to say that "studies based on PH99 implicitly worked with ECS values that were larger than announced." I think you've made the point that they might be using a higher ECS than would be appropriate for emulating AOGCMs, but this is quite different. They declare their ECS values, and the question raised in this manuscript is whether they shouldn't be using ECS but rather some 'effective ECS' or 'transient climate sensitivity' instead. Please rephrase this.
8. p. 8, lines 20-28: The discussion of log-Normal distributions seems to come out of nowhere. I think the reorganization I have suggested above may resolve this, but please make an effort to link this more strongly to the rest of the discussion.

9. p. 8, line 29: Typo. "boefore" should be "before."

10. p. 10, line 7: Typo. "generally" should be "it generally."

---

## Author Comment (AC4) · 24 Dec 2017

First of all, we would like to thank the referee for an exceptionally thorough and thoughtful review!
Original comments by the referee will be highlighted in italic font below.

*General Comments*

*This manuscript investigates the performance of a one-box energy balance model (PH99) as an AOGCM emulator for strong mitigation scenarios. The authors find that this simple climate model (SCM) consistently over-predicts future temperatures*

*when the ECS and TCR are transferred directly from AOGCMs. Fitting the PH99 directly to the AOGCM temperature time series eliminates this bias, and reveals that the AOGCMs time series imply a substantially lower ECS and higher TCR than what they had transferred directly. The manuscript briefly discusses the physical interpretation of this discrepancy, and also explore alternative ways of fitting the one-box model that might be more reasonable for extrapolation in parameter space (of the kind performed when these SCMs are used to investigate the optimal dynamic behaviour of a decision maker under uncertainty).*

We feel our ms perfectly perceived by the referee.

*Before continuing further, I want to briefly highlight two important factors that might reasonably affect how you read this review. First, I have not only read the manuscript, but also the previous reviews and the responses from the authors. My comments primarily address the manuscript itself, but I will also sometimes explicitly agree or disagree with comments that have been made earlier in the process. Second, I have not approached this manuscript as a climate physicist, but rather from the perspective of a researcher who uses the integrated assessment models with SCMs like PH99. My comments will therefore differ in spirit from those of earlier reviewers, and I focus more on issues I believe to be more relevant to those who would use this research.*

While we appreciate comments from all relevant disciplines, we are happy that among the referees there is a referee who uses integrated assessment models.

***Specific comments***

*My overall assessment is that this manuscript offers an interesting contribution and should be published.*

Thank you!

*A previous reviewer expressed concern that the scientific contribution may be inadequate, but I feel that this comment does not adequately consider the policy influence that the one-box model wields (or rather, simple integrated assessment models that use PH99 in one form or another). For instance, two of the three climate-economy models used by the US federal government to calculate the social cost of carbon incorporate one-box energy balance models (notwithstanding recent political developments). The policy analysis in the Stern Review, which was commissioned and used by the UK government to formulate climate policy, was also based on a coupled climate-economy model that incorporated a one-box energy balance model. By their simplicity, these SCMs are also have come to serve as tools for translating new climate science for communities that use climate*

*information but generally lack extensive physics training. Even a relatively small improvement in our understanding and handling of these models would provide a significant contribution.*

We feel our ms perfectly perceived by the referee.

*I do have some concerns about the manuscript, though. First, I think there are parts of the manuscript that will be difficult to decypher for many of the researchers that actually work with SCMs in the context of simple climate-economy models. Second, I think that authors have tended to focus excess attention on concerns related to interpolation and extrapolation of parameter values, at the expense of a fuller and clearer discussion of the physical interpretation of their primary findings. I discuss each point in turn, and offer a few minor comments at the end. Let me state clearly, though, that I expect these concerns can be fully redressed, so I wouldn't consider them reasons for rejecting the manuscript.*

We are thankful to the reviewer for the time and efforts. We are open to any comments that can enhance the manuscript.

1. *The heart of this manuscript, as I see it, is the direct transfer of AOGCM characteristics (section 2.1). The central issue is whether or not it is appropriate to use this physical method for deducing the parameter values in one-box model.*

*The subsequent question about whether alternative methods for fitting the parameter values perform better, is also tied to this baseline method. So section 2.1 is really the foundation for all of the analysis in this paper. Yet two cruicial pieces seem to be missing from it.*

*First, at this point in the manuscript the authors should be offering a childishly clear explanation of how (and which) AOGCM outputs can be used to deduce the values of $\alpha$ and $\mu$, which can then be plugged into equations (2) and (3) to retreive the implicit ECS and TCR, respectively. But I must admit to having some difficulty following their derivations (e.g. not understanding how h is determined in equation (7) where both h and $\mu$ appear to be unknowns, and not seeing any expression for $\alpha$ in terms of AOGCM output). A climate physicist will perhaps be so familiar with this material as to be able to perform these calculations with little prompting from the authors, but as a presumptive member of the intended audience, I would appreciate it if the authors exercised greater pedagogical care in this section.*

We will make V2 clearer regarding the derivatives and present an example for deducing the values of α and $\mu$ to be plugged into equations (2) and (3) to retrieve the implicit ECS and TCR.

*Second, and just as important, is that the authors have not offered any information to suggest that this is how modellers are currently choosing values*

*for the ECS and TCR. In my experience, many users will not themselves try to deduce these parameters from AOGCM outputs, but rather plug in values of ECS and TCR reported in IPCC chapters or specific academic papers without fully understanding how these values are inferred from AOGCM simulations (and sometimes adding a bit of 'calibration' to make sure the results don't look too dissimilar from MAGICC, say). If those reported ECS and TCR values are derived in this way generally, the authors should state this clearly and cite examples. If not, they should consider whether it would be more appropriate to use a different baseline.*

We thank the referee for pointing us to another misunderstanding we might have provoked. In fact, the referee's perception of standard practice is actually what we wanted to refer to in our manuscript, as we believe that users of PH99 have not themselves tried to deduce these parameters from AOGCM outputs. However, given that insight, then comparing AOGCM and PH99 output while both models would be characterized by identical ECS and TCR and be driven by identical radiative forcing is then the most direct way we can currently imagine to demonstrate a bias encoded in PH99. We will elaborate on a clarification of this argument in V2.

*As a suggestion, I think it would be worthwhile to run the model using the actual parameter values assumed in FUND and MIND as a baseline (and maybe PAGE, which the authors seem to have ignored, even though it incorporates a one-box*

*model). If I were to speculate, I would guess that using the default parameter values from these models will give even more discrepant predictions, so in addition to being more relevant to current practice, it might illustrate your point even more clearly.*

This is a nice suggestion. We have already addressed this for MIND under Lorenz' curve which correlates $\alpha$ and $\mu$ to ECS. In V2, we will make this issue clearer and check FUND and PAGE. (So far we had avoided a discussion of PAGE as the link of parameters is not as direct, but we will close this gap in V2.)

2. *SCMs are used for two distinct purposes: (1) as devices for summarizing and communicating climate science to other modelling communities, and (2) as computationally efficient AOGCM emulators. The analysis performed in this manuscript has important implications for both uses, but the authors are failing to distinguish clearly between them. This creates unnecessary confusion (seen especially clearly in the exchanges with previous reviewers), and has in my opinion led to an unbalanced treatment.*
   *The authors appear to recognise the role of PH99 as a communication device when they, in their Discussion (section 5), briefly mention the idea that the 'transient climate sensitivity' might be lower than the ECS, as a potential physical explanation for the lowering of the ECS when the parameters are calculated by fitting the one-box model to AOGCM temperatures instead of derived from*

*AOGCM forcings. But what is the chief physical mechanism behind this? And does this mean that PH99 users should interpret $\frac{\mu}{\alpha}\ln(2)$ as the 'transient climate sensitivity' rather than the ECS? Can we do this without undermining the physical basis for the one-box model? And what about the higher TCR value that you get when fitting the one-box parameters instead of transfering them directly? What is the physical interpretation of this second important change? You also acknowledge that measurement error in AOGCM outputs could lead to biased values of the ECS and TCR in the one-box model, but can you do anything to show that the biases would actually go in the direction of inflating the ECS and deflating the TCR? Perhaps you could just add a random sample of Gaussian deviations to your input data and feed them through the non-linear PH99 mapping to see what the resulting distribution of ECS and TCR would look like?*

When asking for the 'chief physical mechanism' for the observed phenomena the referee asks for climate dynamics input that goes beyond the original scope of our ms. However as also the other two referees had already asked for it, we suggest to derive such an explanation from the simplification occurring when moving from a 2-box to a 1-box model. Preliminary studies of ours indicate that this move can in fact explain the observed phenomena. We offer to elaborate on this explanation in V2. Also, we are open to the numerical experiment referee #3 asks for regarding the effects of Gaussian deviations.

In the end, only from a mixed analytic and numerical analysis we can decide in what

sense PH99 represents a physical model. From this we also offer concluding for what type of integrated assessment analyses it might meaningfully be utilized in the future and how already published results might need to be retro-interpreted.

*I realise I have given you a lot of questions to answer, but I really do feel that this part of the discussion has been unduly neglected, and the paper would benefit greatly from extending it. It seems a very interesting fact that, for a given TCR, the ECS value transferred directly from an AOGCM is systematically higher than the value that would yield the best fit to that same AOGCM (and vice versa for TCR). Anything the authors are able to do to help the reader understand the causes of the differences between fitted and transferred parameter values, and how this might affect the physical interpretation of the one-box model paramters, would be very welcome.*

We are very grateful for the referee sharing our impression that inferred ECS and TCR values are biased as against their fitted counterparts do represent 'a very interesting fact'. The latter and its practical consequences is the key motivation of having this ms! We expect that V2 will deliver here, along the lines outlined in our previous comment.

*I would, compensatingly, recommend shortening the discussion of the second use of PH99, as an AOGCM emulator, which currently takes up the majority of sections 4*

*and 5. I think it is interesting to consider the advantages and disadvantages of alternative methods for choosing ECS and TCR for emulation purposes, but I often felt lost in this discussion and think it can be done more concisely. The fitting method, perhaps unsurprisingly, does a pretty good job of fitting the AOGCM temperature time series. But the key drawback of the fitting-method is that it's inappropriate for obtaining probability distributions for the ECS and TCR that can be used to simulate PH99 under uncertainty. These kinds of simulations are now standard practice for economic assessments of climate policy based on coupled climate-economy models, so this is indeed an important issue to wrestle with.*

We see potential in condensing the emulator discussion indeed.

*The quadratic and cubic fitting in Figure 5 seems useful mostly as a cautionary example of 'what not to do.' The authors already explain that it's likely to lead to unphysical parameter values, and as a previous reviewer pointed out, the curvature seems largely a consequence of a single AOGCM run with a low ECS. Overall, the authors can probably devote less space on this particular exercise and be even clearer that it is ill-advised. Instead, they should focus on the more physical interpolation/extrapolation methods considered in section 4, and try to offer users more concrete advice about when they might prefer the Lorenz curve method, or the ECS-to-ECS and TCR-to-TCR fit, or the ECS/TCR-to-ECS fit, or when all three are likely to perform poorly.*

V2 would comply with this.

*I think a slight reorganization of sections 4 and 5 would probably be the most effective way of accomplishing all of this. The new section 4 would take the first two paragraphs of the current section 5 as its starting point, but elaborate along the lines I have discussed above in order to offer a discussion of the physical interpretation of fitted ECS and TCR relative to the directly transferred ones. The new section 5 would merge the current section 4 with the remainder of the current section 5, in order to offer a discussion of the appropriate and inappropriate ways to interpolate and extrapolate ECS and TCR values in PH99, in light of their physical reinterpretation in the new section 4. This separation would also make it much clearer how the choice of parameter values for PH99 depends on whether one is using it as a communications device or as an emulator.*

We will very carefully consider this advice. In the end also the practitioner must know what could be done in case PH99 should be utilized further. V2 would map application onto an appropriate fitting method much clearer.

*Technical corrections*

1. *p. 1, line 9 (and throughout): The manuscript refers to FUND and MIND as two*

*coupled climate-economy model that employ a one-box model. PAGE does as well (see discussion in Calel & Stainforth, 2017, BAMS, already cited).*

We will include PAGE in V2.

2. *p. 2, line 30: Typically, these models are used to study optimal climate policy, so it would be good if you could cite a few studies where these models are used specifically to study 2 degree stabilisation scenarios.*

We will do this in V2.

3. *p. 5, line 12: Typo. "APGCM" should be "AOGCM."*

We will correct this in V2.

4. *p. 5, lines 17-22: While RCP4.5 is certaintly out-of-sample, it's less obvious to me that it serves the purpose of validating the method for 2 degree stabilization scenarios. As a validation exercise, wouldn't it be preferable to fit $\alpha$ and $\mu$ using RCP4.5 and then drive the one-box model using the RCP2.6 forcings?*

We had thought that fitting a climate model to a climate state #1 (defined by RCP2.6) and then validating by a climate state #2 (defined by RCP4.5) even further away from

the preindustrial state would represent a tougher test than the reversed order of calibration and validation. But we are more than happy to also test the order as suggested by the referee as we expect an even better fit.

5.  p. 8, line 11: Typo. "againsta" should be "against a."

We will correct this in V2.

6.  p. 8, line 15: Typo. "radiative active" should be "radiative activity" or "radiative forcing."

We will correct this in V2.

7.  p. 8, line 19: I think it's inaccurate to say that "studies based on PH99 implicitly worked with ECS values that were larger than announced." I think you've made the point that they might be using a higher ECS than would be appropriate for emulating AOGCMs, but this is quite different. They declare their ECS values, and the question raised in this manuscript is whether they shouldn't be using ECS but rather some 'effective ECS' or 'transient climate sensitivity' instead. Please rephrase this.

We will do this in V2.

8. *p. 8, lines 20-28: The discussion of log-Normal distributions seems to come out of nowhere. I think the reorganization I have suggested above may resolve this, but please make an effort to link this more strongly to the rest of the discussion.*

V2 will avoid this log-Normal discussion which represents a subject on its own.

9. *p. 8, line 29: Typo. "boefore" should be "before."*

We will correct this in V2.

10. *p. 10, line 7: Typo. "generally" should be "it generally."*

We will correct this in V2.

---

## Author Response (AR1)

Dear Editor,

We are grateful for having been granted the chance to improve our ms based on three referees' comments. The main structural changes are as follows:

- We clarify the scope of our article much more explicitly in the beginning of the introduction to avoid further misunderstandings like those that apparently misled reviewer 1 in our version 1.
- We deliver a physical mechanism to explain the structural difference PH99/AOGCMs prior re-calibration. We devote a whole new chapter to this question.
- We deliver the whole matrix of possible calibration vs validation combinations, spanned by the four RCP options to verify our approach.
- We reformulated our ms more like a warning before a naive usage of PH99 rather than an advertisement.
- We involved a native speaker to proofread our ms.

As the ms is already unusually long, we abstain from involving Ruelle's theory. Rather we stay with a very basic approach (1 box vs 2 box intercomparison) that we expect would immediately appeal to the intuition of most readers.
We would be delighted to see that this version of the ms, upgraded along the suggestions of the referees, is found suitable for publication in ESD.

Sincerely,
Mohammad M. Khabbazan

*Reply on*
*The manuscript by Khabbazan and Held assesses the performance of a very simplified climate module currently in use in some IAMs. In particular, the study is motivated by the need to adjust the existing tools to the capability of this module in the light of assessments of below 2°C scenarios. To that end, it is fitted to different CMIP5 RCP2.6 AOGCMs.*

This is the very point of the first version of our ms indeed.

*The manuscript contains no fundamental flaws although a re-read is in order and the literature list should be checked.*

Done.

*One key paper (Foster et al. 2013) is for example missing from the literature list. I presume it's Forster, P. M., T. Andrews, P. Good, J. M. Gregory, L. S. Jackson, and M. Zelinka, 2013: Evaluating adjusted forcing and model spread for historical and future scenarios in the CMIP5 generation of climate models. J. Geophys. Res. Atmos., 118, 1139– 1150*

The referee is right. This is the very reference that represents the data basis for our

work. We sincerely apologize for this flaw. We will make sure that flaws like this one

cannot occur in V2.

*More fundamentally, however, the scientific advancement presented of this study in my assessment is rather poor and it neglects important recent literature in this context (in fact, the literature list is rather short and at least 3 years old).*

In our new introduction we now clearly state our research question. Furthermore we added a semi-analytic section which explains the physical mechanism of the observed PH99-AOGCM discrepancy. We recommend recalibrating PH99 before using it and thereby we also allow for a re-interpretation of existing literature. Finally we demonstrate how PH99 can be re-calibrated such that it would match AOGCM output for *any* RCP. Thereby we have extended the scope of our article. Regarding omitted literature, to our impression the referee mainly refers to the effects outlined in the following paragraph. As outlined below, our ms is *not* about the issue raised by the referee in that paragraph.

*On the methodological approach: What's the justification of using the PH99 model (apart from it 'being there')? The authors argue that it's computational efficiency, ...*

PH99 is can be interpreted as an energy balance model (for details of the justification see Petschel-Held et al., 1999, and references therein; the authors' list contains Klaus Hasselmann! Furthermore, the model was validated in Kriegler and Bruckner, 2004 – however the validation was done in a different way and also did not have the forcing reconstructions by Forster et al., 2013, at hand). Even today, computational efficiency is key, e.g. when it comes to decision-making under endogenous learning (see e.g. Webster et al., 2012).

*...but how exactly are they convinced that their treatment of non-CO2 GHGs is appropriate. For strong mitigation pathways, these 'minor' differences may become very important, last but not least to determine net-zero global GHG forcing etc. I'd think they would need validate their fit using other strong mitigation scenarios with different non-GHG trajectories (if no others are available then from the GeoMIP experiment) rather than RCP4.5. In particular, it appears that non-CO2 gases obscure our assessments of ECS (see e.g. Myre et al. 2016)*

*As it stands, I'm not convinced that the simplified model is capable of including non- GHG forcing in a sufficient fashion for the question at hand (i.e. staying below 2˚C or 1.5˚C).*

Apparently here we provoked a key misunderstanding what our article is about. It is *not* about how to generate a global total forcing out of regional forcings. Our ms is about how to get from a global total forcing to global mean temperature and whether PH99 does a good enough job in that, i.e. could substitute for AOGCMs or more complex integrated assessment climate modules in that regard. We have made this point much clearer in the introduction by phrasing the concrete research question. We nevertheless include all RCPs for validation including RCP6.0 which contains a strong ozone component. Also here, PH99 proves successful.

Finally, any author within integrated assessment of the coupled climate-economy-problem has to deal with how to construct a meaningful global forcing out of regionally disaggregated AOGCM forcings, as AOGCMs cannot directly be utilized in economic optimizations. This is a difficult discussion indeed, but *any* climate module within integrated assessment would face this issue, not only PH99. This discussion, however, is not the subject of our ms.

*On the application: I didn't fully the motivation for step 1. What was the reasoning for the authors to assume that their PH99 model would work with AOGCM diagnostics from Forsters et al. directly? Obviously, the derived feedback response time parameter 1/alpha of 34.5 years in the multi-model mean is quite unphysical. It seems that the PH99 model is not equipped to be used in that context.*

This is the very point. We want to highlight that *current practice* in integrated assessment of directly prescribing ECS and other parameters from AOGCMs leads to biases. As an AOGCM also contains time scales faster than 35 years, it is not immediately clear that an average time scale of smaller and larger time scales would be meaningless. We more clearly cite the relevant literature on FUND, MIND, PAGE.

*In a next step, the authors find that with two free parameters they are capable of achieving better fits. That's not particularly surprising, but a physical interpretation of these differences is virtually absent? ECS is substantial decreased by almost 1˚C. Can this be understood? The authors continue with fitting derived and fitted ECS and TCR, but I would rather like to see a physical interpretation or an extension of the PH99 model that*

*would correct for this. The authors should also consider their approach in the light of alternative simplified approaches out there i.e. based on a response function approach as in*

*Ragone et al. (2016).*

We did not find it easy to dig out what literature the referee is referring to as no exact reference is supplied. We can only guess that the referee refers to Ruelle's response theory. However we find that Ruelle's response theory is much more general (in allowing for the forcing entering the ODE system augmented by some function of the system's state) than our system at hand. Hence we prefer for the context of this article to stick to rather elementary explanations in terms of 1 mode trying to substitute a superposition of 2 modes – this is what the problem at hand simply is about (the length of our ms is already way above 30 pages). In fact, by having found that the discrepancy we report can be explained by the move from a 2-box to a 1-box model, we have generated an anchor for explaining the reported ECS effect. A key problem of the 1-box model is that it replaces the slow-component response by the averaged faster one and hence would lead to an overreaction for peak-and-decline forcing scenarios such as RCP2.6. In that sense we expect RCP2.6 as particularly difficult to emulate. Kriegler and Bruckner, 2004, could not do this validation, as mitigation scenario forcing reconstructions were not available at that time.

*The authors then want to apply an effective correction for their dubious model in the first place. Their results here appear to be prone to outliers. Compare e.g. the low ECS outlier in Fig. 5. When removing it, I guess even a linear fit would deliver decent results and I'm not sure I can deduce any robust trends from these graphs…*

V2 hopefully avoids the impression that we plainly advertised utilizing PH99. We simply want to state how to interpret older work based on PH99, and how it could be used if someone wants to use it in the future, in spite of the discovered problems. In addition to computational efficiency, for some applications also analytic tractability or conceptual simplicity might be arguments for using PH99.

Furthermore we have done the sensitivity study the referee suggested by repeating some analyses w/o outliers. W/o outliers linear fits would make PH99 parameters predictable by ECS indeed (see ms-Fig. 6). The scenario fit results deliver similar quality as before (see ms-Fig. 5). However we would like to stress that our ms recommends direct correction of ECS along ms-Fig.6 where outliers play a less prominent role. V2 have a short discussion of the outlier issue.

*On the application of this. It seems that the model that is used in these IAMs has many flaws.*

*The question then becomes why it is used at all? And not abandoned for a carbon budget approach that would be even more computational effective and can be determined with more complex models also for these low emissions scenarios (i.e. Rogelj 2016). That becomes in particular relevant since the mitigation challenge ahead is to define pathways that hold warming 'well below 2°C'. "Below 2°C" was interpreted as a 66% chance of non-exceedance (IPCC 2014). What's the added value of using a PH99 model in this context? Would they select an ECS at the 66% quantile and then use this as a basis for the IAM derivations? And if so, why not use carbon budgets directly?*

We are grateful for this exciting suggestion. It is a fascinating question indeed when a dynamic climate module could simply be replaced by the carbon budget approach to deliver similar – or even better quality – in emulation of an AOGCM. This discussion is beyond the scope of this ms. In any case, there *are* applications in climate economics where timing matters, such as cost benefit analyses (see e.g. Nordhaus and Sztorc, 2013) or cost risk analyses (see e.g. Neubersch et al., 2014).

In summary we are optimistic that a new version V2 could be acceptable for the reviewer as (i) it clarifies its scope: being about the link from total forcing to temperature and (ii) it delivers a physical interpretation of the observed effects. We are grateful for the referee's comments as they will have triggered an – in our view – considerably upgraded version of our ms.

**References Review 1**

Forster, P. M., Andrews, T., Good, P., Gregory, J. M., Jackson, L. S., & Zelinka, M. Evaluating adjusted forcing and model spread for historical and future scenarios in the CMIP5 generation of climate models. Journal of Geophysical Research: Atmospheres, 118(3), 1139-1150, 2013.

Kriegler, E. and Bruckner, T.: Sensitivity analysis of emissions corridors for the 21st century, Climatic Change, 66, 345–387, 2004.

Neubersch, D., Held, H., & Otto, A. Operationalizing climate targets under learning: An application of cost-risk analysis. Climatic change, 126(3-4), 305-318, 2014.

Nordhaus, W., Sztorc, P., DICE 2013R: Introduction and User's Manual, dicemodel.net , 2013.

Petschel-Held, G., Schellnhuber, H.-J., Bruckner, T., Toth, F. L., and Hasselmann, K.: The tolerable windows approach: Theoretical and methodological foundations, Climatic Change, 41, 303–331, 1999.

Webster, Mort, Nidhi Santen, and Panos Parpas. An approximate dynamic programming framework for modeling global climate policy under decision-dependent uncertainty. Computational Management Science 9.3: 339-362, 2012.

*Reply on*
First of all, we would like to thank the referee for an exceptionally thorough and thoughtful review!

Original comments by the referee will be highlighted in italic font below.

**1 General Comments**

*Khabbazan and Held's paper checks the performance of a one box energy balance model (PH99), currently in use in the integrated assessment models FUND and MIND, against output from AOGCMs before suggesting a simple, improved way to use it in future. Their major conclusion is that, for strong mitigation scenarios, prescribing ECS and TCR to PH99 from Forster et al. (2013) with no further calibration implicitly causes researchers to sample much larger temperature responses than they intend to. They show that a simple fitting exercise rectifies this and validate the fit by checking PH99's performance under one other scenario. This scenario is very similar to the one they*

*used for fitting. They then explore different methods to map AOGCM ECS and TCR onto 'effective' PH99 values which could provide researchers with a simple method of revealing the temperature response they are actually considering.*

We feel our ms perfectly perceived by the referee.

*My major concerns focus on whether the analysis shows that PH99 is a valid energy balance model rather than a fitting tool.*

That is a difficult question, an almost philosophical one. PH99 is physical in the sense that it is an energy balance model, that larger forcing results in larger GMT rise and that there is linear delay between forcing and response. However the time-scale and forcing-shape specific calibration necessary for PH99 points to a not so physical model.

Also, in V2 we hopefully avoid the impression that we would advertise utilizing PH99 without reservation. We simply want to state how to interpret older work based on PH99, and how it could be used if someone wants to use it in the future. In addition to computational efficiency, for some applications also analytic tractability or conceptual simplicity might be arguments for using PH99.

*I also think that the writing style could be greatly improved.*

V2 hopefully is improved in that regard. A native speaker was involved in correcting the ms.

*I think the authors point out some key errors which arise if PH99 is used without care and explore a few ways for modellers to quickly relate their parameters to AOGCM ECS and TCR values. However, given that the authors argue for mapping AOGCM properties onto 'scenario-class-specific values before using them in PH99', which appears to undermine any physical basis for PH99, I'm left wondering if this paper highlights the limitations of PH99 rather than providing strong arguments for its use.*

In fact we expand on the scope of utilizing PH99. We can show that after proper recalibration on AOGCM-RCP2.6, that calibration can be utilized for all RCPs – however with limited accuracy for RCP 8.5.

**2  Major concerns**

1. *The re-callibration of PH99 is only validated for RCP4.5. There is no other testing of the performance outside of RCP2.6 and RCP4.5, two very similar scenarios, nor testing of the effect of different non-CO2 forcing pathways. Thus the authors have shown that a good fit to AOGCM GMT output can be done with two free parameters and that this fit is good for a similar scenario. I wonder if testing over a greater range of other scenarios would strengthen the justification for the use of PH99.*

We show in V2 that PH99 emulates *all* RCPs by the identical recalibration. This also contains RCP6.0 with some stronger ozone component. We also show from a semi-analytical treatment that PH99 is a good emulator of a whole 2-dimensional manifold of mitigation forcing scenarios. Furthermore, this ms is *not* about how to generate a meaningful global forcing from non-CO2 agents, but how to prognose global mean temperature from total global forcing. V2 is much clearer about this scope. The effects by moving from a 2-box to a 1-box model plays a prominent role in V2. How to get to a total forcing, however, is a discussion that hits *any* climate module utilized in integrated assessment, and is beyond the scope of this ms.

2. *The initial testing of the performance of PH99 against AOGCMs reveals a key, hidden, bias of this model if used without validation in strong mitigation scenarios. This is a good bit of analysis. As a result of this analysis, the authors advocate mapping AOGCM climate system properties onto 'scenario-class-specific values before using them in PH99'. Whilst this seems to be necessary for acceptable performance of PH99, it also appears to undermine any physical basis for PH99. If you have to re-callibrate PH99 every time you want to use it in a different scenario class then its parameters lose all physical meaning and instead simply become fitting parameters. Thus the authors appear to advocate shifting PH99 from an energy balance model to a function that can be fitted to AOGCM data and then used for a limited range of scenarios?*

We now know that we undersold PH99 in that regard. This underselling is stopped in V2. Instead the underlying physical mechanisms will be explained from which also the observed discrepancy will be derived. No re-calibration is necessary for generic RCP forcing scenarios.

3. *I don't think I am wrong in saying that this model is ultimately meant to be used by those who are looking for simple emulators of global mean temperature response and hence may not be climate modellers themselves. If this target audience can't pick up this paper and get some sense of what is going on then they will struggle to use any of the fits provided. A paper on 'the most parsimonious climate module' should have a style which reflects its title. Given that parsimonious is synonymous with 'simple' in this context, it makes sense for the communication to be as plain, clear and simple as possible too. With this*

*goal in mind, I suggest numerous technical corrections and ask for multiple clarifications.*

We fully agree and are extremely open to the detailed changes the reviewer detailed below. In particular we deliver, as we believe, a clear-cut interpretation of the observed discrepancy PH99-AOGCM.

4. *The exploration of different possible parameterisations of the relationship between AOGCM ECS/TCR and effective ECS/TCR is, in my opinion, worthwhile. My impression is that they recognise that a parameterisation would be nice but don't have strong enough evidence to recommend any of the ones they have tried and so the results here are underwhelming.*

V2 now explains why a lower ECS in conjunction with a lower time-scale is adequate to emulate AOGCMs for the time-horizon relevant for climate policy evaluation.

**3 Specific Comments**

1. *As an exercise, the fitting that is done is scientifically sound re methods, assumptions, results, and reproducibility as far as I can tell. I can also see that it would be useful for modellers who wish to use a simple emulator but don't wish to do the calibration themselves.*

Thank you!

2. *I think this paper shows that PH99 is closer to a fitting tool rather than a physical model. Hence I wonder, if modellers are after computational simplicity and a fitting tool, why wouldn't they use a simple carbon budget target or emissions pathway to constrain their model. There is already research on how emissions pathways and targets map to temperature targets so this could be used to back out emissions constraints from a given temperature target for a given scenario class. This approach seems far simpler than introducing an energy balance module which requires atmospheric concentration and radiative forcing input, has little physical basis and hasn't been validated over a wide range of CO2 and non-CO2 scenarios so might not produce realistic temperature projections anyway.*

We are grateful for this exciting suggestion. It is a fascinating question indeed when a dynamic climate module could simply be replaced by the carbon budget approach to deliver similar – or even better quality – in emulation of an AOGCM. This discussion is beyond the scope of this ms. Here we simply would like to stress that there are applications in climate economics where timing matters, such as cost benefit analyses (see e.g. Nordhaus and Sztorc, 2013) or cost risk analyses (see e.g. Neubersch et al., 2014).

3. *The introduction calls IAMs an 'indispensable tool'. I acknowledge that this comment is made in the context of 'driving welfare-optimal climate policy scenarios' so it is accurate. However given that there are many who disagree with using economic analyses for determining 'welfare-optimal scenarios' because of the need to monetise many things which arguably can't be monetised (e.g. the environment), using this term seems to open the paper up to unwanted distractions. I think this could be avoided with a simple re-wording; calling IAMs a 'tool which are used to derive welfare-optimal scenarios' rather than an 'indispensable tool used to derive welfare-optimal scenarios'. This change would avoid opening up an economic debate (in the reader's mind) which is completely outside the scope of this paper.*

V2 complies with the reviewer's suggestion – wording changed accordingly.

4. *page 8, line 22: 'personal conviction'. I don't think personal convictions have any place in scientific papers. Either the evidence is there to support using log-normal distributions or it's not. I also don't understand what the sentence beginning with 'This conviction rests' means. Does it mean 'Schneider von Deimlinig et al. show that constraining ECS by paleo data results in thin-tailed distributions'? If yes, then there is no need for a 'personal conviction', circling back to my first point.*

We have eliminated this paleo discussion from V2. For V2 it is sufficient to say that lognormal distributions for ECS are used in the literature and that under an affine transformation (as suggested in ms-Fig.6) they would be mapped onto lognormal distributions the quantiles of which are still compatible with what is reported in IPCC AR5 WGI on ECS.

Yes, the point was that Schneider von Deimling et al., 2006, would allow for a thin tail on ECS. The term 'conviction' might not have been the accurate term for what we tried to express. Our point was that a single paper does not yet trigger a paradigm shift. Major fractions, if not the majority of climate researchers – in contrast to us – doubt that ECS can be constrained by paleo data. This fundamental question is still open and hence also subject to personal believe and intuition that might guide the choice of the always in-part subjective distributions of ECS (as any of them rests on Bayesian learning that needs a subjective prior as an input).

5. *I really appreciated the discussion of low pass filtering and think this was well done.*

Thank you.

6. *In section 4 (page 7, lines 32-34), the authors state that 'regressing both inferred effective ECS and TCR solely against AOGCMs' ECS obviously is the overall better approximation'. Whilst this is borne out by taking a pure average of all the results, there are clearly two strong outliers which are having a major effect on the performance of the ECS-ECS & TCR-TCR mapping. I wonder what is causing such large outliers (they seem hugely anomalous) and if removing them would be justified. If they are removed, how much does this change the conclusions.*

We have a short discussion on that issue in V2. Results on the PH99 parameter-based fit method (see ms-Fig. 5 and ms-Fig. 6) show that eliminating outliers and then moving to a linear fit (suggested by referee #1) will not significantly change the quality of emulation (see Reply-Fig. 2).

**4** *Technical Corrections*

The innumerous suggestions made by the referee appear very meaningful and appropriate to us and have been implemented in V2. Please notice that we have also asked a native speaker to proofread our ms.

In summary we are optimistic that a new version V2 could be acceptable for the reviewer as (i) it clarified its scope: being about the link from total forcing to temperature, (ii) it delivered a physical interpretation of the observed effects, (iii) reflected all of the technical corrections. We

are grateful for the referee's comments as they will have triggered an – in our view – considerably upgraded version of our ms.

**References Review 2**

Neubersch, D., Held, H., & Otto, A. (2014). Operationalizing climate targets under learning: An application of cost-risk analysis. Climatic change, 126(3-4), 305-318.

Nordhaus, W., Sztorc, P., DICE 2013R: Introduction and User's Manual, dicemodel.net , 2013.

Petschel-Held, G., Schellnhuber, H.-J., Bruckner, T., Toth, F. L., and Hasselmann, K.: The tolerable windows approach: Theoretical and methodological foundations, Climatic Change, 41, 303–331, 1999.

Schneider von Deimling, T., Held, H., Ganopolski, A., and Rahmstorf, S.: Climate sensitivity estimated from ensemble simulations of glacial climate, Clim Dyn, 27, 149–163, doi:10.1007/s00382-006-0126-8, 2006.

*Reply on*
First of all, we would like to thank the referee for an exceptionally thorough and thoughtful review!

Original comments by the referee will be highlighted in italic font below.

*General Comments*

*This manuscript investigates the performance of a one-box energy balance model (PH99) as an AOGCM emulator for strong mitigation scenarios. The authors find that this simple climate model (SCM) consistently over-predicts future temperatures when the ECS and TCR are transferred directly from AOGCMs. Fitting the PH99 directly to the AOGCM temperature time series eliminates this bias, and reveals that the AOGCMs time series imply a substantially lower ECS and higher TCR than what they had transferred directly. The manuscript briefly discusses the physical interpretation of this discrepancy, and also explore alternative ways of fitting the one-box model that might be more reasonable for extrapolation in parameter space (of the kind performed when these SCMs are used to investigate the optimal dynamic behaviour of a decision maker under uncertainty).*

We feel our ms perfectly perceived by the referee.

*Before continuing further, I want to briefly highlight two important factors that might reasonably affect how you read this review. First, I have not only read the manuscript, but also the previous reviews and the responses from the authors. My comments primarily address the manuscript itself, but I will also sometimes explicitly agree or disagree with comments that have been made earlier in the process. Second, I have not approached this manuscript as a climate physicist, but rather from the perspective of a researcher who uses the integrated assessment models with SCMs like PH99. My comments will therefore differ in spirit from those of earlier reviewers, and I focus more on issues I believe to be more relevant to those who would use this research.*

While we appreciate comments from all relevant disciplines, we are happy that among the referees there is a referee who uses integrated assessment models.

**Specific comments**

*My overall assessment is that this manuscript offers an interesting contribution and should be published.*

Thank you!

*A previous reviewer expressed concern that the scientific contribution may be inadequate, but I feel that this comment does not adequately consider the policy influence that the one-box model wields (or rather, simple integrated assessment models that use PH99 in one form or another). For instance, two of the three climate-economy models used by the US federal government to calculate the social cost of carbon incorporate one-box energy balance models (notwithstanding recent political developments). The policy analysis in the Stern Review, which was commissioned and used by the UK government to formulate climate policy, was also based on a coupled climate-economy model that incorporated a one-box energy balance model. By their simplicity, these SCMs are also have come to serve as tools for translating new climate science for communities that use climate information but generally lack extensive physics training. Even a relatively small improvement in our understanding and handling of these models would provide a significant contribution.*

We feel our ms perfectly perceived by the referee. We added a reference to the Stern review.

*I do have some concerns about the manuscript, though. First, I think there are parts of the manuscript that will be difficult to decypher for many of the researchers that actually work with SCMs in the context of simple climate-economy models. Second, I think that authors have tended to focus excess attention on concerns related to interpolation and extrapolation of parameter values, at the expense of a fuller and clearer discussion of the physical interpretation of their primary findings. I discuss each point in turn, and offer a few minor comments at the end. Let me state clearly, though, that I expect these concerns can be fully redressed, so I wouldn't consider them reasons for rejecting the manuscript.*

We are thankful to the reviewer for the time and efforts. We are open to any comments that can enhance the manuscript.

1. *The heart of this manuscript, as I see it, is the direct transfer of AOGCM characteristics (section 2.1). The central issue is whether or not it is appropriate to use this physical method for deducing the parameter values in one-box model. The subsequent question about whether alternative methods for fitting the parameter values perform better, is also tied to this baseline method. So section 2.1 is really the foundation for all of the analysis in this paper. Yet two cruicial pieces seem to be missing from it.*

   *First, at this point in the manuscript the authors should be offering a childishly clear explanation of how (and which) AOGCM outputs can be used to deduce the values of $\alpha$ and $\mu$, which can then be plugged into equations (2) and (3) to retreive the implicit ECS and TCR, respectively. But I must admit to having some difficulty following their derivations (e.g. not understanding how h is determined in equation (7) where both h and $\mu$ appear to be unknowns, and not seeing any expression for $\alpha$ in terms of AOGCM output). A climate physicist will perhaps be so familiar with this material as to be able to perform these calculations with little prompting from the authors, but as a presumptive member of the intended audience, I would appreciate it if the authors exercised greater pedagogical care in this section.*

We added a clean recipe how to deduce PH99's parameters from ECS & TCR.

*Second, and just as important, is that the authors have not offered any information to suggest that this is how modellers are currently choosing values for the ECS and TCR. In my experience, many users will not themselves try to deduce these parameters from AOGCM outputs, but rather plug in values of ECS and TCR reported in IPCC chapters or specific academic papers without fully understanding how these values are inferred from AOGCM simulations (and sometimes adding a bit of 'calibration' to make sure the results don't look too dissimilar from MAGICC, say). If those reported ECS and TCR values are derived in this way generally, the authors should state this clearly and cite examples. If not, they should consider whether it would be more appropriate to use a different baseline.*

We thank the referee for pointing us to another misunderstanding we might have provoked. In fact, the referee's perception of standard practice is actually what we wanted to refer to in our manuscript, as we believe that users of PH99 have not themselves tried to deduce these parameters from AOGCM outputs. However, given that insight, then comparing AOGCM and PH99 output while both models would be characterized by identical ECS and TCR and be driven by identical radiative forcing is then the most direct way we can currently imagine to demonstrate a bias encoded in PH99. We added a paragraph at the end of Section 1 accordingly.

*As a suggestion, I think it would be worthwhile to run the model using the actual parameter values assumed in FUND and MIND as a baseline (and maybe PAGE, which the authors seem to have ignored, even though it incorporates a one-box model). If I were to speculate, I would guess that using the default parameter values from these models will give even more discrepant predictions, so in addition to being more relevant to current practice, it might illustrate your point even more clearly.*

Thanks for pointing us to PAGE. We added it to V2. Furthermore we explicitly commented on the standard value of ECS=3°C as utilized in FUND and PAGE. The PH99-equivalent should be about 4°C.

2. *SCMs are used for two distinct purposes: (1) as devices for summarizing and*

*communicating climate science to other modelling communities, and (2) as computationally efficient AOGCM emulators. The analysis performed in this manuscript has important implications for both uses, but the authors are failing to distinguish clearly between them. This creates unnecessary confusion (seen especially clearly in the exchanges with previous reviewers), and has in my opinion led to an unbalanced treatment.*

*The authors appear to recognise the role of PH99 as a communication device when they, in their Discussion (section 5), briefly mention the idea that the 'transient climate sensitivity' might be lower than the ECS, as a potential physical explanation for the lowering of the ECS when the parameters are calculated by fitting the one-box model to AOGCM temperatures instead of derived from AOGCM forcings. But what is the chief physical mechanism behind this? And does this mean that PH99 users should interpret $\frac{\mu}{\alpha} \ln(2)$ as the 'transient climate sensitivity' rather than the ECS? Can we do this without undermining the physical basis for the one-box model? And what about the higher TCR value that you get when fitting the one-box parameters instead of transfering them directly? What is the physical interpretation of this second important change? You also acknowledge that measurement error in AOGCM outputs could lead to biased values of the ECS and TCR in the one-box model, but can you do anything to show that the biases would actually go in the direction of inflating the ECS and deflating the TCR? Perhaps you could just add a random sample of Gaussian deviations to your input data and feed them through the non-linear PH99 mapping to see what the resulting distribution of ECS and TCR would look like?*

We invented a whole new Section to deliver on the 'chief physical mechanism'. In that vein we can explain why ECS must be downsized in PH99 and accordingly also the response time scale.

*I realise I have given you a lot of questions to answer, but I really do feel that this part of the discussion has been unduly neglected, and the paper would benefit greatly from extending it. It seems a very interesting fact that, for a given TCR, the ECS value transferred directly from an AOGCM is systematically higher than the value that would yield the best fit to that same AOGCM (and vice versa for TCR). Anything the authors are able to do to help the reader understand the causes of the differences between fitted and transferred parameter values, and*

*how this might affect the physical interpretation of the one-box model paramters, would be very welcome.*

We are very grateful for the referee sharing our impression that inferred ECS and TCR values are biased as against their fitted counterparts does represent 'a very interesting fact'. The latter and its practical consequences is the key motivation of having this ms! We feel that V2 does deliver here, indeed.

*I would, compensatingly, recommend shortening the discussion of the second use of PH99, as an AOGCM emulator, which currently takes up the majority of sections 4 and 5. I think it is interesting to consider the advantages and disadvantages of alternative methods for choosing ECS and TCR for emulation purposes, but I often felt lost in this discussion and think it can be done more concisely. The fitting method, perhaps unsurprisingly, does a pretty good job of fitting the AOGCM temperature time series. But the key drawback of the fitting-method is that it's inappropriate for obtaining probability distributions for the ECS and TCR that can be used to simulate PH99 under uncertainty. These kinds of simulations are now standard practice for economic assessments of climate policy based on coupled climate-economy models, so this is indeed an important issue to wrestle with.*

*The quadratic and cubic fitting in Figure 5 seems useful mostly as a cautionary example of 'what not to do.' The authors already explain that it's likely to lead to unphysical parameter values, and as a previous reviewer pointed out, the curvature seems largely a consequence of a single AOGCM run with a low ECS. Overall, the authors can probably devote less space on this particular exercise and be even clearer that it is ill-advised. Instead, they should focus on the more physical interpolation/extrapolation methods considered in section 4, and try to offer users more concrete advice about when they might prefer the Lorenz curve method, or the ECS-to-ECS and TCR-to-TCR fit, or the ECS/TCR-to-ECS fit, or when all three are likely to perform poorly.*

We are thankful to the referee for this informative comment. We have included the suggestion by referees to omit an obvious outlier and repeat the regression (see ms-Fig. 5 and ms-Fig. 6). W/o outliers linear fits would make PH99 parameters predictable by ECS

indeed (see ms-Fig. 6). The scenario fit results deliver similar quality as before (see ms-Fig. 5). However we would like to stress that our ms recommends direct correction of ECS along ms-Fig.6 where outliers play a less prominent role and keep all the rest of the analysis in that section as it is for a better comparison. More specifically regarding the Lorenz curve, as it is vastly used by by MIND users, we see the benefits of keeping it in V2 for interested readers.

*I think a slight reorganization of sections 4 and 5 would probably be the most effective way of accomplishing all of this. The new section 4 would take the first two paragraphs of the current section 5 as its starting point, but elaborate along the lines I have discussed above in order to offer a discussion of the physical interpretation of fitted ECS and TCR relative to the directly transferred ones. The new section 5 would merge the current section 4 with the remainder of the current section 5, in order to offer a discussion of the appropriate and inappropriate ways to interpolate and extrapolate ECS and TCR values in PH99, in light of their physical reinterpretation in the new section 4. This separation would also make it much clearer how the choice of parameter values for PH99 depends on whether one is using it as a communications device or as an emulator.*

We very carefully considered this advice. In the end also the practitioner must know what could be done in case PH99 should be utilized further. V2 now maps application onto an appropriate fitting method much clearer. Please notice that we have added a section (section 5) as a response to the reviewers' call for clearer physical interpretation.

***Technical corrections***

1. *p. 1, line 9 (and throughout): The manuscript refers to FUND and MIND as two coupled climate-economy model that employ a one-box model. PAGE does as well (see discussion in Calel & Stainforth, 2017, BAMS, already cited).*

   We included PAGE in V2.

2. *p. 2, line 30: Typically, these models are used to study optimal climate policy, so it would be good if you could cite a few studies where these models are used specifically to study 2 degree stabilisation scenarios.*

For MIND and PAGE, such papers are cited. FUND generically does not result in 2° paths, however, the bias found by us extends to any concave forcing, hence any paper cited on FUND would deliver here.

3. *p. 5, line 12: Typo. "APGCM" should be "AOGCM."*

Corrected for.

4. *p. 5, lines 17-22: While RCP4.5 is certaintly out-of-sample, it's less obvious to me that it serves the purpose of validating the method for 2 degree stabilization scenarios. As a validation exercise, wouldn't it be preferable to fit $\alpha$ and $\mu$ using RCP4.5 and then drive the one-box model using the RCP2.6 forcings?*

We had thought that fitting a climate model to a climate state #1 (defined by RCP2.6) and then validating by a climate state #2 (defined by RCP4.5) even further away from the preindustrial state would represent a tougher test than the reversed order of calibration and validation. But we are more than happy to also test the order as suggested by the referee as we expect an even better fit. V2 now includes calibration of PH99 to any RCP and validating it as against any other RCP.

5. *p. 8, line 11: Typo. "againsta" should be "against a."*

Corrected. Please notice that we have also asked a native speaker to proofread our ms.

6. *p. 8, line 15: Typo. "radiative active" should be "radiative activity" or "radiative forcing."*

Corrected.

7. *p. 8, line 19: I think it's inaccurate to say that "studies based on PH99 implicitly worked with ECS values that were larger than announced." I think you've made the point that they might be using a higher ECS than would be appropriate for emulating AOGCMs, but this is quite different. They declare their ECS values, and the question raised in this manuscript is whether they shouldn't be using ECS but rather some 'effective ECS' or 'transient climate sensitivity' instead. Please rephrase this.*

§ erased.

8. *p. 8, lines 20-28: The discussion of log-Normal distributions seems to come out of nowhere. I*

*think the reorganization I have suggested above may resolve this, but please make an effort to link this more strongly to the rest of the discussion.*

Done.

9. *p. 8, line 29: Typo. "boefore" should be "before."*

Corrected.

10. *p. 10, line 7: Typo. "generally" should be "it generally."*

Corrected.

[revised manuscript text omitted]

$\cancel{\text{In order to derive }h}$Hence if $h$ were known, the forcing and GMT taken from Forster et al. (2013) could be used to test PH99. In order to relate $h$ to the original parameters of PH99, $\alpha$ and $\mu$, we re-consider the limiting $CO_2$-only case of Eq. (4):

$$h\frac{dT}{dt} = -\alpha h T(t) + Q_2 \frac{\ln c(t)}{\ln 2} \tag{6}$$

$Q_2$ denotes the additional forcing from the doubling of the $CO_2$ concentration $\cancel{\text{as against}}$compared to its pre-industrial value and is listed for $\cancel{\text{all}}$any of the above AOGCMs (see Forster et al., 2013, Table 1). When comparing Eq. (1) and Eq. (6), we obtain:

$$\mu = \frac{Q_2}{h \ln 2} \;. \tag{7}$$

$\cancel{\text{Equation (7) is}}$ in line with Kriegler & Bruckner (2004). Equation (7) allows $\cancel{\text{for determining}}$us to determine $h$, and in turn $\cancel{\text{for}}$to use the time-integrating Eq. (5). $\cancel{\text{Thereby from}}$
[revised manuscript text omitted]

---

## Referee Report (RR1)

**General comments**

Khabbazan and Held's paper highlights the nuances which must be considered when using a one box energy balance model for climate projections (the form they focus on is the one presented by Petschner-Held (1999), herein PH99, but any similar one-box model would exhibit the same behaviour). Their major conclusion is captured in the last paragraph of the paper, specifically that callibrating PH99 is 'much more involved than previously assumed' and hence 'future users should carefully consider whether they actually want to use PH99, or whether they prefer a less parsimonious solution'. On top of this, they also present a lovely bit of analysis which shows why a one box model must use a lower ECS than a two-box model if the two are going to respond similarly to a strong mitigation radiative forcing scenario over an ~200 year timescale.

Having already reviewed the paper in its first iteration, my major concerns now focus on its presentation and communication. In particular, having read the comments by reviewer 3, I am also not convinced that the current form of the paper will allow it to have its maximum impact. Suggestions to address this are below.

Despite this, I now feel that the paper presents some very interesting and pertinent results and so, subject to major revisions to fix a couple of errors, as well as to make the structure and messaging suitably clear for readers outside the field, would recommend it for publication.

**Major concerns**

1. The paper is still extremely difficult to read (I acknowledge the irony of this comment given that my review is probably also difficult to read). It has some extremely good, and pertinent, points to make but the style means they are far less accessible than they should be. I think a full rewrite is required if this paper is to have the impact that it should. Given that all of the science is done, that rewrite should not take too much time. However, before the paper is published, it should be proofread by at least a few other people in the group as the number of errors/incomprehensible passages which remain in this revision suggest that this step has not yet been suitably taken.

2. The authors include statements such as 'Over all, the results show that PH99 would be well trained by being calibrated to any RCP scenario.' alongside '[PH99]'s ECS and TCR are re-interpreted as effective, scenario-class-specific values'. These two comments are contradictory; a model cannot simultaneously both be appropriate for use across a wide range of scenarios and require recallibration for each different scenario class. The message would be made far clearer if the authors were to chose some metric of 'emulation accuracy' and use that throughout to explain how

well (or not) PH99 is emulating the target timeseries rather than using vague statements such as 'excellently emulate', 'appropriately mimic' and 'suitably mimic'. Alternately, the authors could simply present their results and make no comment on the suitability of PH99 as an emulation tool, leaving the reader to make up their own mind about whether it's 'suitably accurate' or not.

3. At the end of the paper, the use cases for PH99 are still not clear. How much faster are one box models like it than two box models? Are there any cases where the increased speed of the one box model over a two box model justifies its use, given how much harder it is to callibrate and its degrading performance outside its callibration range? Put another way, what sort of tradeoffs are made in terms of speed and performance? For example, what is the difference between using e.g. MAGICC, which runs in about a second but has excellent emulation ability, and PH99, which runs in some shorter amount of time (I'd guess a thousandth of a second or even much less) and has rapidly degrading emulation ability outside its callibration scenario type and time horizon.

4. The discussion of 'constant effective oceanic heat capacity, $h$' is wrong. See discussion below.

**Possible ways to help the paper structure**

Generally, I feel that the paper's argument gets broken at awkward times and that this makes its contributions unclear. I also think that its derivations are unclear and would be greatly helped by detailed supporting appendices/supplementary material.

The literature review in the introduction and the discussion of the Lorenz curve in Section 4 are introduced in a way which is particularly jarring to the overall flow of the paper. They interrupt the main flow, which makes it harder to tell which contributions the paper is making and what is existing literature. Shifting such sections around so that they're standalone would make it much easier to see what the contributions of this paper are, and then later compare to existing work.

The derivations are at times impenetrable for all but the most dedicated readers. Adding appendixes which walk the readers through the analytic derviations much more slowly will ensure that readers from outside the field have a much better chance of understanding what is going on and what the papers' equations mean.

Keeping the comment above in mind, Section 5 needs to be re-written. Section 5.1 makes an extremely important point but it is currently very hard work to get there. Section 5.2 may also make an important point, but even after multiple reads I am still not sure what it is (that you can do the same conversion between

2-box and 1-box models as between AOGCMs and a 1-box model, but that it's not as good?). Some better balance between a plain English summary of what the equations mean, more explanation of the steps taken in the derivation (perhaps in an appendix) and the mathematical derivation itself needs to be made. Perhaps Section 5 could stick to a plain English summary, leaving the mathematical details to an appendix/supplementary material?

**Use of effective oceanic heat capacity**

Equation (7) is

$$\mu = \frac{Q_2}{h \ln 2}$$

$$\rightarrow h = \frac{Q_2}{\mu \ln 2}$$

given that $\mu \ln 2$ is the radiative forcing due to a doubling of $CO_2$ concentrations for PH99 (herein $Q_{2PH99}$) then it is clear that $h$ is not an 'effective oceanic heat capacity' as stated in the text but rather a dimensionless scaling factor which is the ratio of the forcing due to a doubling of $CO_2$ in the AOGCM to the forcing due to a doubling of $CO_2$ in PH99 (the units of $h$ make this clear). Hence the correction which is given to allow the authors to use Forster et al's radiative forcing timeseries with PH99 is then

$$Q_{PH99}(t) = \frac{Q(t)_{Forster}}{h} = Q(t)_{Forster} \frac{Q_{2\,PH99}}{Q_{2\,AOGCM}}$$

where here we have used $Q$ to denote forcings, rather than a combination of $F$ and $Q$ as in the paper, for consistency.

Hence what the authors are actually doing is simply rescaling the entire forcing timeseries by the ratio of the AOGCM's and PH99's forcing due to a doubling of $CO_2$. For non-$CO_2$ forcing, which is important in strong mitigation scenarios, this seems completely inappropriate as it assumes some connection between $CO_2$ and non-$CO_2$ forcing which may or may not be there.

The question I ask here is why the authors are using equation (1) as a starting point. Why not simply start with equation (5), or some variant thereof, which takes total forcing as input rather than $CO_2$ only forcing e.g.

$$\frac{d\Delta T}{dt} = \frac{Q(t) - \lambda \Delta T}{d}$$

where $d$ is some time constant of the response.

This would remove the need for the factor of $h$ and make the entire analysis much clearer and easier? I'm almost certain the conclusions would be unchanged (as this is still a one box model with two free parameters) and you wouldn't need this awkward scaling factor throughout.

**Specific comments**

1. 'the most parsimonious SCM, PH99, ensures maximum transparency', this is only true if all the parameter settings etc. used to run the model are documented, a practice which isn't yet commonplace. I think solving the transparency problem is related to more than just model choice, namely accessibility of code, documentation of reproduction steps. A more accurate statement may be that, 'the most parsimonious SCM, PH99 (and variants thereof), ensures maximum comprehensibility'.

2. The discussion around the impact of directly prescribing AOGCM ECS values to PH99 rather than callibrating PH99 is, in my opinion, confusing. ECS has a strict definition and if you ran PH99 until it reached equilibrium, its ECS would be the same as the AOGCMs'. Hence researchers who have directly transferred AOGCM ECS's to PH99 have not sampled 'effectively higher ECS', they have sampled the ECS they prescribed. What they haven't done is sample the intended centennial climate response. They have sampled a response which is higher than that of the corresponding AOGCM.

As a result, I think the implications of the paper would be much clearer if the comments about sampling higher ECS values than intended were all rephrased to focus on sampling higher climate responses to radiative forcing than intended. This would reinforce the important conclusion of this paper and also highlight that using ECS to characterise climate response on a few hundred year timescale is fundamentally flawed, given that ECS takes on the order of a thousand years to emerge.

I think this would also facilitate a simpler comparison with Van Vuuren (2011)'s findings. The models Van Vuuren considered were sampling lower temperature responses. You have shown that the opposite problem emerges when researches directly transfer AOGCMs' ECS values to PH99, i.e. researchers were inadvertantly sampling higher temperature responses than they intended to. Section 5 shows that this is not because they are sampling a higher ECS (they are, by construction, sampling the same ECS and would see the same long-term response) but because PH99 has a fundamentally different response shape to an AOGCM and hence ECS alone does not allow you to easily move between the two (reflecting the paper's main conclusion). You show that the solution to this is to adjust PH99's ECS, sacrificing agreement in the long-term response in

order to gain agreement in the centennial response (which is sensible given it is more often than not the timescale of interest).

**Technical corrections**

page 1, line 9: 'MIND and PAGE' → 'MIND and PAGE, widely used in policy making'

page 1, line 9: delete 'recent' (in fact I think you could delete the entire sentence, not needed in abstract)

page 1, line 12: 'although the model was validated in the past', I have no idea what this is referring to

page 1, line 13: 'overestimate mitigation needs and costs' → 'sample a higher range of climate responses than they intend to'? Whilst it's clear that this increases estimated costs, how much is not that clear and so talking about overestimates seems potentially premature. I'm on the fence about this one so this is more of a thought than a recommendation.

page 1, line 14: 'produced by' → 'resulting from'

page 1, line 15: 'good emulator' → 'good emulator (accurate to within 0.1K)' (it might be more accurate than this, my point here is that having a quantification of 'good' in the abstract would be useful)

page 1, line 19: delete 'on the question'

page 1, line 22: 'larger ECS than claimed' → 'larger climate response to forcing than claimed' (by definition the ECS is unchanged, your results show that it's just that the centenial timescale response is too high for a one box model with the same ECS as a two-box model)

page 2, line 2: 'would' → 'may'

page 2, line 4: delete 'Currently'

page 2, line 3: delete the first 'thousand' and 'were' → 'is'

page 3, line 4: 'Third' → 'Thirdly'

page 3, line 6: C missing after 3 degrees

page 4, line 5: delete 'might be in order', either you believe the note should be there, or you don't (in which case you can delete the entire paragraph)

page 4, line 23: somethign has gone wrong with the brackets at the end of the line (one too many or too few, I can't tell)

page 4, line 26: delete '(The right-hand side of the equation has been obtained by utilizing Eq. (2).)'

page 5, line 6: delete '(see)' (or fix whatever was meant to be there)

page 5, line 8: delete extra comma before the first full stop

page 5, line 12: 'multiply it', not clear what 'it' is

page 5, line 14: changing from T dot to dT/dt halfway through the paper is extremely confusing, please choose a convention and stick with it. Similarly for $F$ and $Q$ throughout.

page 5, line 15: I don't understand what this sentence means

page 5, line 22: 'any' → 'all' and delete 'above' (or add something above, there's nothing there at the moment)

page 5, line 32: 'beyond CO2' → 'which includes non-$CO_2$ forcing'

page 6, line 3: why is 1881-1910 used as pre-industrial? Seems an odd choice for pre-industrial for a perturbation model as it's clearly not a pre-industrial period (although this may not matter in the end)

page 6, line 9: lack of quantification of 'tolerable' (see discussion above) makes it very hard to know what you mean here

page 6, line 16: if you need to delete something, this paragraph on drift could go. Whilst it is nicely done, it is also a fairly obvious point that PH99 itself acts as a low-pass filter

page 6, line 28: 'APGCM' → 'AOGCM' (find-replace the whole document to check for others which might have slipped through the cracks)

page 7, line 1: start of sentence is missing

page 7, line 11: 'consists in' → 'is'

page 7, line 14: (subject to discussion of $h$ above), you may as well tune $h$ too

page 7, line 20: delete 'In this regard,'

page 7, line 24: lack of quantification of 'suitably mimic' (see discussion above)

page 7, line 31: looking at Lorenz et al. here interrupts the flow of your paper. I would present your results first, then present the results of Lorenz et al. afterwards as a comparison.

page 8, line 18: The sentence starting with 'therefore' doesn't make sense. Do you mean that you are going to present analytic expressions which calculate $\alpha$ and $\mu$ from the inferred ECS and TCR?

page 8, line 30: turn 'square' into $^2$

page 8, line 33: turn 'square' into $^2$

page 9, line 8: swap this sentence with the next one so that you keep the advantages distinct

page 9, line 11: delete 'the ETE not only yields a better approximation;'

page 9, line 15: 'cubic fit' → 'ETE'?

page 9, line 16: 'For the sake of brevity...', maybe better phrased as 'Given the explorations already done and their performance, we leave explorations beyond the linear approximation for future research'.

page 9, line 17: 'beyond' → 'beyond a'

page 10, line 4: 't1' → 't_1' (twice), similarly for k2 and k1 throughout the text in this section

page 10, line 17: I have no idea how you got to this line, more explanation needed (if only in an appendix)

page 10, line 20: 'being inferred from Eqs. (4) and (7)', do you mean 'by solving Eqs. (4) and (7)' for the forcing given above?

page 11, line 2: 'recapitulate' → 'note, by definition,'

page 11, line 5: put manipulation in an appendix

page 11, line 18: a1 formatted incorrectly

page 11, line 22: '1/2 degree' → '0.5degreeC'

page 12, line 12: 'hereby' → 'where'

page 13, line 7: delete 'If the reader will join us in exploring this line of reasoning,'

page 13, line 18: 'an at' → 'at' and 'least 2-box' → 'least a 2-box'

page 13, line 20: 'excellent emulation' without quantification is confusing (see discussion above)

page 13, line 23: discussion in terms of ECS only is confusing (see comments above)

page 14, line 11: 'excellently emulate' without quantification is confusing (see discussion above)

page 14, line 31: 'parsimonious' misspelt

page 15, line 4: 'recapitulate' → 'rearrange'

page 15, line 17: 'whereby' → 'where'

page 15, line 21: 'appropriately mimic' without quantification is confusing (see discussion above)

page 15, line 22: 'Columns' → 'column' and '5th' → '5'

page 15, line 23: 'Over all, the results show that PH99 would be well trained by being calibrated to any RCP scenario.' is a direct contradiction to the paper's statements such as 'its ECS and TCR are re-interpreted as effective, scenario-class-specific values'. I would delete this sentence entirely as I think you've shown it's not true.

---

## Referee Report (RR2)

**General comments**

Repeating my comments from the second round review, Khabbazan and Held's paper highlights the nuances which must be considered when using a one box energy balance model for climate projections (the form they focus on is the one presented by Petschner-Held (1999), herein PH99, but any similar one-box model would exhibit the same behaviour). Their major conclusion is captured in the last paragraph of the paper, specifically that callibrating PH99 is 'much more involved than previously assumed' and hence 'future users should carefully consider whether they actually want to use PH99, or whether they prefer a less parsimonious solution'. On top of this, they also present a lovely bit of analysis which shows why a one box model must use a lower ECS than a two-box model if the two are going to respond similarly to a strong mitigation radiative forcing scenario over an ~200 year timescale.

My major concerns now focus on making the discussions of emulator quality quantitative, in particular removing vague terms, and the representation of one key reference.

The paper presents some very interesting and pertinent results and so, subject to revisions to fix the concerns above, would recommend it for publication.

**Major concerns**

1. I am happy to be corrected on this, but I don't think the representation of Calel & Stainforth (2017) in line 27 of page 2 is fair. The authors say that 'Calel & Stainforth (2017) highlighted the potential future role of PH99, however if and only is users invested in an application-specific re-callibration of PH99 as a valid future approach to emulation.' I cannot work out where this comes from. As far as I can tell, Calel & Stainforth highlight differences in the uses of PH99 in different IAMs and reveal how to resolve them but I can't see any comment about the need for 'application-specific re-callibration'. Perhaps something like this would be a solution, 'Calel & Stainforth highlighted the potential future role of PH99 and hence further validation of its behaviour is warranted.'

2. The discussion of emulator quality is somewhat varied. In places, the authors have been very careful and ensure that they quantified the meaning of terms such as 'a good emulator' or 'a sufficiently accurate' emulator but in other places these terms remain vague. Making sure that the meaning of all such 'quality' terms was clear would help make the point being made clearer to the reader, I feel.

**Specific comments**

1. The difference between your result and the Van Vuuren et al. (2011) result is still not that clear to me (i.e. why do you find the climate components responding too strongly whilst they found them responding too weakly). Is it simply because Van Vuuren et al. (2011) considered emissions driven results whilst you are considering forcing driven results and the emissions to radiative forcing steps are outweighing the forcing to temperature step? Or something else? Making some comment on possible reasons for the difference would help place this article in the context of other work.

2. The introduction is quite long, is it possible to split it or cut it somehow (some sections might be better included in the discussion rather than in the introduction)?

3. A quick check over your treatment of acronyms would be a nice improvement. Sometimes you introduce the term first, and the acronym next e.g. 'integrated assessment models (IAMs)' whilst other times you introduce the acronym first, and the term next, e.g. 'RCPs (representative concentration pathways)'. This is somewhat confusing and it would be nice to make it uniform.

4. Do you have any intention to make your analysis code available?

**Technical corrections**

page 1, line 14: 'emulator (accurate to within 0.1K for mitigation scenarios and the baseline scenarios RCP4.5 and RCP6.0) of these AOGCMs' → 'emulator of these AOGCMs (accurate to within 0.1K for RCP2.6, RCP4.5 and RCP6.0)'

page 1, line 15: 'time horizon' → 'time horizon (on the order of the time to peak radiative forcing)' or something which quantifies which time horizon you're talking about

page 1, line 16: 'We offer a method to re-interpret already published works based on the 1-box model accordingly.' → 'Accordingly, we offer a method to re-interpret already published works based on the 1-box model.'

page 1, line 18: 'claimed' → 'intended' (if that's what you actually mean)

page 1, line 22: 'would comply' → 'comply'

page 1, line 23: 'the most sophisticated' → 'sophisticated' (Earth System Models are arguably more sophisticated)

page 1, line 25: 'foster' → 'offer'

page 2, line 3: 'Van Vuuren' → 'In previous work, Van Vuuren'

page 2, line 26-27: adjust comment about Calel & Stainforth

page 2, line 29: 'correctly' is vague, although given it's in an overarching question perhaps ok (as long as you define the term later)

page 2, line 32: '2target' → 'well below 2target' (although not being a lawyer, I might be wrong on this one)

page 3, line 5: quantify 'for years' or use another term

page 3, line 7: '- inadequate' → '- as inadequate' (although the massive break with the dashes makes it hard to see exactly how the sentence is meant to fit together, perhaps re-write the whole sentence)

page 3, line 8: 'sufficient' → '0.1K' (is this what you actually mean by 'sufficient'?)

page 3, line 13: 'but likely not beyond it', can you check?

page 3, line 21: add comma after 'For that reason'

page 3, line 28: 'the former' → '(ii)' (or do you mean (i) and (ii))

page 4, line 1: quantify the size of the error

page 4, line 6: 'model market.' → 'model market'.

page 4, line 14-16: 'Among others, one of the most extensively used most parsimonious climate emulators is the 1-box global energy balance 15 model, Eq. (1), introduced by Petschel-Held et al. (1999), which projects the atmospheric GMT anomaly compared to its preindustrial level.' → 'PH99 projects the atmospheric GMT anomaly compared to its preindustrial level.'

page 4, line 28: 'propose the 3-step' → 'propose a 3-step'

page 5, line 1: 'good' is vague, quantify or re-word

page 5, line 5: 'such scenarios are not available for' → 'AOGCMs have not been run for 2-target-compatible scenarios for'

page 6, line 7-11: just to check, the reason you don't use the historical period for validation is that you want to focus on purely projection emulation, not model validity in a range of forcing scenarios? This seems slightly odd to me, especially given you can always chose to compare temperature perturbations between convenient reference periods in later quantifications.

page 6, line 29-31: great bit of sensitivity analysis

page 7, line 6: 'the RCP2.6' → 'RCP2.6'

page 7, line 10-13: comments about Paris Agreement and relevance of 0.5K difference could come out much earlier than Results. Perhaps introducing this earlier would help you set out what acceptable 'error thresholds are' and make the scales you're talking about throughout the paper clearer

page 7, line 23: 'whereby' → 'where'

page 7, line 24: 'so' → 'sufficiently', delete 'suitably' (as you quantify later anyway)

page 7, line 25: '0.14' → '0.14K'

page 8, line 1-2: 'Before diving into our suggestions, it might be worthwhile to first take a look at one of the existing options. (However, a reader mainly interested in our improved method of utilizing PH99 might directly move on to Subsection 4.2.)' → 'Before diving into our suggestions, we examine one of the existing options (a reader solely interested in our improved method of utilizing PH99 can move straight onto Subsection 4.2.)'

page 8, line 4: 'the ECS' → 'ECS'

page 8, line 11-17: Can you make some comment on how the TCR is calculated using the Lorenz approach, do you just leave it constant?

page 8, line 19: heading has typo, 'AOGMC' → 'AOGCM'

page 9, line 11: 'better' → 'best'

page 9, line 17: 'Please note' → 'Note'

page 9, line 26-27: Delete 'Hereby we presuppose that a 2-box model emulates an AOGCM qualitatively better than a 1-box model.' I don't think it adds anything here and you have good discussions elsewhere which explain why you are using a 2-box model at all.

page 10, line 6: 'perspective on' → 'projections under'

page 10, line 11: 'find a' → 'approximate a'

page 10, line 14: 'both summing up to' → 'sum equal to'

page 11, line 14: 'resulting in' → 'where'

page 12, line 2: what does 'exact' refer to here, do you mean '2-box'?

page 12, line 18-19: 'In Section 5.1 we derived an analytic explanation why a naïve transfer of an AOGCM's ECS and TCR to PH99 leads to a too large maximum GMT when driven by a mitigation forcing scenario.' → 'In Section 5.1 we derived an analytic explanation for why a naïve transfer of an AOGCM's ECS and TCR to PH99 results in a maximum GMT which is too large when driven by a mitigation forcing scenario.'

page 12, line 19: delete 'could'

page 12, line 20: 'good' is vague, quantify/reference somewhere else/make clearer

page 12, line 25: 'a order-of-magnitudes' → 'an order-of-magnitude' or → 'an orders-of-magnitude'

page 13, line 1: 'Quite' → 'On'

page 13, line 9: this is fine for exploration but a fairly brute force way of making the models agree as far as I can tell

page 13, line 13: 'an as' → 'a'

page 13, line 14: 'We cannot' → 'Hence we cannot'

page 13, line 23: '$t_1$' → '$t_1$, i.e. on the order of the time to peak forcing' (remind the reader what $t_1$ means)

page 14, line 1: delete 'would like to focus on our main finding and'. If you want to make this the main finding, I think you need to re-structure the article as at the moment the main finding is definitely the improved method to transfer AOGCM ECS/TCR onto PH99 ECS/TCR, with this explanation of why such a new method is necessary being a nice bit of supporting analysis.

page 14, line 8: delete 'i.e. for the upcoming 200 years vs. the time horizon thereafter', it adds nothing

page 14, line 12-15: nice quantification

page 14, line 28: 'utilizig' → 'utilizing' (probably worth checking whether you are using American or English spelling throughout, if English then 'utilizing' → 'utilising')

page 14, line 34: delete 'even'

page 15, line 2: missing bracket after '2014'

page 15, line 18: delete 'a first version of'

page 15, line 20-21: '1-box-based model. (Hereby we assume that a 2-box model mimics an AOGCM better than a 1-box model.)' → '1-box-based model (assuming that a 2-box model mimics an AOGCM better than a 1-box model).'

page 15, line 24: 'sensible' → 'useful' (the irony of me correcting this is not lost given I wrote sensible in my previous review, I just think that in the paper useful, or even applicable, fits better)

page 15, line 28: 'the explanation of which' → 'for which the explanation'

page 15, line 30: 'equivalennt' → 'equivalent'

page 15, line 1: delete 'rather would like to'

page 15, line 4-7: perhaps switch full stops for semi-colons in your numbered phrases i..e 'we propose the following steps: (i) By comparison with more sophisticated, multi-box climate modules it should be tested again whether the effect of a transient climate sensitivity (and TCR) alone could explain our observed PH99- AOGCM discrepancy. (ii) Future discussions with the AOGCM community should illuminate to what extent the further explanations we suggested might also apply, thereby potentially reducing the need to correct for PH99. (iii) An' → 'we propose the following steps: (i) By comparison with more sophisticated, multi-box climate modules it should be tested again whether the

effect of a transient climate sensitivity (and TCR) alone could explain our observed PH99- AOGCM discrepancy; (ii) Future discussions with the AOGCM community should illuminate to what extent the further explanations we suggested might also apply, thereby potentially reducing the need to correct for PH99; (iii) An'

page 17, line 13: 'insure' → 'ensure'

page 25, line 1: I think you could combine Figures 3 and 4

page 30, line 1: I think you could simply reference this figure rather than including it in full

page 35, line 1: I think you could simply reference this figure rather than including it in full

page 36, line 1: could you combine this with Figures 3 and 4

---

## Referee Report (RR3)

**General comments**

Khabbazan and Held's paper highlights the nuances which must be considered when using a one box energy balance model for climate projections (the form they focus on is the one presented by Petschner-Held (1999), herein PH99, but any similar one-box model would exhibit the same behaviour). Their major conclusion is captured in the last paragraph of the paper, specifically that calibrating PH99 is 'much more involved than previously assumed' and hence 'future users should carefully consider whether they actually want to use PH99, or whether they prefer a less parsimonious solution'. On top of this, they also present a lovely bit of analysis which shows why a one box model must use a lower ECS than a two-box model if the two are going to respond similarly to a strong mitigation radiative forcing scenario over an ~200 year timescale.

For final proof reading I have included a number of technical corrections below.

Nonetheless, the paper presents some very interesting and pertinent results and I would recommend it for publication.

**Major concerns**

**Specific comments**

**Technical corrections**

page 1, line 12: 'compatible with a maximum global warming of 2 K' → 'resulting in a maximum global warming of approximately 2 K' (as some models project warming greater than 2K for RCP2.6, saying that it's 'compatible with a maximum global warming of 2 K' is too strong, something you implicitly recognise on page 4, line 3)

page 1, line 16: "their" → "the AOGCM's"

page 1, line 17-20: Delete the sentence 'Accordingly, we offer a method to re-interpret already published works based on the 1-box model.' and make the next sentence 'Results that are based on the 1-box model and have already been published are still just as informative as intended by their respective authors; however, they should be re-interpreted as being influenced by a larger climate response to forcing than intended.'

page 1, line 27: 'project' → 'examine' (?)

page 2, line 8: 'deviations' → 'with deviations'

page 2, line 32: 'temperature equation' → 'temperature response to radiative forcing' (?)

page 2, line 33: 'Hereby' → 'Hereafter' or 'In this article,'

page 2, line 34: 'Quite the contrary' → 'Furthermore'

page 3, line 3: 'Hence in the policy domain, a difference in terms of 0.5 K does matter.' → 'In the policy domain, a difference of 0.5 K matters.'

page 3, line 3: 'In fact we believe that further validation is both necessary and possible at a higher level of consistency' → '[New paragraph] We believe that further validation of PH99 is necessary and possible, at a higher level of consistency than has been performed previously.'

page 3, line 6: '2\degree' → '2 K' (to be consistent with the rest of the article)

page 3, line 7: 'displaying' → 'as these scenarios display'

page 3, line 10: '2\degree' → '2 K' (to be consistent with the rest of the article or change all the K to \degree everywhere)

page 3, line 11: 'compared to the ECS' → 'than the diagnosed ECS'

page 3, line 16: 'might require' → 'requires'

page 3, line 20: 'Hence' → 'To resolve this,'

page 4, line 2: For reasons that aren't that clear to me you've written 'Sect.' to mean section in some places and 'Section' in others (e.g. page 4, end of line 10). I would guess this will be picked up in copy editing but I think using 'Section' (capitalised as it's a proper noun) is the most common choice.

page 4, line 3: '2\degree' → '2 K' (to be consistent with the rest of the article)

page 4, line 4: delete 'generically'

page 4, line 5: 'only RCP2.6' → 'only the RCP2.6'

page 4, line 6: ' and use' → ', use'

page 4, line 7: delete ', for the sake of brevity,'

page 4, line 20: 'section' → 'Section' (proper noun)

page 4, line 20: 'to then describe' → 'and then describes'

page 4, line 21: 'for a' → 'for'

page 5, line 8: degree sign

page 5, line 10: degree sign

page 5, line 22: $CO_2$ should have subscript 2 (and throughout rest of manuscript)

page 7, line 16-19: delete from 'A proclaimed goal' until 'does matter' (you've said it before)

page 10, line 27: 'After having reviewed their results for our order-of-magnitude estimates of PH99's accuracy' I'm not sure I understand this, can you double check? Should it just be, 'After having reviewed their results' ?

page 11, line 16: 'request' → 'choose'

page 11, line 18: 'whereby' → 'where'

page 11, line 21: is 'h' the best choice of notation for the auxillary function given you use it to mean 'effective heat capacity' earlier? Can you choose another letter?

page 12, line 11: degree sign

page 12, line 15: which 'maximum'? The 2-box's?

page 13, line 2: 'can be utilized for any RCP' → 'can be utilised for any RCP and the resulting projections are accurate to within X K' (quantify)

page 13, line 6: delete 'again presupposing that a 2-box model would emulate an AOGCM qualitatively better than a 1-box model', you got rid of that statement earlier

page 13, line 10: ', which generically ranges from' → 'ranging from'

page 13, line 13: 'any' → 'all'

page 13, line 14: 'would fix' → 'fixes'

page 13, line 16: 'we cannot expect any longer T = T' → 'we cannot expect that T=T any longer'

page 13, line 16: 'to the solution' → 'to match the solution'

page 14, line 2: 'The calibration' → 'We find that the calibration'

page 14, line 9: delete 'in the course of time'

page 14, line 9: 'Here' → 'In van Vuuren et al.' (or similar)

page 14, line 13: 'Our article highlights the effects of a naively calibrated PH99 on mitigation scenarios.' → 'Our article highlights the effects of naively calibrating PH99 when assessing mitigation scenarios.'

page 14, line 14: 'However, one should not forget about potential additional mechanisms' → 'Additional mechanisms are also possible'

page 14, line 15: 'mapping on' → 'mapping to'

page 14, line 19: delete 'last but not least'

page 14, line 32: ', to a lesser extent, 8.5' → 'approximately 0.2K for RCP8.5'

page 15, line 2: degree sign

page 15, line 3: degree sign

page 15, line 7: delete 'in the rough sense'

page 15, line 30: 'higher' → 'higher by PH99 than by the corresponding AOGCM'

page 16, line 4: degree sign

page 16, line 8: delete 'assuming that a 2-box model mimics an AOGCM better than a 1-box model', this assumption is not introduced anymore

page 16, line 15: 'for which' → 'whose' (reading that again, I think my previous suggestion was a bad one)

page 16, line 20: 'Hereby' → 'Accordingly,'

---

## Author Response (AR2)

Dear Editor,

We are grateful for having been granted the chance to improve our ms based on the referee's comments. The main structural changes are as follows:

* We clarified the scope of our article much more explicitly in the abstract and introduction.

* We improved the presentation and communication of our ms by following the suggestions by the referee and added a new appendix to clarify the derivations.

* We asked two colleagues to read our ms and feedback on the comprehensibility of our ms.

* We highlighted the metric by which we claim the calibrated climate module performs well.

* We implemented all the technical corrections suggested by the referee.

We would be delighted to see that this version of the ms, upgraded along the suggestions of the referee, is found suitable for publication in ESD.

Sincerely yours,

Mohammad Khabbazan and Hermann Held

**Reply on referee report regarding 'On the Future Role of the Most Parsimonious Climate Module in Integrated Assessment'**

First of all, we would like to thank the referee for a very thorough review. We are convinced it helped improving our manuscript (ms). Please find our detailed comments below. The referee's remarks are highlighted by italic font, while ours are left in roman font.

**General comments**

*Khabbazan and Held's paper highlights the nuances which must be considered when using a one box energy balance model for climate projections (the form they focus on is the one presented by Petschner-Held (1999), herein PH99, but any similar one-box model would exhibit the same behaviour). Their major conclusion is captured in the last paragraph of the paper, specifically that callibrating PH99 is 'much more involved than previously assumed' and hence 'future users should carefully consider whether they actually want to use PH99, or whether they prefer a less parsimonious solution'. On top of this, they also present a lovely bit of analysis which shows why a one box model must use a lower ECS than a two-box model if the two are going to respond similarly to a strong mitigation radiative forcing scenario over an ~200 year timescale.*

We agree.

*Having already reviewed the paper in its first iteration, my major concerns now focus on its presentation and communication. In particular, having read the comments by reviewer 3, I am also not convinced that the current form of the paper will allow it to have its maximum impact. Suggestions to address this are below.*

*Despite this, I now feel that the paper presents some very interesting and pertinent results and so, subject to major revisions to fix a couple of errors, as well as to make the structure and messaging suitably clear for readers outside the field, would recommend it for publication.*

**Major concerns**

1. *The paper is still extremely difficult to read (I acknowledge the irony of this comment given that my review is probably also difficult to read). It has some extremely good, and pertinent, points to make but the style means they are far less accessible than they should be. I think a full rewrite is required if this paper is to have the impact that it should. Given that all of the science is done, that rewrite should not take too much time. However, before the paper is published, it should be proofread by at least a few other people in the group as the number of errors/incomprehensible passages which remain in this revision suggest that this step has not yet*

*been suitably taken.*

The new version was checked within our group by two further independent co-workers.

2. *The authors include statements such as 'Over all, the results show that PH99 would be well trained by being calibrated to any RCP scenario.' alongside '[PH99]'s ECS and TCR are re-interpreted as effective, scenario-class-specific values'. These two comments are contradictory; a model cannot simultaneously both be appropriate for use across a wide range of scenarios and require recallibration for each different scenario class. The message would be made far clearer if the authors were to chose some metric of 'emulation accuracy' and use that throughout to explain how well (or not) PH99 is emulating the target timeseries rather than using vague statements such as 'excellently emulate', 'appropriately mimic' and 'suitably mimic'. Alternately, the authors could simply present their results and make no comment on the suitability of PH99 as an emulation tool, leaving the reader to make up their own mind about whether it's 'suitably accurate' or not.*

We changed our wording. We are now more precise and clearly state that we refer to the time-scale of validity. In fact our new analyses show that AOGCMs can be emulated across all RCPs by the identical parameter set. Moreover, we communicate the accuracy numerically (in units of Kelvin).

3. *At the end of the paper, the use cases for PH99 are still not clear. How much faster are one box models like it than two box models? Are there any cases where the increased speed of the one box model over a two box model justifies its use, given how much harder it is to callibrate and its degrading performance outside its callibration range? Put another way, what sort of tradeoffs are made in terms of speed and performance? For example, what is the difference between using e.g. MAGICC, which runs in about a second but has excellent emulation ability, and PH99, which runs in some shorter amount of time (I'd guess a thousandth of a second or even much less) and has rapidly degrading emulation ability outside its callibration scenario type and time horizon.*

We added the savings in computation time in a new paragraph at the end of Section 6.

4. *The discussion of 'constant effective oceanic heat capacity, h' is wrong. See discussion below.*

Here we disagree – see discussion below. However the new ms follows the referee's suggestion and formally starts the derivation from the energy balance relation. We expect this will preclude further misunderstandings.

**Possible ways to help the paper structure**

*Generally, I feel that the paper's argument gets broken at awkward times and that this makes its contributions unclear. I also think that its derivations are unclear and would be greatly helped by detailed supporting appendices/supplementary material.*

*The literature review in the introduction and the discussion of the Lorenz curve*

*in Section 4 are introduced in a way which is particularly jarring to the overall flow of the paper. They interrupt the main flow, which makes it harder to tell which contributions the paper is making and what is existing literature. Shifting such sections around so that they're standalone would make it much easier to see what the contributions of this paper are, and then later compare to existing work.*

We re-arranged the introduction, including its literature review towards the classic style: first, what is the status of the literature? Second, what is our contribution? The previous version was a reply to the very first referee who completely misunderstood the subject of our ms. For this very referee we injected the research question repeatedly and early-on. This referee has meanwhile been eliminated from the process by the editor, and we return to the more traditional style.

Regarding the Lorenz curve discussion, we would like  to keep the reference to this existing literature at the beginning of Section 4. However to circumvent the problem the referee identified, we introduced a separate subsection with the option to bypass this discussion.

*The derivations are at times impenetrable for all but the most dedicated readers. Adding appendixes which walk the readers through  the analytic derviations much more slowly will ensure that readers from outside the field have a much better chance of understanding what is going on and what the papers' equations mean.*

We added Appendix 3 accordingly and expanded the explanation of further equations in Section 5.

*Keeping the comment above in mind, Section 5 needs to be re-written. Section 5.1 makes an extremely important point but it is currently very hard work to get there.*

The derivations should now be much clearer.

*Section 5.2 may also make an important point, but even after multiple reads I am still not sure what it is (that you can do the same conversion between*

*2-box and 1-box models as between AOGCMs and a 1-box model, but that it's not as good?). Some better balance between a plain English summary of what the equations mean, more explanation of the steps taken in the derivation (perhaps in an appendix) and the mathematical derivation itself needs to be made. Perhaps Section 5 could stick to a plain English summary, leaving the mathematical details to an appendix/supplementary material?*

In retrospect we agree that 5.2 was written in too condensed a manner. We expect that the structure of the argument is now transparent by means of expanded explanations and a more transparent version of writing the equations.

**Use of effective oceanic heat capacity**

*Equation (7) is*

$$\mu = \frac{Q_2}{h \ln 2}$$
$$\rightarrow h = \frac{Q_2}{\mu \ln 2}$$

given that $\mu \ln 2$ is the radiative forcing due to a doubling of $CO_2$ concentrations

This is a misunderstanding. *h* is the heat capacity. Hereby we follow Kriegler & Bruckner, 2004 as well as Geoffroy et al., 2013.

*for PH99 (herein $Q_{2PH99}$) then it is clear that h is not an 'effective oceanic heat capacity' as stated in the text but rather a dimensionless scaling factor which is the ratio of the forcing due to a doubling of $CO_2$ in the AOGCM to the forcing due to a doubling of $CO_2$ in PH99 (the units of h make this clear). Hence the correction which is given to allow the authors to use Forster et al's radiative forcing timeseries with PH99 is then*

$$Q_{PH99}(t) = \frac{Q(t)_{Forster}}{h} = Q(t)_{Forster} \frac{Q_{2\,PH99}}{Q_{2\,AOGCM}}$$

*where here we have used Q to denote forcings, rather than a combination of F and Q as in the paper, for consistency.*

*Hence what the authors are actually doing is simply rescaling the entire forcing timeseries by the ratio of the AOGCM's and PH99's forcing due to a doubling of $CO_2$. For non-$CO_2$ forcing, which is important in strong mitigation scenarios, this seems completely inappropriate as it assumes some connection between $CO_2$ and non-$CO_2$ forcing which may or may not be there.*

*The question I ask here is why the authors are using equation (1) as a starting point. Why not simply start with equation (5), or some variant thereof, which takes total forcing as input rather than $CO_2$ only forcing e.g.*

$$\frac{d\Delta T}{dt} = \frac{Q(t) - \lambda \Delta T}{d}$$

*where d is some time constant of the response.*

*This would remove the need for the factor of h and make the entire analysis much clearer and easier? I'm almost certain the conclusions would be unchanged (as this is still a one box model with two free parameters) and you wouldn't need this awkward scaling factor throughout.*

By starting with the standard energy balance equation (our Eq. (4)) the whole argument should now be much clearer. However given PH99 as introduced in the literature there is now way we could come to a different interpretation than the one we give in our ms.

**Specific comments**

1. *'the most parsimonious SCM, PH99, ensures maximum transparency', this is only true if all the parameter settings etc. used to run the model are documented, a practice which isn't yet commonplace. I think solving the transparency problem is related to more than just model choice, namely accessibility of code, documentation of reproduction steps. A more accurate statement may be that, 'the most parsimonious SCM, PH99 (and variants thereof), ensures maximum comprehensibility'.*
   We agree and we utilized the referee's suggestion both in the beginning and at the end of our ms.

2. *The discussion around the impact of directly prescribing AOGCM ECS values to PH99 rather than callibrating PH99 is, in my opinion, confusing. ECS has a strict definition and if you ran PH99 until it reached equilibrium, its ECS would be the same as the AOGCMs'. Hence researchers who have directly transferred AOGCM ECS's to PH99 have not sampled 'effectively higher ECS', they have sampled the ECS they prescribed. What they haven't done is sample the intended centennial climate response. They have sampled a response which is higher than that of the corresponding AOGCM.*

*As a result, I think the implications of the paper would be much clearer if the comments about sampling higher ECS values than intended were all rephrased to focus on sampling higher climate responses to radiative forcing than intended.*

We in fact utilize 'sampling higher climate responses to radiative forcing' in our new ms. Nevertheless, the practitioner needs to know how to interpret all the papers based on PH99. From a practical point of view we find it helpful to offer the interpretation of inverse-transforming the ECS value. In our new ms, we utilize combinations of both statements – the referee's formulation as well as the effective

*This would reinforce the important conclusion of this paper and also highlight that using ECS to characterise climate response on a few hundred year timescale is fundamentally flawed, given that ECS takes on the order of a thousand years*

*to emerge.*

We highlight this very important point in the conclusion.

*I think this would also facilitate a simpler comparison with Van Vuuren (2011)'s findings. The models Van Vuuren considered were sampling lower temperature responses. You have shown that the opposite problem emerges when researches directly transfer AOGCMs' ECS values to PH99, i.e. researchers were inadvertantly sampling higher temperature responses than they intended to. Section 5 shows that this is not because they are sampling a higher ECS (they are, by construction, sampling the same ECS and would see the same long-term response) but because PH99 has a fundamentally different response shape to an AOGCM and hence ECS alone does not allow you to easily move between the two (reflecting the paper's main conclusion). You show that the solution to this is to adjust PH99's ECS, sacrificing agreement in the long-term response in order to gain agreement in the centennial response (which is sensible given it is more often than not the timescale of interest).*

We added this interpretation to our conclusion.

Overall, we became even more cautious to recommend PH99. Our focus is now on highlighting the underlying mechanisms and the need to re-interpret older work based on PH99.

**Technical corrections**

All technical corrections as suggested by the referee (see below) should have been implemented. Furthermore, we now consequentially utilize roman font subscripts for acronyms and numbers, and italic font subscripts for variables, in the tradition of the mathematically oriented geosciences, and against the tradition of economics.

*page 1, line 9: 'MIND and PAGE' → 'MIND and PAGE, widely used in policy making'*

*page 1, line 9: delete 'recent' (in fact I think you could delete the entire sentence, not needed in abstract)*

*page 1, line 12: 'although the model was validated in the past', I have no idea what this is referring to*

*page 1, line 13: 'overestimate mitigation needs and costs' → 'sample a higher range of climate responses than they intend to'? Whilst it's clear that this increases estimated costs, how much is not that clear and so talking about overestimates seems potentially premature. I'm on the fence about this one so this is more of a thought than a recommendation.*

*page 1, line 14: 'produced by' → 'resulting from'*

*page 1, line 15: 'good emulator' → 'good emulator (accurate to within 0.1K)' (it*

*might be more accurate than this, my point here is that having a quantification of 'good' in the abstract would be useful)*

*page 1, line 19: delete 'on the question'*

*page 1, line 22: 'larger ECS than claimed' → 'larger climate response to forcing than claimed' (by definition the ECS is unchanged, your results show that it's just that the centenial timescale response is too high for a one box model with the same ECS as a two-box model)*

*page 2, line 2: 'would' → 'may'*

*page 2, line 4: delete 'Currently'*

*page 2, line 3: delete the first 'thousand' and 'were' → 'is'*

*page 3, line 4: 'Third' → 'Thirdly'*

*page 3, line 6: C missing after 3 degrees*

*page 4, line 5: delete 'might be in order', either you believe the note should be there, or you don't (in which case you can delete the entire paragraph)*

*page 4, line 23: somethign has gone wrong with the brackets at the end of the line (one too many or too few, I can't tell)*

*page 4, line 26: delete '(The right-hand side of the equation has been obtained by utilizing Eq. (2).)'*

*page 5, line 6: delete '(see)' (or fix whatever was meant to be there)*

*page 5, line 8: delete extra comma before the first full stop*

*page 5, line 12: 'multiply it', not clear what 'it' is*

*page 5, line 14: changing from T dot to dT/dt halfway through the paper is extremely confusing, please choose a convention and stick with it. Similarly for F and Q throughout.*

*page 5, line 15: I don't understand what this sentence means*

*page 5, line 22: 'any' → 'all' and delete 'above' (or add something above, there's nothing there at the moment)*

*page 5, line 32: 'beyond CO2' → 'which includes non-$CO_2$ forcing'*

*page 6, line 3: why is 1881-1910 used as pre-industrial? Seems an odd choice for pre-industrial for a perturbation model as it's clearly not a pre-industrial period (although this may not matter in the end)*

*page 6, line 9: lack of quantification of 'tolerable' (see discussion above) makes it very hard to know what you mean here*

*page 6, line 16: if you need to delete something, this paragraph on drift could go. Whilst it is nicely done, it is also a fairly obvious point that PH99 itself acts as a low-pass filter*

*page 6, line 28: 'APGCM' → 'AOGCM' (find-replace the whole document to check for others which might have slipped through the cracks)*

*page 7, line 1: start of sentence is missing*

*page 7, line 11: 'consists in' → 'is'*

*page 7, line 14: (subject to discussion of h above), you may as well tune h too*

*page 7, line 20: delete 'In this regard,'*

*page 7, line 24: lack of quantification of 'suitably mimic' (see discussion above)*

*page 7, line 31: looking at Lorenz et al. here interrupts the flow of your paper. I would present your results first, then present the results of Lorenz et al. afterwards as a comparison.*

*page 8, line 18: The sentence starting with 'therefore' doesn't make sense. Do you mean that you are going to present analytic expressions which calculate α and μ from the inferred ECS and TCR?*

*page 8, line 30: turn 'square' into $^2$*

*page 8, line 33: turn 'square' into $^2$*

*page 9, line 8: swap this sentence with the next one so that you keep the advantages distinct*

*page 9, line 11: delete 'the ETE not only yields a better approximation;'*

*page 9, line 15: 'cubic fit' → 'ETE'?*

*page 9, line 16: 'For the sake of brevity. . . ', maybe better phrased as 'Given the explorations already done and their performance, we leave explorations beyond the linear approximation for future research'.*

*page 9, line 17: 'beyond' → 'beyond a'*

*page 10, line 4: 't1' → 't_1' (twice), similarly for k2 and k1 throughout the text in this section*

*page 10, line 17: I have no idea how you got to this line, more explanation needed (if only in an appendix)*

*page 10, line 20: 'being inferred from Eqs. (4) and (7)', do you mean 'by solving Eqs. (4) and (7)' for the forcing given above?*

*page 11, line 2: 'recapitulate' → 'note, by definition,'*

*page 11, line 5: put manipulation in an appendix*

*page 11, line 18: a1 formatted incorrectly*

*page 11, line 22: '1/2 degree' → '0.5degreeC'*

*page 12, line 12: 'hereby' → 'where'*

*page 13, line 7: delete 'If the reader will join us in exploring this line of reasoning,'*

*page 13, line 18: 'an at' → 'at' and 'least 2-box' → 'least a 2-box'*

*page 13, line 20: 'excellent emulation' without quantification is confusing (see discussion above)*

*page 13, line 23: discussion in terms of ECS only is confusing (see comments above)*

*page 14, line 11: 'excellently emulate' without quantification is confusing (see*

*discussion above)*

*page 14, line 31: 'parsimonious' misspelt*

*page 15, line 4: 'recapitulate' → 'rearrange'*

*page 15, line 17: 'whereby' → 'where'*

*page 15, line 21: 'appropriately mimic' without quantification is confusing (see discussion above)*

*page 15, line 22: 'Columns' → 'column' and '5th' → '5'*

*page 15, line 23: 'Over all, the results show that PH99 would be well trained by being calibrated to any RCP scenario.' is a direct contradiction to the paper's statements such as 'its ECS and TCR are re-interpreted as effective, scenario- class-specific values'. I would delete this sentence entirely as I think you've shown it's not true.*

[revised manuscript text omitted]

---

## Author Response (AR3)

Dear Editor,

We are grateful for having been granted another chance to improve our ms based on the referee's comments. The main changes are as follows:

* We have modified the sentence about the contribution by Calel & Stainforth (2017) according to the suggestion by the referee.

* We have made the discussion of emulator quality clearer throughout the ms.

* We have implemented all the technical corrections suggested by the referee.

We would be delighted to see that this version of the ms, upgraded along the suggestions of the referee, is found suitable for publication in ESD.

Sincerely yours,

Mohammad Khabbazan and Hermann Held

**Reply on referee report regarding 'On the Future Role of the Most Parsimonious Climate Module in Integrated Assessment'**

We would like to thank the referee for another thorough review. It helped improving our manuscript (ms). Please find our detailed comments below. The referee's remarks are highlighted by italic font, while ours are left in roman font.

**General comments**

*Repeating my comments from the second round review, Khabbazan and Held's paper highlights the nuances which must be considered when using a one box energy balance model for climate projections (the form they focus on is the one presented by Petschner-Held (1999), herein PH99, but any similar one-box model would exhibit the same behaviour). Their major conclusion is captured in the last paragraph of the paper, specifically that callibrating PH99 is 'much more involved than previously assumed' and hence 'future users should carefully consider whether they actually want to use PH99, or whether they prefer a less parsimonious solution'. On top of this, they also present a lovely bit of analysis which shows why a one box model must use a lower ECS than a two-box model if the two are going to respond similarly to a strong mitigation radiative forcing scenario over an ~200 year timescale.*

We agree.

*My major concerns now focus on making the discussions of emulator quality quantitative, in particular removing vague terms, and the representation of one key reference.*
*The paper presents some very interesting and pertinent results and so, subject to revisions to fix the concerns above, would recommend it for publication.*

**Major concerns**

*1. I am happy to be corrected on this, but I don't think the representation of Calel & Stainforth (2017) in line 27 of page 2 is fair. The authors say that 'Calel & Stainforth (2017) highlighted the potential future role of PH99, however if and only is users invested in an application-specific re-calibration of PH99 as a valid future approach to emulation.' I cannot work out where this comes from. As far as I can tell, Calel & Stainforth highlight differences in the uses of PH99 in different IAMs and reveal how to resolve them but I can't see any comment about the need for 'application-specific re-callibration'. Perhaps something like this would be a solution, 'Calel & Stainforth highlighted the potential future role of PH99 and hence further validation of its behaviour is warranted.'*

Done.

*2. The discussion of emulator quality is somewhat varied. In places, the authors have been very careful and ensure that they quantified the meaning of terms such as 'a good emulator' or 'a sufficiently accurate' emulator but in other places these terms remain vague. Making sure that the meaning of all such 'quality' terms was clear would help make the point being made clearer to the reader, I feel.*

Done. We have made quality terms clear throughout the ms.

**Specific comments**

*1. The difference between your result and the Van Vuuren et al. (2011) result is still not that clear to me (i.e. why do you find the climate components responding too strongly whilst they found them responding too weakly). Is it simply because Van Vuuren et al. (2011) considered emissions driven results whilst you are considering forcing driven results and the emissions to radiative forcing steps are outweighing the forcing to temperature step? Or something else? Making some comment on possible reasons for the difference would help place this article in the context of other work.*

We are grateful for this hint. We added a paragraph at the beginning of our discussion section and also sharpened our abstract accordingly.

*2. The introduction is quite long, is it possible to split it or cut it somehow (some sections might be better included in the discussion rather than in the introduction)?*

Thanks for your comment. However, we feel that the introduction would lose some important introductory messages if we make it shorter or transfer some sections to discussion.

*3. A quick check over your treatment of acronyms would be a nice improvement. Sometimes you introduce the term first, and the acronym next e.g. 'integrated assessment models (IAMs)' whilst other times you introduce the acronym first, and the term next, e.g. 'RCPs (representative concentration pathways)'. This is somewhat confusing and it would be nice to make it uniform.*

Thanks for your suggestion. We now follow the standard of introducing the term first and the acronym next.

*4. Do you have any intention to make your analysis code available?*

We indicate that the analysis code will be available subject to request from the authors. However, notice that the data availability are not authors' property right.

**Technical corrections**

*page 1, line 14: 'emulator (accurate to within 0.1K for mitigation scenarios and the baseline scenarios RCP4.5 and RCP6.0) of these AOGCMs' → 'emulator of these AOGCMs (accurate to within 0.1K for RCP2.6, RCP4.5 and RCP6.0)'*

Done.

*page 1, line 15: 'time horizon' → 'time horizon (on the order of the time to peak radiative forcing)' or something which quantifies which time horizon you're talking about*

Done.

*page 1, line 16: 'We offer a method to re-interpret already published works based on the 1-box model accordingly.' → 'Accordingly, we offer a method to re-interpret already published works based on the 1-box model.'*

Done.

*page 1, line 18: 'claimed' → 'intended' (if that's what you actually mean)*

Done (intended).

*page 1, line 22: 'would comply' → 'comply'*

Done.

*page 1, line 23: 'the most sophisticated' → 'sophisticated' (Earth System Models are arguably more sophisticated)*

Done.

*page 1, line 25: 'foster' → 'offer'*

Done.

*page 2, line 3: 'Van Vuuren' → 'In previous work, Van Vuuren'*

Done.

*page 2, line 26-27: adjust comment about Calel & Stainforth*

Done, as the referee suggested above.

*page 2, line 29: 'correctly' is vague, although given it's in an overarching question perhaps ok (as long as you define the term later)*

Right after that question we added: '(Hereby 'correctly' refers to an accuracy on the order of magnitude of the standard deviation of natural variability.)'

*page 2, line 32: '2target' → 'well below 2target' (although not being a lawyer, I might be wrong on this one)*

Done.

*page 3, line 5: quantify 'for years' or use another term*

Phrase eliminated.

*page 3, line 7: '- inadequate' → '- as inadequate' (although the massive break with the dashes makes it hard to see exactly how the sentence is meant to fit together, perhaps re-write the whole sentence)*

We agree that the whole sentence reads somewhat convoluted. We re-organized the whole paragraph.

*page 3, line 8: 'sufficient' → '0.1K' (is this what you actually mean by 'sufficient'?)*

Done.

*page 3, line 13: 'but likely not beyond it', can you check?*

We omitted the phrase. Note that we do not have data for checking it.

*page 3, line 21: add comma after 'For that reason'*

Done.

*page 3, line 28: 'the former' → '(ii)' (or do you mean (i) and (ii))*

Done.

*page 4, line 1: quantify the size of the error*

*page 4, line 6: 'model market.' → 'model market'.*

Done.

*page 4, line 14-16: 'Among others, one of the most extensively used most parsimonious climate emulators is the 1-box global energy balance 15 model, Eq. (1), introduced by Petschel-Held et al. (1999), which projects the atmospheric GMT anomaly compared to its preindustrial level.'* → *'PH99 projects the atmospheric GMT anomaly compared to its preindustrial level.'*

Done.

*page 4, line 28: 'propose the 3-step'* → *'propose a 3-step'*

Done.

*page 5, line 1: 'good' is vague, quantify or re-word*

Done.

*page 5, line 5: 'such scenarios are not available for'* → *'AOGCMs have not been run for 2-target-compatible scenarios for'*

Done.

*page 6, line 7-11: just to check, the reason you don't use the historical period for validation is that you want to focus on purely projection emulation, not model validity in a range of forcing scenarios? This seems slightly odd to me, especially given you can always chose to compare temperature perturbations between convenient reference periods in later quantifications.*

Yes. Also, the available data were not compatible for all AOGCM models. Finally, it is current practice to initialize the SCM at the very period which is of interest in the assessment – see e.g. Kriegler & Bruckner, 2004, as well as most integrated assessment studies as assembled in IPCC AR5 WGIII.

*page 6, line 29-31: great bit of sensitivity analysis*

Thanks!

*page 7, line 6: 'the RCP2.6'* → *'RCP2.6'*

Done.

*page 7, line 10-13: comments about Paris Agreement and relevance of 0.5K difference could come out much earlier than Results. Perhaps introducing this*

*earlier would help you set out what acceptable 'error thresholds are' and make
the scales you're talking about throughout the paper clearer*

We agree. We removed that comment in that section and added it in the introduction, in the
same paragraph where we pose our research question.

*page 7, line 23: 'whereby' → 'where'*

Done.

*page 7, line 24: 'so' → 'sufficiently', delete 'suitably' (as you quantify later
anyway)*

Done.

*page 7, line 25: '0.14' → '0.14K'*

Done.

*page 8, line 1-2: 'Before diving into our suggestions, it might be worthwhile
to first take a look at one of the existing options. (However, a reader mainly
interested in our improved method of utilizing PH99 might directly move on to
Subsection 4.2.)' → 'Before diving into our suggestions, we examine one of the
existing options (a reader solely interested in our improved method of utilizing
PH99 can move straight onto Subsection 4.2.)'*

Done.

*page 8, line 4: 'the ECS' → 'ECS'*

Done.

*page 8, line 11-17: Can you make some comment on how the TCR is calculated
using the Lorenz approach, do you just leave it constant?*

Please notice that here there is no need to calculate TCR. However, we added "Notice that
TCR can readily be calculated using Eq. 3."

*page 8, line 19: heading has typo, 'AOGMC' → 'AOGCM'*

Done.

*page 9, line 11: 'better' → 'best'*

Done.

*page 9, line 17: 'Please note' → 'Note'*

Done.

*page 9, line 26-27: Delete 'Hereby we presuppose that a 2-box model emulates an AOGCM qualitatively better than a 1-box model.' I don't think it adds anything here and you have good discussions elsewhere which explain why you are using a 2-box model at all.*

We deleted this sentence.

*page 10, line 6: 'perspective on' → 'projections under'*

Done.

*page 10, line 11: 'find a' → 'approximate a'*

*Done.*

*page 10, line 14: 'both summing up to' → 'sum equal to'*

Done.

*page 11, line 14: 'resulting in' → 'where'*

Done.

*page 12, line 2: what does 'exact' refer to here, do you mean '2-box'?*

We omitted "exact".

*page 12, line 18-19: 'In Section 5.1 we derived an analytic explanation why a naïve transfer of an AOGCM's ECS and TCR to PH99 leads to a too large maximum GMT when driven by a mitigation forcing scenario.' → 'In Section 5.1 we derived an analytic explanation for why a naïve transfer of an AOGCM's ECS and TCR to PH99 results in a maximum GMT which is too large when driven by a mitigation forcing scenario.'*

Done.

*page 12, line 19: delete 'could'*

Done.

*page 12, line 20: 'good' is vague, quantify/reference somewhere else/make clearer*

Done.

*page 12, line 25: 'a order-of-magnitudes' → 'an order-of-magnitude' or → 'an orders-of-magnitude'*

Done.

*page 13, line 1: 'Quite' → 'On'*

Done.

*page 13, line 9: this is fine for exploration but a fairly brute force way of making the models agree as far as I can tell*

It is common practice in integrated assessment modelling to initialize all components not in the pre-industrial phase (for which no adequate economic data exist) but 'present-day' (for the MIND model either 1995, 2010 or 2015). The equation the referee is referring to is then the exact method of initialization within the framework of linear differential equations.

*page 13, line 13: 'an as' → 'a'*

Done.

*page 13, line 14: 'We cannot' → 'Hence we cannot'*

Done.

*page 13, line 23: '$t_1$' → '$t_1$, i.e. on the order of the time to peak forcing' (remind the reader what $t_1$ means)*

Done.

*page 14, line 1: delete 'would like to focus on our main finding and'. If you want to make this the main finding, I think you need to re-structure the article as at the moment the main finding is definitely the improved method to transfer AOGCM ECS/TCR onto PH99 ECS/TCR, with this explanation of why such*

*a new method is necessary being a nice bit of supporting analysis.*

Done.

*page 14, line 8: delete 'i.e. for the upcoming 200 years vs. the time horizon thereafter', it adds nothing*

Done.

*page 14, line 12-15: nice quantification*

Thanks!

*page 14, line 28: 'utilizig' → 'utilizing' (probably worth checking whether you are using American or English spelling throughout, if English then 'utilizing' → 'utilising')*

Done.

*page 14, line 34: delete 'even'*

Done.

*page 15, line 2: missing bracket after '2014'*

Done.

*page 15, line 18: delete 'a first version of'*

Done.

*page 15, line 20-21: '1-box-based model. (Hereby we assume that a 2-box model mimics an AOGCM better than a 1-box model.)' → '1-box-based model (assuming that a 2-box model mimics an AOGCM better than a 1-box model).'*

Done.

*page 15, line 24: 'sensible' → 'useful' (the irony of me correcting this is not lost given I wrote sensible in my previous review, I just think that in the paper useful, or even applicable, fits better)*

Done.

*page 15, line 28: 'the explanation of which' → 'for which the explanation'*

Done.

*page 15, line 30: 'equivalennt' → 'equivalent'*

Done.

*page 15, line 1: delete 'rather would like to'*

Done (page 16).

*page 15, line 4-7: perhaps switch full stops for semi-colons in your numbered phrases i..e 'we propose the following steps: (i) By comparison with more sophisticated, multi-box climate modules it should be tested again whether the effect of a transient climate sensitivity (and TCR) alone could explain our observed PH99- AOGCM discrepancy. (ii) Future discussions with the AOGCM community should illuminate to what extent the further explanations we suggested might also apply, thereby potentially reducing the need to correct for PH99. (iii) An' → 'we propose the following steps: (i) By comparison with more sophisticated, multi-box climate modules it should be tested again whether the*

*effect of a transient climate sensitivity (and TCR) alone could explain our observed PH99- AOGCM discrepancy; (ii) Future discussions with the AOGCM community should illuminate to what extent the further explanations we suggested might also apply, thereby potentially reducing the need to correct for PH99; (iii) An'*

Done (page 16).

*page 17, line 13: 'insure' → 'ensure'*

Done.

*page 25, line 1: I think you could combine Figures 3 and 4*
*page 30, line 1: I think you could simply reference this figure rather than including it in full*
*page 35, line 1: I think you could simply reference this figure rather than including it in full*
*page 36, line 1: could you combine this with Figures 3 and 4*

In the previous stage of preparation of the ms, we tried to combine Figures 3, 4, and 14 into one figure. However, we were not satisfied that the combined figure is clearer enough and more informative. Also, we think that Figures 8 and 13 are important in the context of our ms and should be ready for an interested reader.

[revised manuscript text omitted]